# Soft, miniaturized, wireless olfactory interface for virtual reality

Yiming Liu [1,8], Chun Ki Yiu[1,2,8], Zhao Zhao[3,4,8], Wooyoung Park [1,8], Rui Shi[1], Xingcan Huang [1], Yuyang Zeng[1], Kuan Wang[3], Tsz Hung Wong[1], Shengxin Jia [1,2], Jingkun Zhou [1,2], Zhan Gao[1], Ling Zhao[1], Kuanming Yao [1], Jian Li [1,2], Chuanlu Sha[1], Yuyu Gao[1,2], Guangyao Zhao[1], Ya Huang [1,2], Dengfeng Li [1,2], Qinglei Guo[5], Yuhang Li[3,6] ✉ & Xinge Yu [1,2,7] ✉

Recent advances in virtual reality (VR) technologies accelerate the creation of a flawless 3D virtual world to provide frontier social platform for human. Equally important to traditional visual, auditory and tactile sensations, olfaction exerts both physiological and psychological influences on humans. Here, we report a concept of skin-interfaced olfactory feedback systems with wirelessly, programmable capabilities based on arrays of flexible and miniaturized odor generators (OGs) for olfactory VR applications. By optimizing the materials selection, design layout, and power management, the OGs exhibit outstanding device performance in various aspects, from response rate, to odor concentration control, to long-term continuous operation, to high mechanical/ electrical stability and to low power consumption. Representative demonstrations in 4D movie watching, smell message delivery, medical treatment, human emotion control and VR/AR based online teaching prove the great potential of the soft olfaction interface in various practical applications, including entertainment, education, human machine interfaces and so on.

Recent human-machine interfaces highlight the importance of human sensation feedback, including vision, audio, and haptics, associating with wide applications in entertainment, medical treatment, and VR/AR[1–3]. Olfaction plays a significant role in human perceptual experiences, which is equally important to visual and auditory feedbacks[4–7]. As one of the typical five senses, olfaction has shown a crucial influence in shaping human lives, as most aspects of daily life associated with olfaction coming from manmade materials, industry, transport, household products, etc.[8,9]. For example, olfaction can help us building up a relationship between a mother and a child, picking out the preferred foods, and getting warned of dangers[10]. With great progress in chemistry, biology, and neuroscience, research in odor detection (gas sensors) has subsequently expanded in real-time decoding the components of odor mixtures at extremely low concentrations by both conventional rigid electronics and novel flexible sensing electronics[11,12]. Considering the olfaction feedback/generating, the research is still in its infancy. The current olfaction-generating technologies mainly associate either with big instrument to generate smell in a closed area/ room or in-built bulky VR set, where the wired connection, large demission, dull smell generating function, and low response time make these olfaction feedback methods far behind the development of visual/auditory based VR devices, and thus dramatically limit their application fields[13,14]. Distinguished with these smell-generating/ releasing systems, odor generators (OGs) in wearable formats enable to

[1]Department of Biomedical Engineering, City University of Hong Kong, Kowloong Tong, Hong Kong. [2]Hong Kong Center for Cerebra-Cardiovascular Health Engineering, Hong Kong Science Park, New Territories 999077, Hong Kong. [3]Institute of Solid Mechanics, Beihang University, Beijing 100191, China. [4]China Special Equipment Inspection and Research Institute, Beijing 100029, China. [5]Center of Nanoelectronics, School of Microelectronics, Shandong University, Jinan 250100, China. [6]Aircraft and Propulsion Laboratory, Ningbo Institute of Technology Beihang University, Ningbo 315100, China. [7]City University of Hong Kong Shenzhen Research Institute, City University of Hong Kong, Shenzhen 518057, China. [8]These authors contributed equally: Yiming Liu, Chun Ki Yiu, Zhao Zhao, Wooyoung Park. ✉e-mail: liyuhang@buaa.edu.cn; xingeyu@cityu.edu.hk

create a personalized and localized odorous environment, avoiding odors interference and long response time of smells switching. To date, there are some reports of wearable OGs with the working principle mainly based on commercial liquid atomizers to atomize perfume into tiny droplets for later blowing out[15]. While the clumsy working mechanics, fully rigid package with bulky bottles of liquid perfumes, and special maintenance requirements have indicated the inherent defects of these wearable OGs for realizing high-channel odor generators in a miniaturized, lightweight, and flexible format (Supplementary Table 1)[13,14,16–26]. In addition, a paired intelligent electronic control panel is also essential for tele-operating the OGs with numerous selective odor types according to users' requirements[27–31]. As a result, to provide personalized odors in a small localized area close to users' nose, new wearable olfaction interfacing technologies should exhibit advances as following: (a) the whole system should be built up on a soft substrate in a wearable or even skin-integrated format with miniaturized size and light weight; (b) as many odors with adjustable concentrations and long operation duration to support long term utilization without frequent replacement/maintenance; (c) the olfactory interface should support wireless and programmable operation, capable of interacting with users for various applications. (d) the odor sources should be easy-access and biocompatible. (e) rapid response time in bursting or suppressing odors and accurate odor concentration control are required for the olfaction system in VR/AR applications.

Here, we report a set of materials selection, device designs, integration schemes, and system layouts for wirelessly controlled olfaction interfaces that incorporate arrays of millimeter-scale OGs in thin, soft, and flexible sheets of electronics. The wearable formats of the olfaction interfaces are devised, from direct skin surface mounted above the lips, to face-mask-based electronics, and therefore can establish a bridge between the electronics and users for broad application fields ranging from immersive VR/AR experiences, to message delivery towards disabled users, to emotion control and to medical treatment. Like traditional visual/auditory based VR/AR technologies, the fully programmable olfaction interface offers the interaction between users and virtual world via olfactory stimulation that expands the VR/AR technologies into one more dimension (Fig. 1a). OGs serving as the key elements for the olfaction interface are based on a multilayer stack, including (1) a layer of food grade paraffin mixed with various safe liquid perfume as the odor generating sources; (2) a thermal actuator based on polyimide (PI) supported gold (Au) traces with a thermistor as a temperature controllable heat source for melting and solidification of the odorous paraffin; and (3) a cantilever structured mechanical actuator based on the electromagnetic induction between a copper (Cu) coil and a permanent magnet for thermal management/heat dissipation to realize fast response in turning on/off of the odor generators, as lifting or pushing down the cantilever can heat or cool down the temperature of the thermal actuator monitored by the embedded thermistor (Supplementary Fig. 1). A soft silicone based square frame acting as the shell for the odor generating unit not only offers sufficient room for the mechanical actuator to travel up and down with the thermal actuator, but also allows the whole OG soft enough to be able to mount onto the curved human skin (upper lip) (Supplementary Figs. 2 and 3). The miniaturized feature of the OGs enables to form a large array of OGs with different flavors, where the smell type of the OGs can be distinguished by their frame colors, as shown in Fig. 1e and Supplementary Fig. 4. For instance, darker color-based OGs can symbolize a more pungent smell, such as minty, lemon, etc.

## Results
### Concepts and design principle
Figure 1b and c show the schematic illustrations of the two representative olfaction interfaces, including the skin-integrated Device 1 based on 2 OGs, and the face-mask-based Device 2 incorporating a 3 × 3 OGs array. Both types of devices consist of an array of OGs and an

electronic module as the control panel for wireless data transmission and interaction with computers, whereas the flexible electronics-based control panel adopts a multilayer stacking structure. The first layer associated with a flexible, thin soft Polydimethylsiloxane (PDMS, 2.4 mm, 145 kPa) film as the encapsulation layer to protect the electronics against various external stimulus. For the skin-integrated based olfaction interface, an additional substitutable adhesive layer under the PDMS provides strong physical bonding between electronics and human skin (Fig. 1b, d). For the face-mask-based device, the encapsulated control panel is fixed inside the customized face mask by instant glues (Fig. 1e). The second layer associates with a self-designed flexible printed circuit board (FPCB) interconnecting a series of electronic components, including a microcontroller unit (MCU), a Bluetooth module, resistors, capacitors, a battery, and soft OGs (Fig. 1e, Supplementary Figs. 5 and 6). Due to the very limited skin area of the upper lip, the skin-integrated Device 1 only integrates two OGs, providing two odor options at one time. Despite a small quantity of odors selection, the skin-integrated olfaction interface could be directly mounted onto the curved human skin for programmable smell feedback without any movement burden for long-term utilization (Fig. 1d). Instead, the face mask based Device 2 can integrate 9 different OGs together to realize the portfolio of hundreds odors stimulation by programing the activated numbers, heating temperature and operation time of the 9 OGs (Fig. 1e and Supplementary Fig. 6). Here, more odor options (9 OGs), higher OGs array density (0.88/cm$^3$), faster odor generation response time (1.44 s), and good flexibility enable wider application fields than the reported olfaction systems listed in Supplementary Table 1. The ultrashort distance between both types of olfaction interfaces and the human nose enables the response time as short as 1.44 s under the heating temperature of 50 °C (the melting point of odorous paraffin), as shown in Figs. 1d and 2f.

Supplementary Fig. 7 shows the circuit design for the olfaction interfaces. By integrating Bluetooth modules in the control panel, both the olfaction interfaces could realize long wireless communication distance of up to 2.8 m for Device 1 and 5.9 m for Device 2, which is sufficient to cover the basic requirements during practical applications (Supplementary Fig. 8). For operating the OGs of the two olfaction interfaces, both two control panels require a powering management system that incorporate two batteries (12 V, 1800 mAh; 3.7 V, 2000 mAh) with one Low-dropout regulator (LDO) and a direct-current-to-direct-current (DC-to-DC) voltage boost converter to provide three levels of voltage inputs (3.3 V, 5 V, and 16 V) for the whole circuit operation (see details in Materials and Methods section). The current flow to the thermal actuators is controlled by a Metal-Oxide-Semiconductor Field Effect Transistor (MOSFET) for each OG, and the resistance variation of the thermistor is monitored through an Analog-to-Digital Converter (ADC) and General-purpose input/outputs (GP I/Os) of the MCU continuously during the operation. Therefore, the OGs can efficiently and stably operate at the programmed states, by real-time reading the temperature of the thermal actuators and fast turning on/off of the MOSFET. By adopting the H-Bridge controlled by two digital GP I/Os, the mechanical actuation system of each OG could realize a programmable up-and-down displacement. After collecting the real-time temperature data from each OG, the Bluetooth module can wirelessly transmit serial data between the MCU and paired computers for programmable operation. For the face mask-based Device 2, the 9 OGs independent operation requires additional four 8 bits shift registers to manage 9 MOSFETs and H-Bridges. In addition, an 8-channel multiplexer serves as a control terminal for reading the multiple-channel temperature data of the thermistors. 3D printed thermoplastic polyurethane (TPU) serving as the backbone material of the face mask offers good flexibility, adaptivity, and customizable sizes for various face shapes, and thus realizes a user-friendly olfaction interface (Supplementary Figs. 6e, f, and 9). For aesthetics, the face mask can be also painted into customized colors, such as ash black that

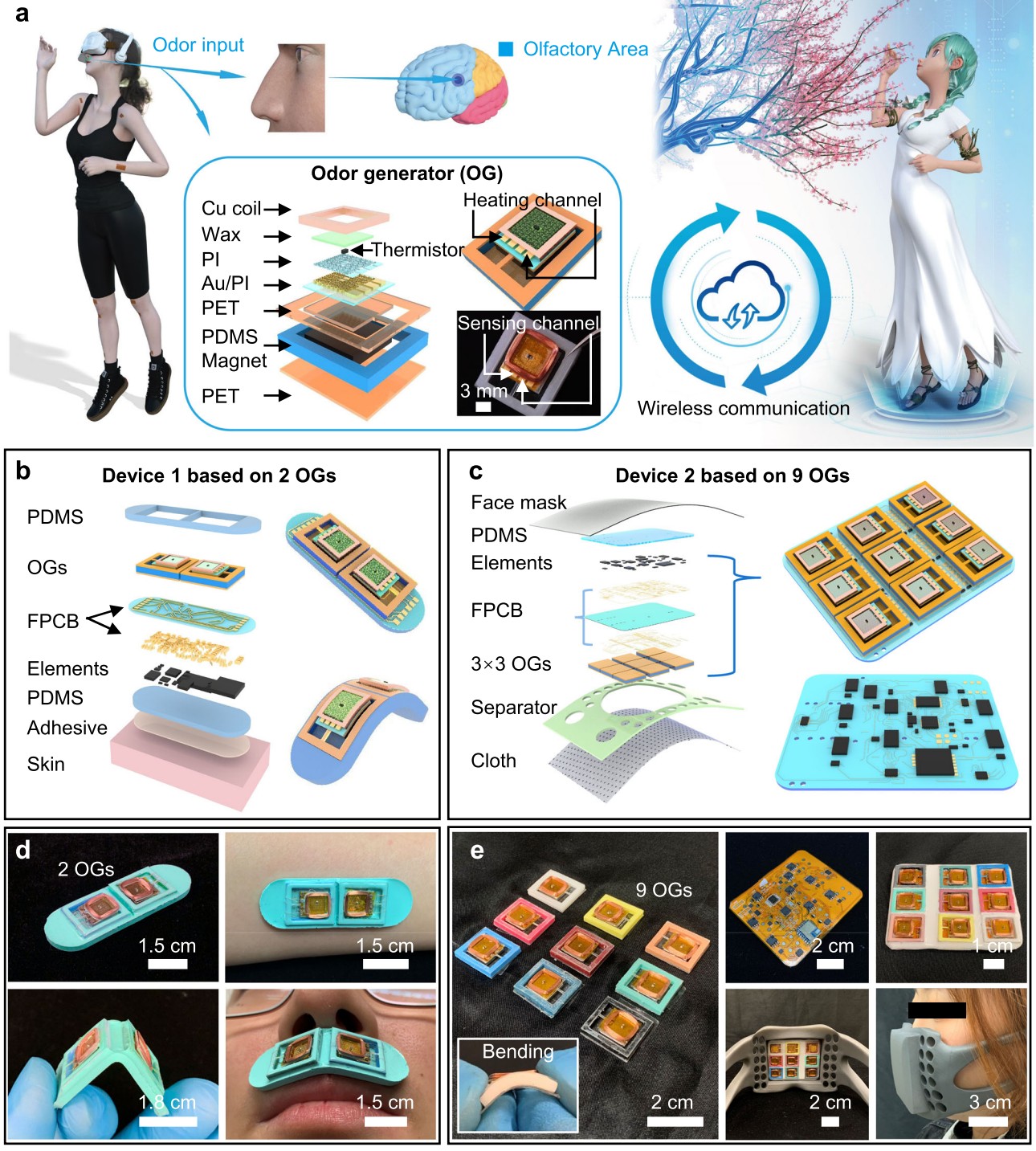

**Fig. 1 | Architectures of the two olfaction interfaces. a** Schematic diagram of the skin-integrated, wireless olfaction interface for providing olfaction feedback to a user for an immersive VR experience. The schematic illustration in the frame shows an exploded view of the OG. **b**, **c** Exploded-view illustrations of the two olfaction interfaces, where Device 1 is in a skin-integrated format based on two OGs (**b**), and Device 2 is built on a flexible face mask platform with 9 OGs embedded (**c**). **d** Optical images of the skin-integrated olfaction interface (Device 1) mounted onto human curved skins under the bending mechanical deformation. **e** Optical images of the face mask-based olfaction interface (Device 2) with 9 OGs, capable of yielding 9 different original odors, where the details of circuit design, device layout, and assembled face mask have been displayed.

VR glasses/set typical use. In addition, the breathing holes in the face mask could ensure users' smooth breathing, which may result in a small number of odor leakage from the face mask.

## Electrical performance of OGs

Optimizations in materials selection, device design, power management, and mechanical optimization of the OG guided by a series of

theoretical and experimental efforts enable the OG to exhibit high efficiency in operation, rapid response time, and accurate in temperature control for the olfaction interfaces (Fig. 2 and Supplementary Figs. 10–12). Figure 2 shows the results of the optimization of the device performance for the OGs, where the investigation of the B value of the thermistors, power inputs of the thermal actuators, and lift displacements of the actuation cantilever in the mechanical actuators

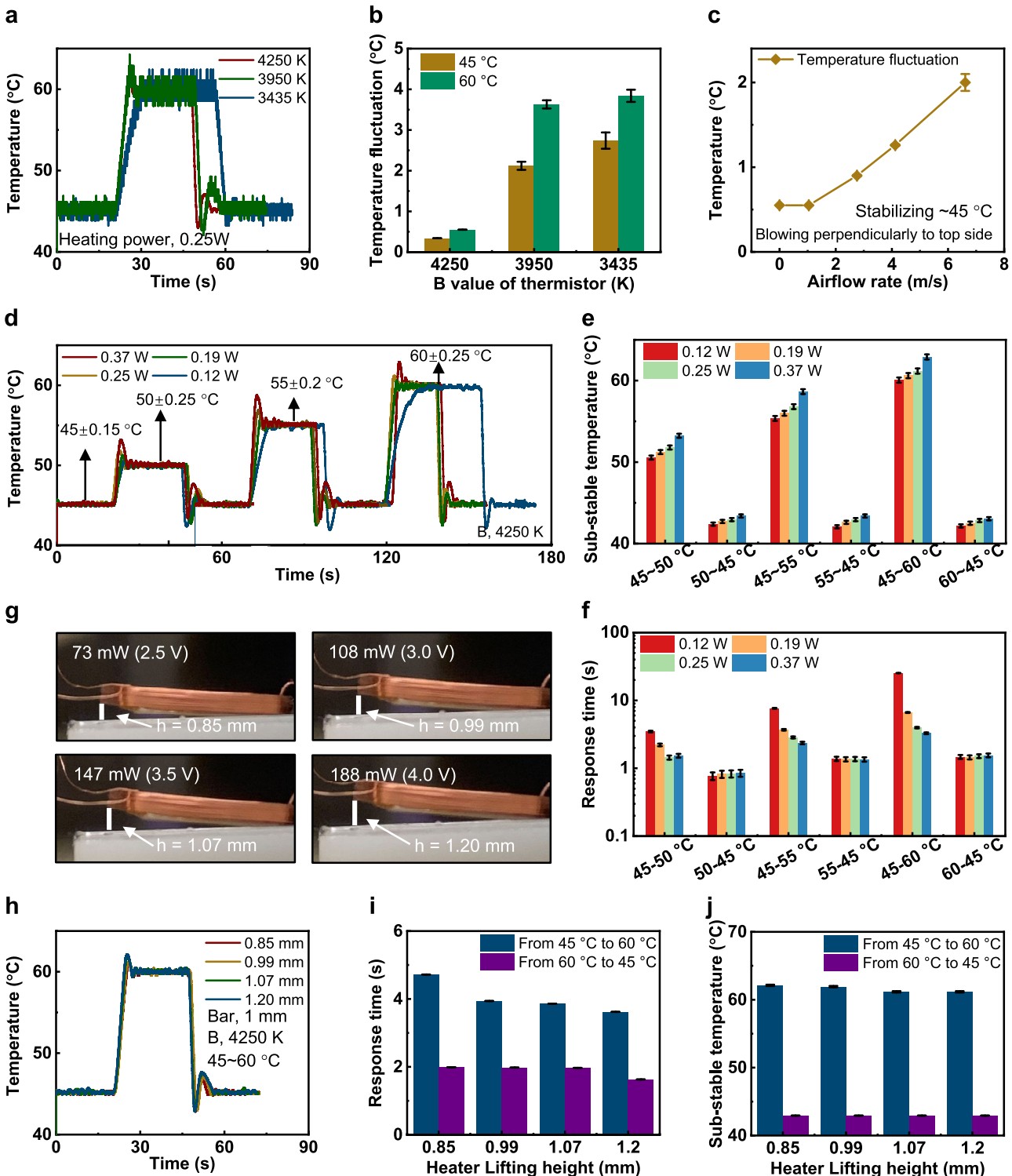

**Fig. 2 | Optimized operation of the OG. a**, **b** Electrical response of the OG as a function of B values of thermistors at a constant heating power of 0.25 W as the heating temperature is switched between 45 °C and 60 °C, and the corresponding analyzed result (**b**). **c** Temperature fluctuation of the OG as a function of air flow rate at a target-controlled temperature of 45 °C. **d**–**f** Electrical response of the OG as a function of heating power ranging from 0.12 W to 0.37 W as the heating temperature is switched between 45 °C and 50 °C, 55 °C, and 60 °C, respectively, with the analyzed results displaying sub-stable temperature (**e**) and response time (**f**). **g**–**j** Optimization of the power input to the electromagnet Cu coil of the OG with the optical images (**g**) and corresponding electrical response of the heating temperature (**h**) with the analyzed results showing the response time (**i**) and sub-stable temperature (**j**). In this figure, all the error bars denote the standard deviation.

can guide us to realize excellent electrical performance for the OGs in terms of working efficiency, power consumption and response time. Figure 2a shows the temperature responses of the thermistors with different B values (3 typical used thermistors with B of 3435 K, 3950 K,

and 4250 K) as a function of the operation time when the temperature of the thermal actuator of the OG switches from 45 °C to 60 °C and then back to 45 °C. It is obvious that with the increasing of the thermistor B values, the temperature fluctuation of the thermal actuators

decreases significantly, from 3.8 °C @3435 K to 0.5 °C @4250 K at 60 °C, and 2.74 °C @3435 K to 0.3 °C @4250 K at 45 °C (Fig. 2b), demonstrating that thermistors with high B are suitable for the OGs to accurately control the temperature (Supplementary Figs. 10j, k, l). Here, we select the thermistor with B of 4250 K as the sensing component, as this thermistor is the one with the highest B that is commercially available. The operational temperature stability of the thermal actuator is the key for the device performance.

To investigate the temperature stabilizing properties of the thermal actuators, tests associating with air perpendicularly blown onto the OGs with 2 mg paraffin wax layer on the heating electrode is shown in Fig. 2c, where the temperature fluctuation increases from 0.3 °C to 2 °C at operation temperature of 45 °C with the increased airflow rate from 0 to 6.6 m/s (typical most in-door situations related range). The study of stand-by temperature versus fluctuation under wind suggest that 45 °C is the optimized number for the stand-by temperature, as the temperature response rate of the OGs exhibits an excellent level of 1.44 s, while the highest fluctuating temperature (47 °C) is still lower than the melting point of the odorous paraffin wax at a high airflow rate of 6.6 m/s. Figure 2d–f presents the relationship of power inputs on the temperature variation of the thermal actuators at the range from 45 °C to 60 °C with detailed illustrations in sub-stable temperatures and response times. It's worth mentioning that the fast response in thermal actuator typically results in over heating or cooling and causes temperature fluctuation, so it needs to balance the temperature fluctuation and response time. Here we defined the temperature fluctuation as sub-stable temperature that represents the highest or lowest temperature of the thermal actuators as raised or dropped to the target temperature, for example, as the temperature of the heating electrode is increased from 45 °C to the target 50 °C, the sub-stable temperature is 51.8 °C for the power input of 0.25 W (Fig. 2e). It is obvious that higher power inputs to the thermal actuators lead to faster response time, however, a balance between response time and power consumption should be also considered. As shown in Fig. 2f, power input greater than 0.25 W allows to boost the response time significantly in heating up, while further increasing the power input to 0.37 W doesn't significantly improve the response time. Furthermore, the response time of dropping temperature down doesn't rely on the power inputs to the thermal actuators (Fig. 2f). In views of sub-stable temperature-caused fluctuation, 0.25 W input power exhibits a much lower temperature fluctuation amplitude than that of 0.37 W, specifically, 51.8 °C@45 °C-50 °C for 0.25 W compared to 53.3 °C for 0.37 W, 56.8 °C@45 °C-55 °C for 0.25 W compared to 58.7 °C for 0.37 W, and 61.2 °C@45 °C-60 °C for 0.25 W compared to 62.9 °C for 0.37 W. The results show that higher input power could induce larger temperature fluctuation during heating temperature drastic variations (Fig. 2e). Therefore, 0.25 W is selected as the optimized input power for the OGs.

Another parameter that needs to be carefully optimized is the power inputs of the mechanical actuators, as the input power to the Cu coil determines the lift height of the mechanical actuation cantilever. The optimization associates with analyzing temperature response of the OGs as a function of the input power to Cu coil, where the input power of 73 mW (2.5 V), 108 mW (3.0 V), 147 mW (3.5 V), and 188 mW (4.0 V) correspond to the lift heights in the actuator of 0.85 mm, 0.99 mm, 1.07 mm and 1.20 mm (Fig. 2g, h). The response time is highly relevant with the lift height, as 0.85 mm lift height results in a longer temperature response time of 4.72 s that of higher lift heights 0.99 mm @ 3.94 s, 1.07 mm @ 3.86 s, and 1.2 mm @ 3.62 s, when the heating temperature increases from 45 °C to 60 °C (Fig. 2i). Sub-stable temperatures of the OGs for the four lift heights exhibit a similar value with the highest fluctuating temperature from 1.2 °C to 2.1 °C (Fig. 2j). As a result, 0.99 mm lift height with the corresponding power of 108 mW is selected as the optimal choice for the OGs considering the balance between electrical performance and power consumption.

Supplementary Fig. 11 shows the temperature of the thermal actuator in the OGs increases from the stand-by temperature (45 °C) to 60 °C at a constant interval of 2 °C, illustrating that the OGs could realize an accurate temperature control. To prove the superiority of the mechanical actuation system, we recorded temperature responses of two OGs with and without cantilever structured in the mechanical actuators at the range from 45 °C to 50 °C, 55 °C 60 °C, respectively (Supplementary Movie 1 and Supplementary Fig. 12). Here, the mechanical actuator serves as the active cooling system for the capability in fast dropping down the temperature of the OGs. By analyzing the heating electrode temperature variations of the OGs without the active cooling mechanics, it can be found that the OGs with the active cooling mechanics realize an extremely fast response time in temperature cooling down, specifically, 0.8 s @50 °C-45 °C for active cooling but 3.9 s without the active cooling, 1.37 s @55 °C-45 °C for active cooling but 6.4 s without the active cooling, and 1.5 s @60 °C-45 °C for active cooling but 8 s without the active cooling (Supplementary Fig. 12b).

Supplementary Fig. 10 shows the performance of the OGs in consideration of temperature response time and the duration of continuous odor stimulation. Careful selection of the paraffin mass and design of the placement angle of the paraffin are two key points to improve the performance of the OGs. Large paraffin wax mass could significantly slow down the response time, due to the longer thermal actuation time to reach the phase transition temperature from solid to liquid (Supplementary Figs. 10a, b). As shown in Supplementary Figs. 10b, c, increasing the pure paraffin wax mass from 2 mg to 30 mg with the corresponding wax layer thicknesses from 0.05 mm to 0.76 mm significantly increases the temperature response time of the OGs from 3.9 s to 12.1 s for 45 °C-60 °C, and 1.5 s to 6.9 s for 60 °C-45 °C. The continuous working capability of the OGs with integrated odorous paraffin wax associates with the recording the odor duration as functions of heating temperatures, embedded odorous perfume types and ratios in paraffin wax mixture, and the paraffin wax mass (see the details in Materials and Methods section, Supplementary Figs. 10d–f). The results show that lower heating temperature, more pungent odors, the higher weight ratio of the perfumes in the mixture, and larger perfume/paraffin wax mass could contribute to a longer odor duration, with a remarkable duration of 52 hrs achieved for 30 mg paraffin wax/perfume (mixture weight ratio, 10: 3) at the working temperature of 60 °C (Supplementary Fig. 10f). Supplementary Figs. 10g–i show the investigation of the overflow time issue for the melting paraffin wax, where the thermal actuator and Cu coil in the mechanical actuator serve as a function of placement angle to the ground, ranging from 90° to 180°, under three different heating temperatures (50 °C, 55 °C, and 60 °C). By investigating the wax overflow time, we could set a heating duration limit for each working OG during operation, which could ensure the stable, long-term operation of the OGs and avoid unnecessary cleaning matters. The OGs with 30 mg paraffin wax at 60 °C heating temperature exhibit the shortest flow time of 40 s under 90° placement, while the flow time is 8 h for the paraffin wax with a mass of 2 mg (Supplementary Fig. 10i). Here, 8 h are the pre-set longest recording time for this testing, which are sufficient for most practical applications.

## Odors generation performance of the olfaction systems

Supplementary Fig. 13 shows the gaseous ethanol generation performance of the OGs. To minimize the unpredictable ambient wind effect on the performance of the OG, the whole experimental setup is placed inside an open box, where a commercial ethanol sensor is fixed 1 cm above a working OG, continuously monitoring the nearby ethanol concentration by reading the voltage variation of a 5 kΩ resistor connected in series (Supplementary Figs. 13a–c). By switching the heating temperature of the OG from 45 °C to 50 °C, to 55 °C, and finally, to 60 °C (Supplementary Fig. 14), the ambient ethanol concentrations

could reach up to 531 ppm, 2821 ppm, and 4531 ppm, respectively, where the corresponding response times in raising up and recovering are 4 s @ 40 s (recovery time, RT), 7 s @ 72 s, and 8 s @ 129 s (Supplementary Fig. 13d). Since the human smell threshold for ethanol concentration is 80 ppm[32], it is obvious that higher heating temperature could contribute stronger olfactory feedback. In addition to the ethanol concentration measurement, we also conducted a human sensory test to investigate the heating temperature effect on the generated odor concentration, as shown in Supplementary Fig. 15. During the testing, all volunteers were wearing Device 2 with 9 working OGs integrated for testing their responses to 9 different odors, where the temperatures of each OG was increased from 45 °C to 50 °C, 55 °C, and 60 °C with the lasting time of 1 min for each temperature point. By summarizing all volunteers' responses, we found that the recognition rates to these smells range from 0.73 to 1 with an average value of 0.93, further proving that the heating temperature of OGs is key for recognizing odor concentration, and the heating temperatures ranging from 45 °C to 60 °C are sufficient to generate odors with the concentrations much larger than the human thresholds. Supplementary Fig. 13e shows the recovery time of generated ethanol concentration in air as a function of parallelly blown wind speed, ranging from 0 to 6.61 m/s, corresponding to a decreasing RT from 129 s to 9 s, which is induced by the increasing gas diffusion rate triggered by the ambient wind. Since the indoor wind speed can reach up to 1.92 m/s, the odor recovery time can be significantly shortened with the assistance of air flow[33]. While using the olfaction interface outdoors is more favorable on the stand point of recovery time, as outdoor typically owns stronger wind. Supplementary Fig. 13f presents the electrical response of the commercial ethanol sensor as a function of the distance between the sensor and a working OG, and it is clear that longer distance could result in a higher delay time to trigger the sensor, including 1 cm @ 1.2 s, 3 cm @ 9.1 s, and 5 cm @ 15.6 s, which is further verified by the volunteer sensory test shown in Supplementary Fig. 16. To further investigate the odor generating performance of OGs insides Device 2, the ethanol sensor is embedded in Device 2 near to the OGs (Supplementary Figs. S13g, h and 18). Here, two different testing conditions are introduced: Condition one is that Device 2 releases ethanol for 5 mins, then shut down meanwhile an experimenter starts wearing the Device 2 until the monitored ethanol concentration recovers to original state (lower than 80 ppm), as shown in Supplementary Figs. 13i, j; Condition two is that Device 2 releases ethanol for 5 mins, then shut down without further motion (Supplementary Figs. 13i, j), where it is obvious that human breathing could shorten ethanol recovery time (1.2 min @ Condition one and 3.1 min @ Condition two). To further investigate the human breathing effect on generated odor recovery time, we conducted a user study test (Supplementary Fig. 19), that also associates with two testing conditions same as that in Supplementary Figs. 13i, j. Here, as there are no available commercial odor sensors to monitor the concentration of the odor adopted in Supplementary Fig. 19, we adopted a volunteer test to obtain the odors recovery time with 11 volunteers involved. In Supplementary Fig. 19, 9 different odors are adopted including lavender (a), orange (b), pineapple (c), green tea (d), lemon (e), peach (f), strawberry (g), minty (h), and lilac (i) with the average recovery times of 114 s @ Condition one and 71 s @ Condition two for lavender, 91 s, and 68 s for orange, 80 s and 70 s for pineapple, 64 s and 56 for green tea, 84 s and 65 s for lemon, 77 s and 66 s for peach, 135 s and 102 s for strawberry, 83 s and 93 s for minty, and 100 s and 99 s for lilac, respectively. Except the minty odor type, the other 8 odor types show a longer recovery time of Condition one than of Condition two. In addition, the overall average recovery time of Condition one is 92.1 s, longer than Condition two of 77.1, which is consistent with the conclusion shown in Supplementary Figs. 13i, j. As a result, it is obvious that the formerly generated odors would not interfere with the newly generated odors after the corresponding odor recovery time (Supplementary Figs. 13 and 19). Supplementary

Figs. 13k, l shows ethanol concentration generated by Device 2 as an experimenter is wearing the device, where the heating temperature of the working OG is targeted at 60 °C. Continuous breathing could obviously decrease the ethanol concentration to 1338 ppm around the ethanol sensor with a large fluctuation (±180 ppm), which also demonstrates that the ethanol generation rate at the heating temperature of 60 °C is larger than the human odor inhalation rate. As the generated odor concentration can be adjusted (Supplementary Figs. 12 and 16), it is possible to avoid odor accumulation around the human nose by adjusting the heating temperature of corresponding OGs in Device 2.

## Stability and safety of the olfaction systems

A series of computational based mechanical designs offer the OGs to exhibit excellent robustness, stability and thermal characteristics (Fig. 3 and Supplementary Fig. 17). In addition, experimental tests associated with the detection of the thermal status and heat dissipation of the operated OG in air by a infra-red (IR) thermal camera further validate the computational results and prove the controlling accuracy of thermal properties of the thermal actuators (Supplementary Figs. 17a, b). Figure 3a and Supplementary Fig. 17c show the mechanical deformation resistance capacity of the OGs in both the PET supporting layer and Au-based heating electrode under the four mechanical deformations, including bending upwards, bending in-plane, twisting along cantilever beam, and twisting perpendicular to cantilever beam. It can be seen from Fig. 3a that the maximum strain of Au electrodes is lower than 0.25% when the device has bent and twisted deformation under the working state without damage, as shown in Fig. 3b. The device can be raised at an angle of 41.5° in bending upward mode and deflected at 12.5° in bending in-plane mode. In addition, the structure can be twisted 19.2° and ± 6.9° along and perpendicular cantilever beam, respectively. Subsequently, a series of stability tests performed on the OGs further prove their good tolerance against the external mechanical stimulus, as shown in Fig. 3c–g. Figure 3c shows the heating temperature response of the OG as repeatedly switching the temperature between 45 °C and 50 °C for over 7000 working cycles, which is realized by a self-developed circuit (Supplementary Movie 1), with enlarged data details in Fig. 3d. Along the long-term, continuous operation, the heating temperature signals of the OGs maintains stable and consistent values without any interruption, demonstrating its high stability and robustness. In addition to the long-term operation, vibration-induced uncurtains are always a great concern as the wearer may keep moving. Figure 3e and Supplementary Figs. 20 and 21 show two typical stability tests for providing vibration to a working OG with controllable vibration amplitude, frequency, and bending angles by adopting a commercial oscillator (Supplementary Fig. 20) and a self-developed programmable bending platform (Supplementary Fig. 21). As a result, millimeter-scaled vibration with an adjustable frequency ranging from 0 to 10 Hz doesn't bring obvious interference on the electrical performance of the OG (Fig. 2f, g), but the wide angle and range of rotation could induce a slight temperature fluctuation of 1 °C to the OG (Supplementary Fig. 21), which is resulted from the acceleration of the heat dissipation during the wide range movement induced forced convection heat transfer between the OG and the surrounding air[34]. However, this negligible fluctuation of temperature doesn't cause obvious performance deterioration during operation. Supplementary Fig. 22 demonstrates the high stability of Device 1, where the device is fixed on a programmable bending platform for 2000 continuous bending cycles at the constant frequency and angle of 0.33 Hz and 40°. During the bending process, Device 1 wirelessly sends the voltage value into the electromagnet coils of OGs integrated to the paired receiver, and stable voltage input (3 ± 0.02 V) proves the excellent stability of Device 1. To demonstrate the flexibility of Device 1, the device is mounted onto human upper lip with different mouth motions, and enlarged optical image details prove the tight

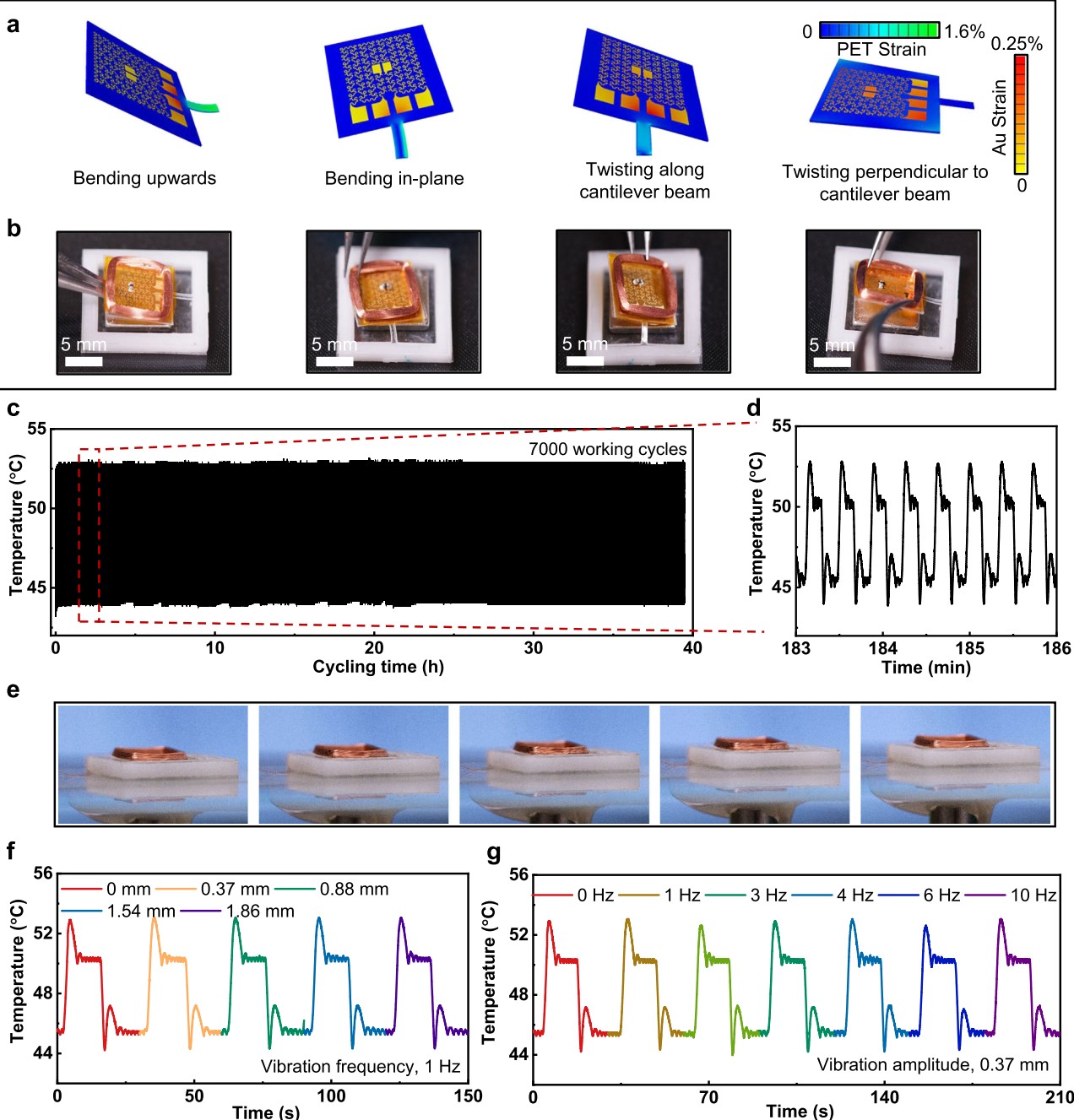

**Fig. 3 | Stability testing of OGs. a, b** Finite element modeling (**a**) and corresponding optical images (**b**) of the OG in bended upwards, bended in plane, twisted along cantilever beam, and twisted perpendicular to cantilever beam configurations. **c, d** Electrical response of an OG operating for over 39 h (**c**) with the enlarged details shown in (**d**). **e** Optical images of OGs fixed on a programmable vibration platform for investigating vibration effect on its electrical performance. **f, g** Vibration amplitude (**f**) and frequency (**g**) on the OG electrical performance.

attachment of the device onto human skin during various skin deformations (Supplementary Fig. 23). Supplementary Fig. 24 shows the safety testing of the olfactory devices as the heating temperatures of the OGs are stabilized around 60 °C. The thermal images of a working OG illustrate the fact that the ambient air temperature with the distance of 1.5 mm from the working OG is around at room temperature (Supplementary Figs. 24b, c). Then, we measured the distance between human nose to working OGs from 11 volunteers (Supplementary Fig. 24d). It is clear that all volunteers (including 4 males and 7 females) could safely wear the Device 2 with all the distance values greater than 1.5 mm (Supplementary Figs. 24e, f). For the Device 1, the open design allows the OGs to exhibit good thermal convection and also to

maintain a safe distance to users with a big gap (23 mm) between OGs and user's nose, since the temperature at the skin interface is only 32.2 °C, as shown in Supplementary Figs. 24g, h.

## Demonstrations of the olfaction systems
The olfaction interfaces based on flexible OGs can be utilized in various scenarios ranging from 4D movies, smell message delivery, medical treatment, VR/AR experiences, and emotion controls (Fig. 4). According to the requirements on the odor supply number and types, users could select either the skin-integrated small scale olfaction interface or the face mask based plus version. The skin-integrated, light-weight Device 1 could be directly mounted onto human skin near

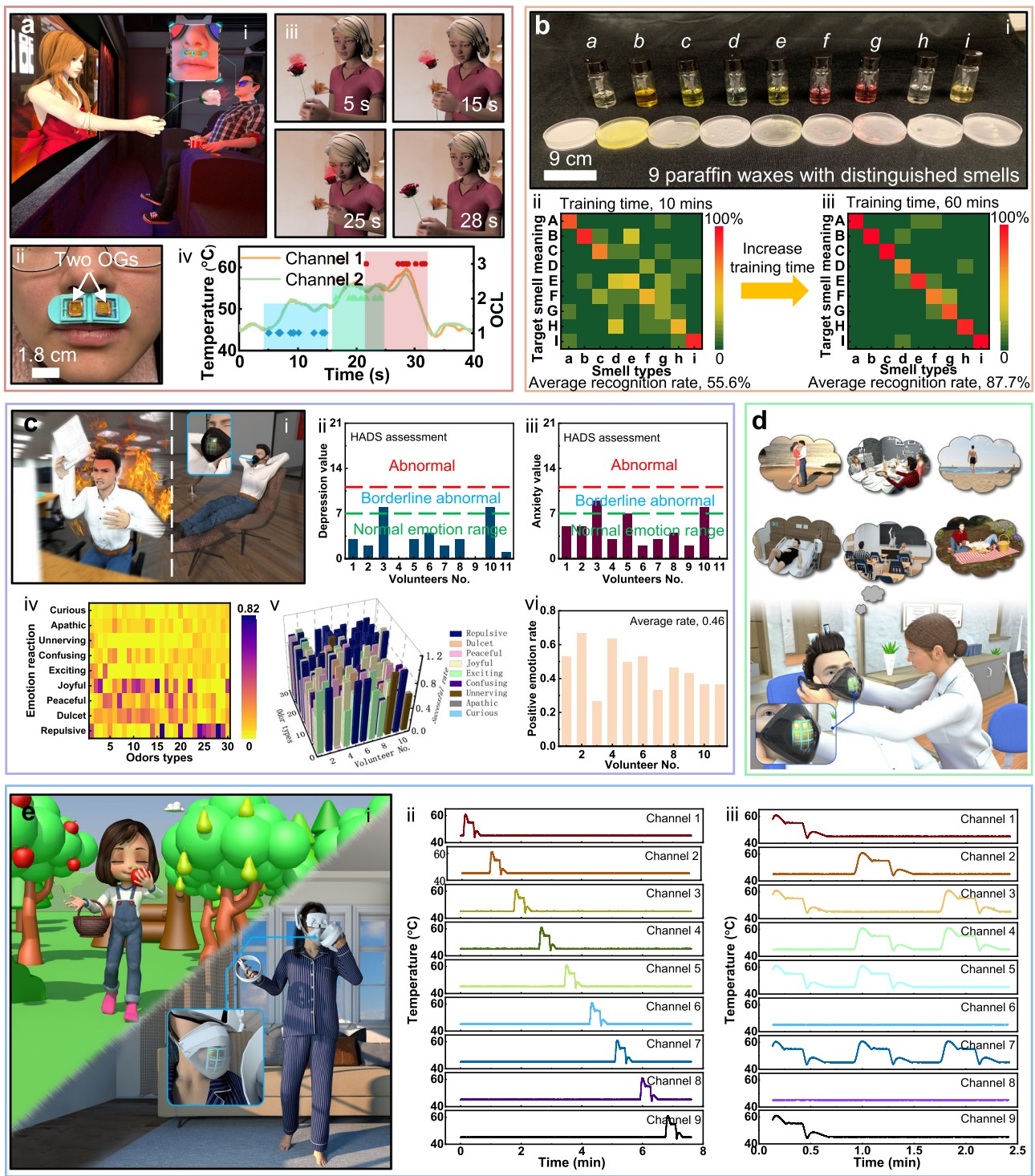

**Fig. 4 | Demonstrations of the olfaction interfaces. a** A demonstration of the skin-integrated Device 1 in displaying olfaction feedback for providing an immersive experience to users during movie watching. Here, a girl in the movie is smelling a flower with the decreasing distance between the flower and her while the watcher could smell a more and more intense floral odor by wearing the olfaction interface. Here, OCL demonstrates the odor concentration level ranging from 1 to 3, corresponding to the low, middle, and high levels (**iv**). **b** A demonstration of the face-mask based olfaction interface in providing an alternative communication method, smell messages, for the disabled without the capability of vision and audio. By training users to link odors to specific information, the disabled users could efficiently communicate with others. **c** A demonstration of the Device 2 in assisting users to control their emotions as some specific odors could recall different human emotions. Here, all volunteers involved in the emotion reaction test are verified in normal emotion range (**cii** and **ciii**). **d** A potential demonstration of the Device 2 for helping amnesic patients recall lost memories as odors perception is modulated by experience, leading to the recall of emotional memories. **e** Demonstration of the wireless olfaction interfaces in real time interaction between the user and a virtual subject for walking in a virtual garden surrounded by various fruit fragrance. By programming the Device 2, 9 OGs inside the device could work independently (**eii**), or work together (**eiii**).

the nose with two different basic odor types for fast olfaction display, while the face mask-based Device 2 integrating 9 different original odors could also be worn in a comfortable way, capable of creating a personalized, localized odorous environment. Figure 4a and Supplementary Movie 2 show a representative application of the skin-integrated olfaction interface for providing an immersive 4D movie watching experience (Fig. 4ai). Here, the subject in the movie is smelling a flower as it is approaching to her step by step while the OGs heating temperature simultaneously increases with the decreasing of the distance between the odor source and the subject nose, releasing a growing fragrance of roses for users in real world (Fig. 4aii and 4aiii). In the beginning of the movie as the flower is far away from the subject, the two OGs are in a state of stand by (heating temperature, 45 °C), then the heating temperatures start rising step by step from 45 °C, to 50 °C@5 s, to 55 °C@15 s, to 60 °C@25 s, then back to 45 °C@28 s (Fig. 4aiv). Here, we conducted a volunteer sensory test to investigate if users could react to fast temperature variation of OGs integrated in Device 1 (Supplementary Fig. 25). Following the time points in increasing the heating temperature of OGs shown in Fig. 4aiii and iv, all volunteers' reaction times to variating heating temperature of two OGs in the Device 1 are recorded in Fig. 4aiv and Supplementary Fig. 25, including 9.7 s (reaction time) @ 5 s (time point in increasing heating temperature), 20.8 s @ 15 s, and 28.4 s @ 25 s. As a result, our Device 1 could generate distinguished odor concentration to users, further demonstrating the great potential in movie watching. Supplementary Fig. 26 shows a volunteer sensory test result in judging the presence or absence of odors in silent OGs at room temperature. It is found that the recognition rates range from 0.22 to 0.89 with an average value of 0.43 for all volunteers, which demonstrates that human beings have distinguished odor concentration thresholds. In addition, different odor types also have distinguished recognition rate, ranging from 0.09 to 0.82. A second representative application of the olfaction interfaces is the message delivery through smell for a visual/auditory impaired person (Fig. 4b). By training visual/auditory impaired users to link some odors to specific events, such as lavender odor@ walking outsides, pineapple odor@ having a lunch, and green tea odor@ teatime, the users could simultaneously react to the released odor from the face mask-based olfaction interface with 9 different odor options. During the training time, all the volunteers are required to continuously smell the odors generated by Device 2. The training associates with short-term (67 s for each odor, total of 10 min) and long-term (400 s for each odor, total of 60 min). When taking the test, the volunteers only have one chance to give their response to the generated odors with no fixed testing time. By adjusting various combinations of the 9 original odors, the olfaction interface is capable of offering extensive messages for users efficiently after training them a certain time. To prove the training time effect on the messages recognition rate, 9 volunteers worn the olfaction interface system with their eyes blinded and ears plugged for later 9 basic smell messages recognition test. Here, volunteers are asked to continuously smell 9 odors with equal training time for each odor during the training process. As a result, the average recognition rate of the 9 smell messages is 55.6 % after a very short training time (10-min), while a longer training time (60-min) can significantly increase the recognition rate to 87.7 % (Fig. 4bii and biii). In this demonstration, we developed the smell message delivery system based on the OGs array instead of single OG due to the facts: (1) human beings could easily distinguish different odor types, where one odor type corresponds to one message. (2) the delivered messages quantity could be significantly enhanced by blending several odors together, which could easily be realized by our Device 2, as shown in Fig. 4eiii. Although the single OG could also deliver smell messages by generating different odor concentration for users, it is difficult to build up a standard odor concentration division for different users who has distinguished odor concentration thresholds, as shown in Supplementary Fig. 26. Figure 4c shows another

application scenario where some odors generated from the olfactory interface system could be used for smoothing users' emotion. As odors could arouse human emotion by leading to the recall of emotional memories, the olfactory interface could be adopted for smoothing users' depressed mood from the stress[35] (Fig. 4ci). To verify the effect of odors on human emotion, 11 volunteers' reactions to 30 different odors (see details in Characterization) from the face mask based olfactory interface were collected as shown in Fig. 4c (all volunteers were in normal emotion before the testing, Fig. 4cii and iii), and the result demonstrates that the possibility of being joyful for these volunteers could reach up to 56%, 65%, 44%, and 56% when they smelled the grape, peach, orange, and strawberry odor, respectively (Fig. 4civ, v). It is also possible to give other emotions to users as they smelled clove and gardenia odors, inducing repulsive and peaceful emotions with the corresponding possibility of 0.82 and 0.51, as shown in Fig. 4civ. In addition, it is obvious that different users have distinguished emotion reactions to the same odors (Fig. 4cv, vi), which may be due to the volunteers' different emotional memories for same odor type. The large emotion reaction difference for each user may increase the difficulty in giving various desired emotions to users. Fig. 4cv shows the volunteers' emotional reaction to 30 odor types, where the results were collected from five times test for each volunteer. It is found that for each volunteer, their emotion reactions to some odor types keep constant, demonstrating that we could customize the odor types in the Device 2 for different users. Evidence has witnessed a fact that reinforcing olfactory stimuli could activate human brain areas such as the orbitofrontal cortex (OFC), meaning that odors can be particularly effective stimuli for amnesic patients to recall their lost memories, which is another one representative application for the olfaction interfaces[4] (Fig. 4d). Another important typical application of the olfaction interfaces is in VR/AR, as which arise extensive attentions recently, especially, since the conception of Metaverse for building a 3D virtual world in connection with real people through a wearable VR interface. To provide an immersive experience for users, the odor feedback provided by olfaction interfaces could enrich the functions and entertainment of the VR interface. Fig. 4ei shows a girl wearing a series of VR devices containing our olfaction interface at home for interacting with virtual environment, where the VR scenario associates with walking in a virtual garden surrounding by various plants, including apple trees, pear trees, etc. By picking up an apple near to the virtual subject nose, the user could synchronously smell apple odor generated by the olfaction interface. Fig. 4eii, iii show the electrical response of the face mask-based olfactory interface with 9 OGs operating independently or at the same time for proving the capability of the olfaction system in coordinating each OGs working independently. To realize the 9 channel OGs independent operation, four 8 bits shift registers (74HC595, Texas Instruments Inc.) and an 8-channel multiplexer (ADG708BRUZ-REEL7, Analog Devices Inc.) are connected to the MCU for extending the digital GP I/Os and digital pins of the microcontroller, capable of controlling 9 OGs independently. To ensure the stable operation of the circuit, the operating OGs number is limited to 5 as the potential large ripple voltage and coupled electromagnetic field induced by the multiple working OGs (>5) may affect the operation of the microcontroller. While blending 5 odors simultaneously can provide a great portfolio of odors feedbacks.

COVID-19 has raised global panic for its highly contagious and pathogenic virus features, forcing educational institutions to turn the face-to-face teaching mode into remote online teaching[36,37]. As the next generation remote teaching method, the online VR instrument could reshape the traditional face-to-face education model, enabling the direct, efficient communications between teachers and students. Figure 5a shows a typical application of our olfaction interfaces, where a teacher is stating the classification of various plants with corresponding characters, including odor and appearance. Here, the

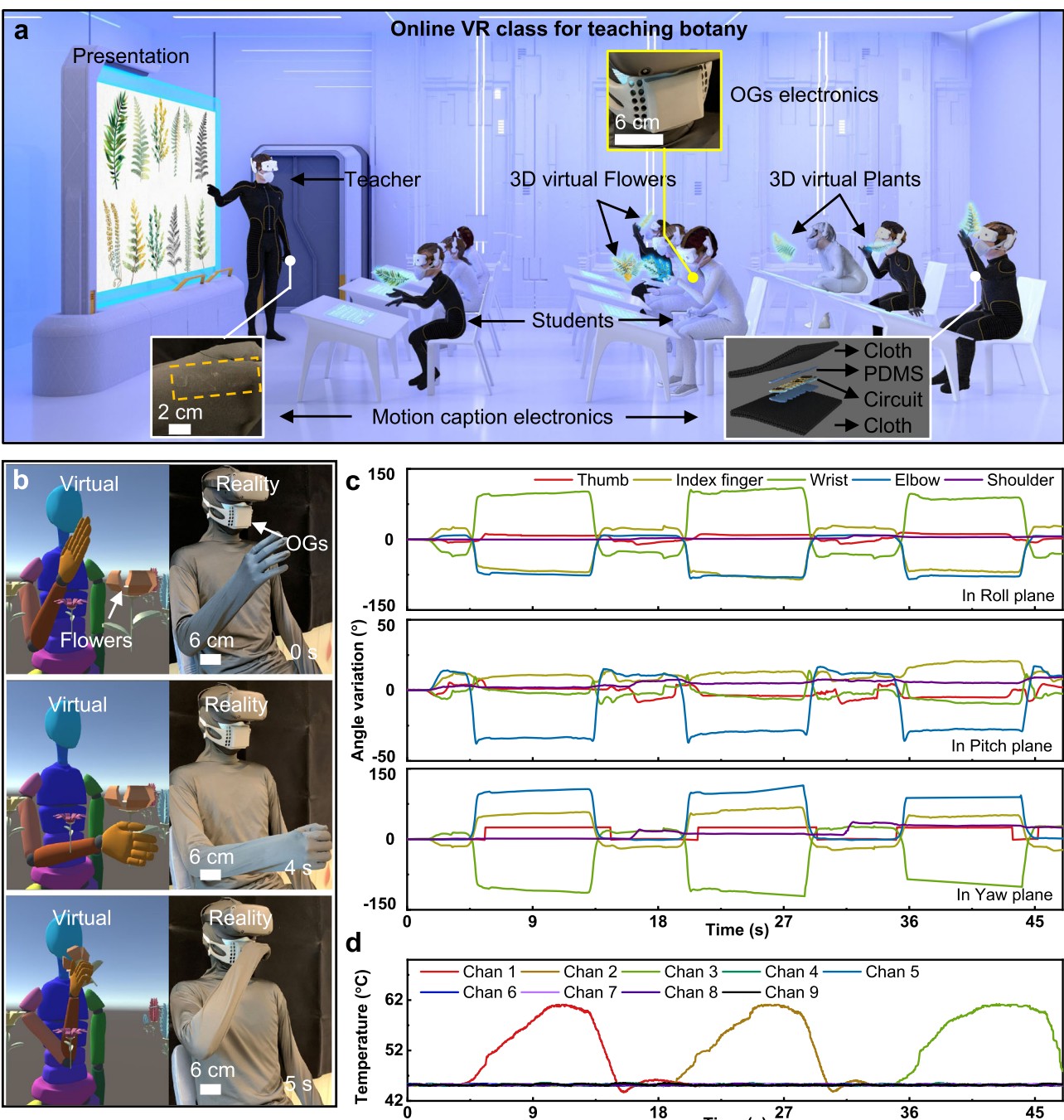

**Fig. 5 | Application the face mask-based olfaction interface in virtual online teaching. a** A teacher is teaching botany in a VR class with all people wearing a series of VR interface, including the VR glasses for vision feedback, a cloth integrated motion caption system, and a face-mask-based olfaction interface for olfaction display. Here, students could smell the fragrance of various studied plants through the olfaction interface. **b–d** Demonstration of the entire VR interface physically connecting a user to the virtual subject for manipulating it to pick up flowers in front of him, then put them near to the nose for smelling (**b**) with the electrical responses of motion caption electronics (**c**) and olfaction interface (**d**).

students could experience these odors by wearing the face mask-based olfaction interface. To real-time capture users' motion for manipulating the virtual subjects in the 3D virtual classroom, a state-of-art soft activity tracking system is developed based on accelerometers (Fig. 5a and Supplementary Figs. 27–29). For operating the activity tracking system, a programmable MCU is utilized to receive 3 Euler angles (roll, yaw, and pitch) from each accelerometer chips through serial peripheral interface (SPI) communication (Supplementary Fig. 27c). By calculating the relative angle change between the two accelerometer chips in real time, the activity tracking system could

continuously detect its bending angle, where one accelerometer chip serves as a reference for stabilizing the data of the motion in any displacement. Then, the processed angle data is transmitted wirelessly to a computer through a Bluetooth module for a VR application. As a result, the self-developed activity tracking system could accurately detect the bending angle variations of human joints, such as fingers, wrist, elbow, and shoulder (Supplementary Fig. 27d–f). To further demonstrate the stability of the activity tracking system, the device is continuously bent between 0 and 60° for over 13500 cycles without obvious accuracy decay (Supplementary Figs. 27g, h). By adopting an

80 mAh Li-ion battery as the power source, the motion caption electronics could maintain normal operation for over 100 mins, which is sufficient for a lecture time (Supplementary Fig. 27i). Supplementary Movie 3 and Fig. 5b–d show a user wearing an olfaction interface, a VR glass, and the activated tracking system manipulating a virtual subject to pick up virtual flowers hanging in front of him. As a flower is approaching to virtual subject nose, the heating temperature of the corresponding OGs will increase linearly, releasing the odor to the user (Fig. 5c, d), which further proves the possibility of the next generation VR online teaching.

In summary, the soft olfaction interfacing systems reported in this work exploit the two wearable formats where one device could realize an extremely short distance between OGs and users' nose by directly mounting onto human upper lip for fast olfaction display, and the other one built on a flexible face mask platform provide a localized environment with hundreds of odor options. Based on flexible, miniaturized OGs, the two olfaction interfaces are capable of building a wireless connection with 3D virtual worlds with programmable odor concentration control in real time. Working principles, operation logic, and wearable formats have largely distinguished this technology with a few previous attempts. Extensive electrical performance experiments, theoretical studies, and stability testing on the soft OGs ensures the stable and sensitive odor supplies in various unexpected situation. Demonstrations in 4D movie watching, smell message delivery, medical treatment, human emotion control, and VR/AR experience further exhibit the great potentials of the olfaction interfaces in various application areas ranging from entertainment, to education, to healthcare, and to human-machine interfaces.

The improvement of the system we could make in the future is illustrated in the following: The long odor remaining time could induce a long delay time when users need switching odor types frequently (Supplementary Figs. 13d, e, i and 19). To solve this issue, miniaturizing the overall size of OGs and enhancing airtightness of the olfaction systems would be helpful. In addition, smaller size of OGs could also realize a higher channel OGs array.

## Methods

### Fabrication of the OG

The fabrication first commenced with cleaning a quartz glass using acetone, alcohol, and deionized water (DI water) consecutively. By adopting double-sided tape, a thin layer of polyimide (PI) with a thickness of 25 μm was attached to the glass sheet. Following, Au/Cr (200/40 nm) were deposited onto the PI film by magnetron sputtering using a machine (Q150TS, QUORUM) and then, through photolithography and etching, yielding metal patterns in the desired geometries. For the photolithography, a positive photoresist (PR, AZ 5214, AZ Electronic Materials) was spin-coated at 3000 rpm for 30 s, soft-baked on a hot plate at 110 °C for 4 min, and then, exposed to ultraviolet light for 5 s. After that, the photoresist coated was developed by a developer (AZ 300 MIF) for 15 s, and then, by using acetone and DI water, the PR was rinsed away. After that, spin casting a layer of PI (2 μm, 3000 rpm for 30 s, annealed at 250 °C for 30 min) and then selectively etched by Oxford Plasma-Therm 790 RIE system (patterns defined by photolithography similar as the previous step) at the power of 200 W for 10 min, formed encapsulation layers for serpentine-designed Au/Cr traces while exposing the connecting patches for later wired connections. After rinsing the PR away, we could obtain the heating electrode. Fix a thermistor (QN0402X104F4250FB, Advanced Materials Electronics Co., Ltd) onto the reserved patches in the middle of the heating electrode by silver paste, then a copper coil is directly attached onto the heating electrode with the interface sealed by instant glue. Here, the thermistor (1 mm × 0.5 mm × 0.5 mm) is a ceramic semiconductor based on transition-metal oxides, showing a negative temperature coefficient of resistance. Due to the miniaturized size, high-temperature sensitivity (Fig. 2a, b), and excellent accuracy

(±0.1 °C)[38], the thermistor is adopted as the temperature sensor for real-time monitoring the heating temperature of OGs. After the glue cured, stick the sample onto a prepared PET platform fabricated by a laser cutting machine. Then, fix the sample onto a chamber with a magnet embedded in the middle, where the back cover is a flat PET film, and the supporting wall is the flexible PDMS ring (Supplementary Fig. 2). Finally, put 2 mg odorous paraffin wax into the heating chamber, then melting it by heat gun at 60 °C for 5 s. The mass of the OG is 1.8 g.

### Fabrication of the olfaction interfaces

Flexible circuits were developed by printed circuit board processing techniques based on copper (Cu, thickness 10 μm) plated with gold (thickness 50 nm) as the conductive layer. Furthermore, the Cu layer was applied to both sides with exposing circle electrodes and square patches. Low-temperature soldering paste (LF999, KELL YSHUN Technology Co. LTD) was used to bond and electrically connect all of the components, including the Bluetooth (WH-BLE103), the microcontroller (ATmega328p-mu), capacitors (14-22 pF), resistors (0.8 MΩ), crystal oscillator (16 MHz), and odor generators, to corresponding contact pads on the Cu/PI substrate. After the integration of all components, PDMS (crosslink: PDMS = 1:30, 145 kPa, 2.4 mm thick) was poured onto the device fixed in a customized mold, followed by curing at 110 °C for 5 min. Then, we could get the skin-integrated olfaction interface and the control panels of face-mask based olfaction interface. The weight of Device 1 is 7.4 g. To assemble Device 2, the control panel is fixed insides the chamber of the customized, flexible face mask by instant glue. After that, connect the power wires to the prepared power management circuit, where the power management could supply three voltages, including 16 V, 5 V, and 3.3 V. The weight of Device 2 control panel is 26.3 g.

### Operation of the olfaction interfaces

For operating Device 1, 6 digital and 2 ADC GP I/Os of a MCU (ATMEGA328P-MU, Microchip Technology Inc.) are utilized while Bluetooth module (WH-BLE103, Jinan USR IOT Technology Limited) embedded on the electronics wirelessly transmits serial data transferred from the MCU to the paired computer. The whole system is managed by one external 12 V battery (1800 mAh) and one 3.7 V battery (2000 mAh) by exploiting one LDO (LM2940LD5.0/NOPB, Texas Instruments Inc.) and one DC-to-DC voltage boost converter (LM2731XMFX/NOPB, Texas Instruments Inc.), where 12 V is stepped down to 3.3. 5, and 16 V for use, respectively. For each OG, the heating electrode operated by 16 V is installed with a thermistor with a reference voltage of 5 V for simultaneously heating up the paraffin wax and monitoring the temperature of heat released. The current flow of the heating electrode is controlled by a MOSFET (CSD17318Q2, Texas Instruments Inc.). Therefore, the current is able to flow to the heating electrode when the gate of the MOSFET is turned on by 5 V from a digital pin of the MCU. Furthermore, the output voltage from the thermistor is monitored through an ADC pin of the MCU continuously during the operation. The measured voltage from the analog pin is directly converted into temperature by the MCU and the data is transmitted wirelessly to a computer. Concurrently, a magnetic coil mounted on the heating electrode shifts its displacement toward and away from the heater by reversing the magnetic field generated according to the change of the sign of the voltage applied (3.3 V) resulting in an H-Bridge (BD6211F-E2, ROHM Semiconductor). When the heating electrode targets a lower temperature, the magnetic coil is touched with the nether magnet for cooling down rapidly by releasing heat out to the environment through thermal conduction while the H-Bridge module is controlled by two digital GP I/Os of the MCU. When the heating electrode is required to stabilize around a specific temperature, the power into the heater keeps switching on and off at a high frequency

(50 Hz) realized by the corresponding MOSFET according to the heating temperature monitored by the thermistor.

For operating the Device 2, 7 digital and 2 ADC GP I/Os of a MCU (ATMEGA328P-MU, Microchip Technology Inc.) are exploited while the data collected by the MCU is directly transmitted to a Bluetooth module (WH-BLE103, Jinan USR IOT Technology Limited) installed on the equipment by serial communication for wireless communication with a computer. To provide 3.3, 5, and 16 V to the system, an external 12 V battery (1800 mAh) and a 3.7 V battery (2000 mAh) are converted through one LDO (LM2940LD-5.0/NOPB, Texas Instruments Inc.) and one DC-to-DC voltage boost converter (LM2731XMFX/NOPB, Texas Instrument Inc.), as shown in Supplementary Fig. 30. Here, the power management system mass is 107.6 g. By adopting the PDMS and cloth as the two encapsulation layers (PDMS and cloth), the power system could be flexible and comfortable to users for long-term application. For operating each OG, 16 V is applied to the flexible heating electrode as a power source by controlling the current flow with a MOSFET (CSD17318Q2, Texas Instruments Inc.). By applying 5 V to the gate of MOSFET, the current is allowed to flow through the heating electrode. Furthermore, to cool down the heating electrode in a short period, a magnetic coil mounted on the heater shifts its displacement toward and away from the magnet by reversing the magnetic field generated according to the change of the sign of the voltage applied (3.3 V) resulting in an H-bridge (BD6211F-E2, ROHM Semiconductor). Therefore, when the heating electrode targets at a lower temperature, the magnetic coil is touched with the magnet for cooling down rapidly by releasing heat out to the environment through thermal conduction. As 1 and 2 digital GP I/Os are required respectively for controlling each MOSFET and H-bridge, at least 27 digital GP I/Os should be involved to manage 9 MOSFETS and 9 H-bridges at the same time. Therefore, to extend the number of GP I/O for the MCU, four 8 bits shift registers (74HC595, Texas Instruments Inc.) are connected to the MCU serially by exploiting 3 digital GP I/Os of the MCU. As a result, the whole 27 bits can be controlled independently for controlling the heating electrodes and magnetic coils at the same period. Finally, thermistors installed on each OGs continuously monitor the temperature of heat released according to the output voltage resulting from the thermistors with a reference voltage of 5 V during the operation. To achieve 9 ADC channels for temperature sensing, an 8-channel multiplexer (ADG708BRUZ-REEL7, Analog Devices Inc.) is connected to MCU with 4 digital pins while 1 ADC GPIO is directly connected to a thermistor. Each channel of the multiplexer is opened one by one according to the 4 bits received from the MCU while the measured voltage from an analog pin is directly converted into temperature by the MCU and the data is transmitted wirelessly to a computer.

## Fabrication of the motion caption electronics
Pattern and circuit were fabricated by printed circuit board processing techniques of copper (thickness 10 μm) plated with gold (thickness 50 nm). Furthermore, the cover layer was applied to both sides with exposing circle electrodes and square patches. Low temperature soldering paste (LF999, KELL YSHUN Technology Co. LTD) was used to bond and electrically connect all of the components, including the Bluetooth (WH-BLE103), the microcontroller (ATmega328p-mu), capacitors (14-22 pF), resistors (0.8 MΩ), crystal oscillator (16 MHz), and two accelerometers (BNO080), to corresponding contact pads on the Cu/PI substrate. After the integration of all components, PDMS (crosslink: PDMS = 1:30, 145 kPa, 2.4 mm thick) were poured onto the device, followed by curing at 110 °C for 5 min. Finally, embed the electronics in cloth fixed by sewing technics.

## Operation of the soft activity tacking system
For each activity tacking system targeting each human joint, each 3 Euler angles (roll, yaw, and pitch) of two accelerometer chips (BNO080, Hillcrest Laboratories, Inc.) are involved for VR application while one set of Euler angles from a single IMU chip is employed as a reference point for measuring an accurate and constant motion changes at a target joint at any 3D-position. For controlling the system and receiving accelerometers data from the chips, a programmable MCU (ATMEGA328P-MU, Microchip Technology Inc.) is exploited while the serial Transmit (TX) and Receive (RX) of the MCU is directly connected to the RX and TX of Bluetooth module (WH-BLE103, Jinan USR IOT Technology Limited) for supporting wireless communication between the electronics and a computer for proceeding VR application. The MCU of the motion capture system is managed by one external 5 V battery and through a LDO (TPS76933DBVR, Texas Instruments Inc.), 5 V is stepped down to 3.3 V for operating the accelerometer chips and the Bluetooth module. The accelerometer chips are directly connected to the MCU for data transmission through SPI sharing three different digital lines from the MCU (MOSI, MISO, and SCK) while each four digital GPIOs from the MCU are linked to control a single accelerometer chip (INT, RST, WAKE, and CS). Following, the Euler angles that are recorded in one decimal place are converted into 2-byte number groups for an efficient and rapid wireless transmission by reducing the total number of bytes required to be proceeded per each operational cycle. Thereafter, 6-byte data representing roll, pitch, and yaw are transmitted wirelessly to a computer via the Bluetooth module for VR application on the Unity platform.

## Mechanical simulation
The FEA commercial software ABAQUS (Analysis User's Manual 6.14) was used to calculate the bent, stretched, and twisted of structure. The PI, adhesive and PET layers were modeled by 0.54 million linear hexahedral elements of type C3D8R. Au layer were modeled by 0.02 million linear quadrilateral elements of type S4R. The minimal element size was 0.001 mm, which ensured the convergence and the accuracy of the simulation results. The elastic modulus ($E$) and Poisson's ratio ($v$) used in the analysis were $E_{PI} = 3.15$ GPa, $v_{PI} = 0.34$, $E_{Au} = 79$ GPa, $v_{Au} = 0.41$, $E_{Adhesive} = 1$ GPa, $v_{Adhesive} = 0.48$, $E_{PET} = 3.5$ GPa and $v_{PET} = 0.35$.

## Thermal simulation
The device was modeled in ABAQUS (Analysis User's Manual 6.14), where eight-node linear hexahedral heat-transfer element DC3D8 and four-node linear quadrilateral heat-transfer elements DS4 were adopted to discretize the geometry. The grid size was constrained in an appropriate range of 0.001 and 0.05 mm with the total number of grids being about 1.17 million, which ensured both acceptable convergence and computational efficiency. The thermal conductivity of Cu, Wax, PI, Au, Adhesive, PET, Magnet, and PDMS are 394 Wm$^{-1}$K$^{-1}$, 0.2 Wm$^{-1}$K$^{-1}$, 0.16 Wm$^{-1}$K$^{-1}$, 317 Wm$^{-1}$K$^{-1}$, 0.15 Wm$^{-1}$K$^{-1}$, 0.3 Wm$^{-1}$K$^{-1}$, 9 Wm$^{-1}$K$^{-1}$ and 0.15 Wm$^{-1}$K$^{-1}$. The natural air convection coefficient based on the ambient temperature of 21 °C is set as $h = 10$ Wm$^{-2}$K$^{-1}$.

## Characterization
The perfumes adopted in Supplementary Fig. 10d are minty (perfume 1) and jasmine incense (perfumer 2). The odorous liquids used in Fig. 4b are lavender (a) @ walking outsides(A), sweet orange (b) @ happiness (B), pineapple (c) having a lunch@ (C), green tea (d) @ teatime (D), lemon (e) @ summer (E), peach (f) @ sweety (F), strawberry (g) @ lovely (G), minty (h) @ teeth brushing (H), and lilac (i) @ pure (I). Here, we measured the odor retention time for each pure perfumes shown in Fig. 4b, ranging from 3.9 h to 141.8 h (Supplementary Fig. 31), and corresponding odor types and components have been summarized in Supplementary Table 2. Supplementary Fig. S32 shows the retention time of 30 different odor types adopted in Fig. 4c at a heating temperature of 200 °C. Among the 30 odor types, a low-volatile odor compound with boiling point >200 °C (mojito/wax, 30 mg) could continuously release odor for 91 mins. Supplementary Table 3 shows the physical status of the 30 odor types used in Fig. 4c at the heating

temperature of 240 °C, and it is obvious that some of odor types will not boil at 240 °C, including rice, vanilla, coffee milk, etc. It is concluded that our olfaction system could adopt low-volatile odor types as odorous additive into wax for providing olfaction display to users. By mixing the perfumes and wax at a heating temperature of 60 °C, 9 different odorous wax samples can be obtained, which can be still functional after exposing in air for three weeks (Supplementary Fig. 33). The odorous liquids used in Fig. 4c are ethanol (1), pineapple (2), grape (3), minty (4), rice (5), cream (6), gardenia (7), watermelon (8), vanilla (9), coffee milk (10), candy (11), coconut milk (12), coconut (13), milk (14), peach (15), pancake (16), orange (17), green tea (18), caramel (19), durian (20), lemon (21), strawberry (22), morning (23), ginger (24), clary sage (25), rosemary (26), lavender (27), clove (28), mojito (29), and cake (30). In the volunteer tests, 11 volunteers (4 males and 7 females) are recruited for volunteer experiments. Before the volunteer tests, all participants' informed consents have been obtained. In addition, each volunteer has been paid 200 HKD as the participant compensation for the volunteer tests.

## Reporting summary

Further information on research design is available in the Nature Portfolio Reporting Summary linked to this article.

## Data availability

The data that support the findings of this study are available from the corresponding authors upon request.

## Code availability

The data that support the findings of this study are available from the corresponding authors upon request.

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

## Acknowledgements

This work was supported by the National Natural Science Foundation of China (Grants No. 62122002), City University of Hong Kong (Grants No. 9667221, 9678274, and 9680322), and the Research Grants Council of the Hong Kong Special Administrative Region (Grants No. 21210820, 11213721 and 11215722), in part by InnoHK Project on Project 2.2—AI-based 3D ultrasound imaging algorithm at Hong Kong Centre for Cerebro-Cardiovascular Health Engineering (COCHE). YH.L. acknowledges the funding from The Natural Science Foundation of Zhejiang Province of China (No. LY21A020001), and Ningbo Scientific and Technological Innovation 2025 Major Project (No. 2021Z108). Q.G. acknowledges the funding from the Natural Science Foundation of Shandong Province in China (Grant No. ZR2021MF008), the State Key Laboratory of Functional Materials for Informatics (Grant No. SKL202101).

## Author contributions

YM.L and X.Y. conceived the ideas and designed the experiments. YM.L, C.Y., Z.Z., YH.L, Q.G., and X.Y. wrote the manuscript. YM.L, W.P., X.H., K.W., T.W., R.S., Y.Z., S.J., J.Z., Z.G., L.Z., K.Y., J.L, C.S., Y.G., G.Z., Y.H., and D.L. performed experiments and analyzed the experimental data. Z.Z., K.W., and YH.L. performed structural designs, and mechanical and thermal modeling. C.Y. and W.P. performed the circuit design and software programming.

## Competing interests

The authors declare no competing interests.
