## [Peer Review File · Nature Communications]

REVIEWER COMMENTS

Reviewer #1 (Remarks to the Author):

In this work, Liu et al. developed highly integrated millimeter-scale OGs arrays with high flexibility, stability, and durability. The proposed device shows outstanding performance in basic working parameters such as short response time (1.44 s), low-temperature fluctuation, low power consumption (100 mins operation with 80 mAh battery), long-term operation stability (13500 cycles), etc. In addition to that, the authors have successfully demonstrated the applications and importance of the proposed device in multiple fields, including 4D movies, healthcare, VR/AR, etc., In my opinion, the authors have done a complete and interesting work, and this manuscript can be recommended for publication in Nature Communications after clarifying the following comments/questions.

1. In Fig. 1, it can be noticed that the proposed OG includes a sensor between the Wax layer and the PI layer. But I didn't find a corresponding explanation about the functions and working principles of such a sensor. The authors should provide a discussion about that.
2. Sometimes the OGs need to be heated to 60 °C to generate the required odor information while placed inside the face mask and very close to the human face. Will it cause discomfort or potential scald to users?
3. The authors are suggested to include more detailed information regarding Fig. 4. For example, for the message delivery, although it shows that longer training time can lead to an improvement in recognition rate, how did the authors design and define the training time? Does that mean letting the volunteers smell each odor for 60 mins? Besides, did the author give a fixed testing time for the users? In addition, the data shown in Fig. 4c indicates that odor can only efficiently give joyful and repulsive emotions to users. Is this because different users will have different feedback regarding the same smell? Is it possible to design different emotion systems for different users?
4. What are the advantages of applying OGs arrays compared to a single OG element? If the principle of the OG is to generate different smell information by tuning the ethanol gas concentration as depicted in extended data fig. 3, then one OG element should be enough.
5. More previous works related to the wearable devices for human-machine interfacing systems in VR technologies should be included to enrich the background comprehensiveness, such as: [<https://doi.org/10.1038/s41467-022-32745-8>], [<https://doi.org/10.1021/acsnano.2c04043>]. [<https://doi.org/10.1126/sciadv.aaz8693>]

Reviewer #2 (Remarks to the Author):

The manuscript entitled “Olfactory Virtual Reality Enabled by Wireless Skin-Interfaced Electronics” is not acceptable in the current form. However, it might have somewhat novelty. The manuscript is reconsidered if it is thoroughly revised.

1. The reviewer understood the authors have succeeded in making miniaturized odor generator using flexible material. They should mainly focus on miniaturized device since the application part seems insufficient.

Thus, the authors should reconsider the title.

Skin-interfaced small device to generate smells?

Olfactory virtual reality is too strong at the current stage

The application part does not have gas concentration measurement and most of them do not have the sensory test result. Thus, the sufficient data are not presented. The application part should be drastically compressed and the organization of the paper should be changed.

2. It seems that face shape changes from person to person. Is it possible to fit your face mask to everyone even if the flexible materials are used? Is there any possibility that odor generator touches the skin around nose? Is it ok if the smell leaks from the face mask?

3. Fig.1

The authors use PDMS. Since smell seems to be absorbed by PDMS, the odor generator might not be able to be used many times because of remaining smell. Moreover, is there any smell adsorption at the face mask?

Although ethanol is easily evaporated, low-volatile compounds might stick to housing for long time. The author should mention the influence of remaining smell.

4. Fig.1

The proposed devices do not seem to have the capability of blending smells. If so, the number of smells to be presented is limited. Moreover, it is necessary to change the cartridge if a user changes the smell. The author should mention those points.

5. Where is the battery placed. Since it seems the power of a few hundreds mW is consumed for actuators and heater, In Extended data Fig.2, 12V and 3.7 battery are used. Their weights might be the problem.

6. The temperature of wax is about 60 degree Celsius. Is it enough to evaporate low-volatile compounds since those boiling points are typically around 200 degree Celsius? Moreover, is there any possibility that a user feels heat?

7. Is there any possibility that the smell is accumulated in space between the face mask and face even if the temperature changes rapidly? The authors should show the data of gas sensor or sensory test result.

8. The reviewer understood that the temperature of the olfactory generator is precisely and rapidly controlled. However, The gas sensor response highly fluctuated in Extended data Fig.3 I. What is the reason for the fluctuation? Is there rapid recovery when the gas concentration decreases? What happens if low-volatile compound instead of ethanol is used? Although the authors show the temperature change in several situations, they should show the gas sensor response, not the temporal temperature change.

9. Is it necessary to control the temperature precisely? If the authors would like to control the concentration, only Extended data Fig.3 I is not enough.

10. Fig.4 aiV

The authors showed the temperature change. However, the concentration change should be shown here. Otherwise the sensory test should be done.

It is not clear whether a participant can recognize the change of smell intensity because there is no result of sensory test.

The authors mentioned the temporal sequence of the temperature at the top of page 11. How did they determine it? What is the relationship between temperature and concentration?

11. Fig.4b

The authors should mention that types of smells appear in Material and Method.

12. Fig.4 cii

Did the authors use nine depressed volunteers? If so, how did the authors control the mental situation before sniffing the smell?

The author mentioned the sweat orange was useful. Do they mean the sweet orange?

13. Fig.4e

The relationship between Fig.4ei and Fig. 4eii is not clear.

In the text, the author mentioned they proved the capability of the olfaction system in coordinating each OG working independently.

However, it seems temperatures of all channels changed synchronously.

14. Fig.5 d

Just the temperature of the device is not enough. Gas sensor response or sensory test result should be shown.

15. Table S1 seems insufficient. OG part includes only a few reports and does not cover most of related works. On the other hand, the survey of odor sensor is not necessary.

Reviewer #3 (Remarks to the Author):

This paper presents skin-interfaced olfactory feedback systems with wirelessly programmable capabilities based on arrays of flexible and miniaturized odor generators (OGs) for olfactory VR applications.

The authors insist on significance in that the olfactory display proposed here can establish a bridge between the electronics and users for broad application fields ranging from immersive VR experiences because they are wireless and wearable format, showing two types of devices: skin-mountable form and glasses form. However, while they mentioned only two existing related papers where olfaction display systems mainly associate with a large instrument to generate smell in a closed area or an in-built bulky VR set, they expressed it as if this were the only paper to solve this problem.

According to the prior study, wireless and portable olfactory display (OD) system has already been reported¹. Furthermore, the development of wireless OD masks for the VR experience is not new^{2, 3, 4}. And the miniaturized and skin-mountable form factors of the OD were explored⁵. However, this manuscript does not include and mention these prior articles that are directly related to the presented works. Also, the odor generator employing the heater and paraffin wax was reported earlier⁶. Thus, this odor generation system in this paper is not novel.

Considering all points mentioned above, the novelty and significance of this paper are very limited, making this work unacceptable to be published in Nature communication.

Similar previous works that were not cited in this paper:

1. Amores J, Dotan M, Maes P. Development and Study of Ezzence: A Modular Scent Wearable to Improve Wellbeing in Home Sleep Environments. *Frontiers in psychology*, 550 (2022).
2. de Paiva Guimarães M, Martins JM, Dias DRC, Guimarães RdFR, Gnecco BB. An olfactory display for virtual reality glasses. *Multimedia Systems*, 1-11 (2022).
3. Yang P, et al. Self-powered virtual olfactory generation system based on bionic fibrous membrane and electrostatic field accelerated evaporation. *EcoMat*, e12298 (2022).
4. Kato S, Nakamoto T. Wearable olfactory display with less residual odor. In: 2019 IEEE International Symposium on Olfaction and Electronic Nose (ISOEN). IEEE (2019).
5. Wang Y, Amores J, Maes P. On-face olfactory interfaces. In: *Proceedings of the 2020 CHI Conference on Human Factors in Computing Systems* (2020).
6. Tiele A, Menon S, Covington JA. Development of a Thermal-Based Olfactory Display for Aroma Sensory Training. *IEEE Sensors Journal* 20, 631-636 (2019).

There are additional comments that need to be addressed:

1) For clarification, this manuscript should add and compare the device's performance in the previous articles mentioned above in table S1. It could be better to search with the keyword 'olfactory display.'

2) The author has emphasized the form factor of OD as the skin-interfaced system, although only two OG modules can be equipped to the format. Therefore, it is hard to understand the description below.

As a result, to provide personalized odors in a small, localized area, new wearable olfaction interfacing technologies should exhibit advances as following: (a) the whole system should be built up on a soft substrate in a wearable or even skin-integrated format with miniaturized size and light weight; (b) as many odors with adjustable concentrations and long operation duration to support long term utilization without frequent replacement/maintenance.

Could you explain why the OD should be directly mounted onto human skin in detail?

3) Also, fabricated device 1 (skin-mounted device) in Figure 1D does not seem flexible enough to be conformally attached to curved human skin for a long time. Therefore, quantitative data support is needed to show the high flexibility of the device.

4) When the OD generates odors, the temperature of the heater could be over 50 °C. Could you ensure it does not cause damage to the skin where the device is mounted under continuous heating conditions?

5) This OG hires paraffin wax containing the perfume. The odor resource is in an open system, not a closed reservoir. Therefore, it is required to quantitatively verify the slowly evaporated perfume under room temperature for an extended period.

6) Could the volunteers who put the device below their nose feel any perfume odor when it is not working? You described the human odor threshold as 80 ppm. So, if the perfume gas concentration value is shown at room temperature, the reader can check it.

7) To help understand the reader, can you express the temperature change as time-dependent data in Extended Data Figure 3I?

8) Also, as seen in the Extended Data Figure 3I, the concentration of emitted ethanol gas seems unstable despite keeping the same temperature. Can you explain the instability of measured gas concentration? The author has exhibited stable and reliable temperature data that is not the output data the user wants to know (perfume gas concentration) in Figures 4a, e, and 5d. Therefore, it should be clearly explained to prove the reliability and fidelity of this OG system employing the heater and paraffin.

Responses to comments of Referee #1

Comments from Referee #1:

Summary Comment: In this work, Liu et al. developed highly integrated millimeter-scale OGs arrays with high flexibility, stability, and durability. The proposed device shows outstanding performance in basic working parameters such as short response time (1.44 s), low-temperature fluctuation, low power consumption (100 mins operation with 80 mAh battery), long-term operation stability (13500 cycles), etc. In addition to that, the authors have successfully demonstrated the applications and importance of the proposed device in multiple fields, including 4D movies, healthcare, VR/AR, etc., In my opinion, the authors have done a complete and interesting work, and this manuscript can be recommended for publication in Nature Communications after clarifying the following comments/questions.

Our response: We thank the referee for these positive comments. We carefully addressed the issues, as listed below, and we revised our manuscript accordingly.

Modifications: None

Comment 1: In Fig. 1, it can be noticed that the proposed OG includes a sensor between the Wax layer and the PI layer. But I didn't find a corresponding explanation about the functions and working principles of such a sensor. The authors should provide a discussion about that.

Our response: We thank the referee for this useful comment. The sensor shown in Fig. 1a is a commercial thermistor with a size of 1 mm × 0.5 mm × 0.5 mm for temperature sensing. Based on a ceramic semiconductor, the thermistor could accurately detect the ambient temperature, showing a negative temperature coefficient of resistance. Due to the tiny size and high accuracy (± 0.1 °C), the thermistor is adopted to serve as a temperature sensor for continuously monitoring the heating temperature of OGs, as shown in Fig. 1a and Supplementary Fig. 1. To stabilize the heating temperature of OGs around a specific temperature, the heating power into OGs keeps switching on and off in a high frequency (~ 50 Hz) according to the real-time heating temperature by the thermistor. As a result, due to the excellent electrical performance of the thermistor, the heating temperature of OGs could be maintained at any temperature with a minor temperature fluctuation (± 0.25 °C) between 45 °C and 60 °C, as shown in Fig. 2 and Supplementary Fig. 8. We have modified the content and figures for a clear illustration on Page 20 Line 14.

Modifications: On Page 20 Line 14, the text has been modified as “Fix a thermistor (QN0402X104F4250FB, Advanced Materials Electronics Co., Ltd) onto the reserved patches in the middle of the heating electrode by silver paste, then a copper coil is directly attached onto the heating electrode with the interface sealed by instant glue. Here, the thermistor (1 mm × 0.5 mm × 0.5 mm) is a ceramic semiconductor based on transition-metal oxides, showing a negative temperature coefficient of resistance. Due to the miniaturized size, high temperature sensitivity (Figs. 2a, b), and excellent accuracy (± 0.1 °C) (38), the thermistor is adopted as the temperature sensor for real time monitoring the heating temperature of the OGs.”

On Page 21 Line 22, the text has been modified as “The current flow of the heating

electrode is controlled by a MOSFET (CSD17318Q2, Texas Instruments Inc.). Therefore, the current is able to flow to the heating electrode when the gate of the MOSFET is turned on by 5 V from a digital pin of the MCU. Furthermore, the output voltage from the thermistor is monitored through an ADC pin of the MCU continuously during the operation. The measured voltage from the analog pin is directly converted into temperature by the MCU and the data is transmitted wirelessly to a computer. Concurrently, a magnetic coil mounted on the heating electrode shifts its displacement toward and away from the heater by reversing the magnetic field generated according to the change of the sign of the voltage applied (3.3 V) resulting in an H-Bridge (BD6211F-E2, ROHM Semiconductor). When the heating electrode targets at a lower temperature, the magnetic coil is touched with the nether magnet for cooling down rapidly by releasing heat out to the environment through thermal conduction while the H-Bridge module is controlled by two digital GP I/Os of the MCU. When the heating electrode is required to stabilize around a specific temperature, the power into the heater keeps switching on and off in a high frequency (50 Hz) realized by the corresponding MOSFET according to the heating temperature monitored by the thermistor.”

Fig. 1. Architectures of the two olfactory interfaces. **a** Schematic diagram of the skin-integrated, wireless olfaction interface for providing olfaction feedback to a user for immersive VR experience. The schematic illustration in the frame shows an exploded view of the OG. **b**, **c** Exploded-view illustrations of the two olfaction interfaces, where Device 1 is in a skin-integrated format based on two OGs (b), and Device 2 is built on a flexible face mask platform with 9 OGs embedded (c). **d** Optical images of the skin-integrated olfaction interface (Device 1) mounted onto human curved skins under the bending mechanical deformation. **e** Optical images of the face mask based olfaction interface (Device 2) with 9 OGs, capable of yielding 9 different original odor, where the details of circuit design, device layout, and assembled face mask have been displayed.

Fig. S1. Optical images of the odor generator under two mechanical conditions, including uplift for increasing or stabilizing temperature, and dropping down for fast decreasing temperature.

Comment 2: Sometimes the OGs need to be heated to 60 °C to generate the required odor information while placed inside the face mask and very close to the human face. Will it cause discomfort or potential scald to users?

Our response: We thank the referee for this good comment, which is very important to evaluate the practical applications. To investigate the safety issue of face mask device, we conducted a series of new works in revision, and the new results are shown in Supplementary Fig. 18. Firstly, we took thermal images by an IR camera of a working OG from two sides, and it can be found that the air temperature of the area 1.5 mm above the working OG is nearly room temperature, as shown in Supplementary Figs. 18a, b, c. So, if the working distance between the user skin and the thermal working area of the device greater than 1.5 mm for sure is safe to users and without causing any discomfort feeling. Then, we conducted user studies by calling 11 volunteers (4 males and 7 females) to join the test. We measured the distance between their nose and the working OGs when they were wearing the mask, as shown in Supplementary Fig. 18d. It is found that all volunteers' distance values are larger than 1.5 mm, which demonstrates the safe application of the mask device.

Meanwhile, we also studied the safety issue for the skin-integrated patches (Device 1), we measured the distance between the user's nose and the OG integrated in the Device 1, as shown in Supplementary Fig. 18g. The distance value (23 mm) is much larger than the 1.5 mm safety distance (Supplementary, Fig. 18b), demonstrating that the working OGs in Device 1 are safe to users. Then, we measured the bottom-side thermal distribution of Device 1 with the peak temperature value of 32.2 °C after operating for 10 mins, illustrating that the Device 1 will not induce any discomfort to users when they are wearing the Device 1. As a result, the Device 1 has no safety issue to users. We put comments on this point on Page 14.

Modifications: On Page 14 Line 6, the text has been modified as “**Supplementary Fig. 18** shows the safety testing of the olfactory devices as the heating temperatures of the OGs are stabilized around 60 °C. The thermal images of a working OG illustrate a fact that the ambient air temperature with distance of 1.5 mm from the working OG is around at room temperature (**Supplementary Figs. 18b, c**). Then, we measured the distance between human nose to working OGs from 11 volunteers (**Supplementary Fig. 18d**). It is clear that all volunteers (including 4 males and 7 females) could safely wear the Device 2 with all the distance values greater than 1.5 mm (**Supplementary Figs. 18e, f**). For the Device 1, the open design allows

the OGs to exhibit good thermal convection and also to maintain a safe distance to users with a big gap (23 mm) between OGs and user's nose, since the temperature at the skin interface is only 32.2 °C, as shown in **Supplementary Figs. 18g, h.**"

Fig. S18. Safety operation of Device 2 and Device 1. (a) A OG 3D model built up in software ABAQUS (Analysis User's Manual 6.14). (b, c) Two views of thermal distribution of a working OG with the working temperature stabilized around 60 °C, where the working temperature is the one around the thermistor for melting odorous wax insides OGs. (d) Schematic diagram of the inside layout of Device 2 when users are wearing the Device 2. Here, the distance between users' nose and working OGs decides the safety of Device 2 during operation. (e, f) 11 volunteers' distance values between volunteers' nose and working OGs, including 4 males and 7 females. All volunteers can safely wear the Device 2 during its operation as the minimum distance value is 2 mm, where the air temperature is room temperature. (g) Optical image of a user wearing the Device 1 on his upper lip with a value (23 mm) showing the distance between the OG and the nearby human nose. (h) Thermal distribution of the Device 1 bottom side with the working temperatures of two OGs stabilized around 60 °C. The temperature peak is 32.2 °C, demonstrating that the Device 1 has no potential risk to the attached human skin.

Comment 3: The authors are suggested to include more detailed information regarding Fig. 4. For example, for the message delivery, although it shows that longer training time can lead to an improvement in recognition rate, how did the authors design and define the training time? Does that mean letting the volunteers smell each odor for 60 mins? Besides, did the author give a fixed testing time for the users? In addition, the data shown in Fig. 4c indicates that odor can

only efficiently give joyful and repulsive emotions to users. Is this because different users will have different feedback regarding the same smell? Is it possible to design different emotion systems for different users?

Our response: We thank the referee for the useful comment. We have enriched Fig. 4, with more detailed information and subfigures. Please find the response to each sub-question as following:

- (1) For the first question “how did the authors design and define the training time”, the training time in Fig. 4b is defined as the duration of the volunteers in continuously smelling the required odors generated by Device 2. During the volunteer tests, all volunteers wear the Device 2, and they could select the generated odor type sequence with the corresponding smell messages recorded in a paper for their reference. There are two training modes of short-term training (10 min) and long-term training (1 hr). The training time for each odor types is same, where the long-term training needs 400 s for each odor as the total training time is set at 1 h, and the short-term training needs 67 s for each odor with a total time of 10 min. The updated data is summarized in Fig. 4b. We have modified the text on Page 15 Line 15.
- (2) For the second question “Does that mean letting the volunteers smell each odor for 60 mins?”. We have clarified the training time in the revised manuscript, and as we answered in sub-question 1, each odor need to be smelled for 67 s and 400 s by the volunteers. We have modified the text on Page 15 Line 15.
- (3) For the third question “Besides, did the author give a fixed testing time for the users?”, there is no fixed testing time for all the volunteers after they were doing the recognition test for the smell message delivery. All volunteers were given one chance to guess the smell message as the 9 different odor types are randomly generated by the Device 2. We have put comments on this point on Page 15 Line 15.
- (4) For the fourth question “the data shown in Fig. 4c indicates that odor can only efficiently give joyful and repulsive emotions to users. Is this because different users will have different feedback regarding the same smell?”, we conducted new experimental tests associating with user study to investigate the human emotion reaction to different odor types generated by Device 2, as shown in the modified Fig. 4c. Before the test, 11 volunteers’ emotions were evaluated by filling in a standard human emotion assessment for, Hospital Anxiety and Depression Scale (HADS), and the results have shown that all the volunteers are in a normal emotion range. During the test, all volunteers are required to smell 30 different odor types generated by the Device 2 (see the odor type details in Characterization), and write down their emotion reaction to each odors (Figs. 4civ, v). Here, all volunteers could decide the odor generation sequence and smelling time, and they are required to repeat the test once per day for five times, meaning that we obtained the 55 filled human emotion reaction forms at last. It is found that the clove odor could induce the volunteers to be repulsive with the possibility reaching up to 0.82, gardenia odor could raise up peaceful emotions with the possibility of 0.51, and peach odor could trigger a joyful emotion with the possibility of 0.65, as shown in Fig. 4civ. Therefore, our device could efficiently give repulsive, peaceful, and joyful emotions to users on average. It is also found that different volunteers may have distinguished emotion reaction to same odors. For example, for the lemon odor, the volunteers’ emotions contain all types, including repulsive, dulcet, peaceful, joyful, exciting, confusing, unnerving, apathic, and curious, as shown in Fig. 4civ. The different emotion reaction to same odor types among users may be due to the different users’ past memories to same odors (*Chemical senses* 35, 3-20, 2010), which demonstrates that it is difficult to efficiently give various desired emotion to different users. We have modified

the text on Page 16 Line 11.

- (5) For the fifth question “Is it possible to design different emotion systems for different users?”, we would like to thank the reviewer for this great question. In revision, we conducted a volunteer test as illustrated above in Fig. 4c. All volunteers are required to repeat the test for five times in five days. Then, we analyzed their emotion reaction variation of the five times for each volunteers with the results shown in Figs. 4cv, iv. It is obvious that the emotion reactions to some odor types for each volunteers keeps constant. For example, one volunteer will always become exciting as she smells ethanol, but another one will become repulsive when smelling the ethanol, as shown in Fig. 4v. Therefore, it is possible to customize a specific emotion system for each user. We have put comments on this point on Page 16 Line 11.

Modifications: On Page 15 Line 15, we have modified the text as “A second representative application of the olfaction interfaces is the message delivery through smell for a visual/auditory impaired person (**Fig. 4b**). By training visual/auditory impaired users to link some odors to specific events, such as lavender odor@ walking outdoors, pineapple odor@ having a lunch, and green tea odor@ teatime, the users could simultaneously react to the released odor from the face mask-based olfaction interface with 9 different odor options. During the training time, all the volunteers are required to continuously smell the odors generated by Device 2. The training associates with short-term (67 s for each odor, total of 10 min) and long-term (400 s for each odor, total of 60 min). When taking the test, the volunteers only have one chance to give their response to the generated odors with no fixed testing time.”

Page 16 Line 11 as “**Fig. 4c** shows another application scenario where some odors generated from the olfactory interface system could be used for smoothing users’ emotion. As odors could arouse human emotion by leading to the recall of emotional memories, the olfactory interface could be adopted for smoothing users’ depressed mood from the stress (23) (**Fig. 4ci**). To verify the effect of odors on human emotion, 11 volunteers’ reactions to 30 different odors (see details in Characterization) from the face mask based olfactory interface were collected as shown in **Fig. 4c** (all volunteers were in normal emotion before the testing, **Figs. 4cii and iii**), and the result demonstrates that the possibility of being joyful for these volunteers could reach up to 56%, 65%, 44%, and 56% when they smelled the grape, peach, orange, and strawberry odor, respectively (**Figs. 4civ, v**). It is also possible to give other emotions to users as they smelled clove and gardenia odors, inducing repulsive and peaceful emotions with the corresponding possibility of 0.82 and 0.51, as shown in **Fig. 4civ**. In addition, it is obvious that different users have distinguished emotion reactions to the same odors (**Figs. 4cv, vi**), which may be due to the volunteers’ different emotional memories for same odor type. The large emotion reaction difference for each user may increase the difficulty in giving various desired emotions to users. **Fig. 4cv** shows the volunteers’ emotion reaction to 30 odor types, where the results were collected from five times test for each volunteer. It is found that for each volunteer, their emotion reactions to some odor types keep constant, demonstrating that we could customize the odor types in the Device 2 for different users.”

Fig. 4. Demonstrations of the olfactory interfaces. **a** A demonstration of the skin-integrated Device 1 in displaying olfaction feedback for providing an immersive experience to users during movie watching. Here, a girl in the movie is smelling a rose with the decreasing distance between the flower and her while the watcher could smell a more and more intense rose odor by wearing the olfaction interface. Here, OCL demonstrates the odor concentration level ranging from 1 to 3, corresponding to the low, middle, and high levels (iv). **b** A demonstration of the face-mask based olfaction interface in providing an alternative communication method, smell messages, for the disabled without the capability of vision and audio. By training users to link odors to specific information, the disabled users could efficiently communicate with others. **c** A demonstration of the Device 2 in assisting users to control their emotions as some specific odors could recall different human emotions. Here, all volunteers involved in the

emotion reaction test are verified in normal emotion range (**cii and ciii**). **d** A potential demonstration of the Device 2 for helping amnesic patients recall lost memories as odors perception is modulated by experience, leading to the recall of emotional memories. **e** Demonstration of the wireless olfaction interfaces in real time interaction between the user and a virtual subject for walking in a virtual garden surrounded by various fruit fragrance. By programming the Device 2, 9 OGs inside the device could work independently (**eii**), or work together (**eiii**).

Comment 4: What are the advantages of applying OGs arrays compared to a single OG element? If the principle of the OG is to generate different smell information by tuning the ethanol gas concentration as depicted in extended data fig. 3, then one OG element should be enough.

Our response: We thank the referee for this comment. The first advantage of adopting OG arrays rather than single OG, is that users could get more smell information, such as get much more comprehensive messages as one odor type corresponding to one message. The second advantage of adopting the OG arrays is to blend several odors together to deliver a combined odor information, which is very important in the application of immersive VR. The OGs developed in this project could generate different odor concentrations by controlling the heating temperature. Human could only sense the large range of odor concentration variation with a rough division level, such as none, low, middle, and high concentrations, which would greatly limit delivered smell messages number if we only adopt a single OG. So, we believe these facts can greatly support the importance of developing OG arrays. To future prove the advantages of the array compare to the single unit based on the consideration of human olfactory ambiguous sensitivity range, we performed user study and the results are shown in Supplementary Fig. 20, where results show that there is no standard odor concentration division method for different users in delivering same smell messages. The recognition rate results shown in Fig. S20 associate with the study for 11 volunteers, who were asked if they could sense OGs with 9 different odors embedded at room temperature when they are wearing the Device 1 on the upper lips. It is obvious that each volunteer has distinguished recognition rates, proving that each user with different odor concentration thresholds for same odor types. Compared to the odor concentration, human being could easily distinguish different odor types, which could increase smell message recognition rate. We put comments on this point on Page 16 Line 5.

Modifications: On Page 16 Line 5, the text has been modified as “In this demonstration, we developed the smell message delivery system based on the OGs array instead of single OG due to the facts: (1) human being could easily distinguish different odor types, where one odor type corresponds to one message. (2) the delivered messages quantity could be significantly enhanced by blending several odors together, which could easily be realized by our Device 2, as shown in **Fig. 4eiii**. Although the single OG could also deliver smell messages by generating different odor concentration for users, it is difficult to build up a standard odor concentration division for different users who has distinguished odor concentration thresholds, as shown in **Supplementary Fig. 20.**”

Fig. 4. Demonstrations of the olfactory interfaces. **a** A demonstration of the skin-integrated Device 1 in displaying olfaction feedback for providing an immersive experience to users during movie watching. Here, a girl in the movie is smelling a rose with the decreasing distance between the flower and her while the watcher could smell a more and more intense rose odor by wearing the olfaction interface. Here, OCL demonstrates the odor concentration level ranging from 1 to 3, corresponding to the low, middle, and high levels (iv). **b** A demonstration of the face-mask based olfaction interface in providing an alternative communication method, smell messages, for the disabled without the capability of vision and audio. By training users to link odors to specific information, the disabled users could efficiently communicate with others. **c** A demonstration of the Device 2 in assisting users to control their emotions as some specific odors could recall different human emotions. Here, all volunteers involved in the

emotion reaction test are verified in normal emotion range (**cii and ciii**). **d** A potential demonstration of the Device 2 for helping amnesic patients recall lost memories as odors perception is modulated by experience, leading to the recall of emotional memories. **e** Demonstration of the wireless olfaction interfaces in real time interaction between the user and a virtual subject for walking in a virtual garden surrounded by various fruit fragrance. By programming the Device 2, 9 OGs insides the device could work independently (**eii**), or work together (**eiii**)

Fig. S20. A volunteer testing showing the recognition rate of volunteers in smelling the OGs at room temperature when they are the Device 1. Here, a, b, c, d, e, f, g, h, and i stand for lavender, orange, pineapple, green tea, lemon, peach, strawberry, minty, and lilac.

Comment 5: More previous works related to the wearable devices for human-machine interfacing systems in VR technologies should be included to enrich the background comprehensiveness, such as: [<https://doi.org/10.1038/s41467-022-32745-8>], [<https://doi.org/10.1021/acsnano.2c04043>]. [<https://doi.org/10.1126/sciadv.aaz8693>]

Our response: We thank the referee for this useful comment. We have added these references in the text.

Modifications: On Page 3 Line 1, the text has been modified as “Recent human machine interfaces highlight the importance of human sensation feedback, including vision, audio, and haptics, associating with wide applications in entertainment, medical treatment, and VR/AR (1-3). Olfaction plays a significant role in human perceptual experiences, which is equally important to visual and auditory feedbacks (4-7). As one of the typical five senses, olfaction has shown a crucial influence in shaping human lives, as most aspects in daily life associate with olfaction coming from manmade materials, industry, transport, household products, etc. (8,9).”

The newly added references:

1. Z. Sun, M. Zhu, X. Shan, C. Lee, *Augmented tactile-perception and haptic-feedback rings as human-machine interfaces aiming for immersive interactions. Nature communications* 13, 1-13 (2022).
2. M. Zhu, Z. Sun, C. Lee, *Soft modular glove with multimodal sensing and augmented haptic feedback enabled by materials' multifunctionalities. ACS nano* 16, 14097-14110 (2022).
3. M. Zhu et al., *Haptic-feedback smart glove as a creative human-machine interface (HMI) for virtual/augmented reality applications. Science Advances* 6, eaaz8693 (2020).

Responses to comments of Referee #2

Comments from Referee #2:

Summary Comment: The manuscript entitled “Olfactory Virtual Reality Enabled by Wireless Skin-Interfaced Electronics” is not acceptable in the current form. However, it might have somewhat novelty. The manuscript is reconsidered if it is thoroughly revised.

Our response: We thank the referee for these positive comments. We carefully addressed the issues, as listed below, and we revised our manuscript accordingly.

Modifications: None

Comment 1: The reviewer understood the authors have succeeded in making miniaturized odor generator using flexible material. They should mainly focus on miniaturized device since the application part seems insufficient. Thus, the authors should reconsider the title. Skin-interfaced small device to generate smells? Olfactory virtual reality is too strong at the current stage

Our response: We thank the referee for this comment. We have modified the title as “Soft, Miniaturized, Wireless Olfactory Interface for Virtual Reality”.

Modifications: The title of the manuscript is modified as “Soft, Miniaturized, Wireless Olfactory Interface for Virtual Reality”

Comment 2: The application part does not have gas concentration measurement and most of them do not have the sensory test result. Thus, the sufficient data are not presented. The application part should be drastically compressed and the organization of the paper should be changed.

Our response: We thank the referee for this comment. In revision, we have conducted a series of experiments to systemically test and characterize the odor concentration measurement of the olfactory interface. The newly added results include the gas concentration measurement, a series of volunteer sensory studies, and many other essential sensing characterizations, and the new results are shown in Fig. 4, Extended Data Fig. 4, and Supplementary Figs. 11, 13, 19, 20.

For ethanol concentration measurement in Extended Data Fig. 4, we investigated three potential effects on ethanol generation performance of OGs with paraffin/ethanol embedded, including the heating temperature of OGs (Extended Data Fig. 4d), ambient wind speed (Extended Data Fig. 4e), and distance between working OGs and the ethanol sensor (Extended Data Fig. 4f), where all the tests were performed in an open box for minimizing the unpredictable ambient wind around working OGs. It is found that increasing heating temperature of OGs could contribute higher ethanol concentration, higher ambient wind could induce faster ethanol recovery time (RT), and larger distance between OGs and the ethanol sensor will take a longer time to detect the gas, meaning a longer delay time. We also measured the remaining time of ethanol in Device after the Device 2 releasing ethanol for 5 mins, as

shown in Extended Data Figs. 4g, h, i, j. Here, we setup two test conditions: Condition one is that the experimenter starts wearing the Device 2 after it releasing ethanol for 5 mins. Condition two is that the Device 2 keeps static in the open box after releasing ethanol for 5 mins. It is obvious that Condition one takes a shorter recovery time than Condition two, which is induced by the human breathing. Extended Data Fig. 4 shows the electrical response of the ethanol sensor embedded in the Device 2 as the experimenter is wearing it, where the heating temperature of the working OG in Device 2 is target at 60 °C. It is found that human breathing could decrease the ethanol concentration around the sensor with obvious concentration fluctuation (± 180 ppm). We have modified the text on Page 10 Line 22.

For the volunteer tests, in revision, we also conducted a series of volunteer sensory tests, as shown in Fig. 4, Supplementary Figs. 11, 13, 19, 20. The reason why we conducted these volunteer sensory tests is that we couldn't find specific odor sensors currently to measure perfumes-based odor concentrations generated by the OGs. All the volunteer tests involve 11 volunteers with 4 males and 7 females. The volunteer test shown in Supplementary Fig. 11 is to investigate if volunteers could obviously sense the enhanced odor concentrations for 9 different odor types (see details in Characterization) generated by Device 2 when the heating temperature of OGs are increased from 45 °C to 50 °C, 55 °C, and 60 °C step by step. The high average recognition rate, 0.93, demonstrates that all volunteers confirm to sense the odor concentration variations by Device 2. We also measured the odor remaining time in Device 2 after the device release odors for 5 mins, where we setup two testing conditions. Condition One is that volunteers continuously smell the odors when wearing the Device 2. Condition Two is that the volunteers come to smell the odors of the face mask every 10 s. Then, we recorded the time for 9 different odors in the two testing conditions when the volunteers couldn't sense the corresponding odors, with the result shown in Supplementary Fig. 13. Supplementary Fig. 19 is another one volunteer sensory test, focusing on recording the human reaction to the fast odor concentration variation realized by the heating temperature variation of OGs in Device 1. During the test, the Device 1 will be programmed to increase temperatures of OGs from 45 °C to 50 °C at 5 s, from 50 °C to 55 °C at 15 s, from 55 °C to 60 °C at 25 s, then lasting 1 min until shutting down Device 1. Once the volunteers can sense the enhanced odor concentration, the time point will be recorded meanwhile. Two different odor types for two OGs in Device 1 are adopted, including lavender and lilac. Supplementary Fig. 20 shows a new volunteer sensory test for testing if volunteers could sense the odors by OGs at room temperature, and it is found that each volunteers have distinguished odor concentration thresholds to various odor types with the recognition rate ranging from 0.22 to 0.89. As a result, volunteers can obviously sense some specific odor types, such as lavender, strawberry, and lilac, but the possibility in sensing minty, orange, green tea, lemon, and peach is low, ranging from 0.27 to 0.09. Finally, we collected all volunteers' emotion reactions to 30 different odors generated by Device 2 (see details in Characterization), as shown in Fig. 4c. It is found that the possibility of being joyful for these volunteers could reach up to 56%, 65%, 44%, and 56% when they smelled the grape, peach, orange, and strawberry odor, respectively. In addition, it is obvious that different users may have distinguished emotion reactions to same odor types (Figs. 4cv, vi), which means that users could select their preferred odor types integrated insides Device 2 for emotion smoothing. We have modified the text on Page 10 Line 22.

For reorganization of application part, we reorganize the applications data in Fig. 4. We reperfomed a volunteer test to investigate the human emotion reaction to 30 different odor types with the details mentioned above (Fig. 4c). In addition, we also demonstrate the capacity

of the Device 2 in controlling the 9 integrated OGs to work independently, as shown in Fig. 4e. It can prove a fact that the Device 2 could operate each OG independently, and it is accessible to operate several OGs together for blending odors, which is required in VR/AR applications. The layout of Fig. 4 has been adjusted to enrich the 5 typical applications details. We have modified the text on Page 14 Line 22.

Modifications: On Page 10 Line 22 as “**Extended Data Fig. 4** shows the gaseous ethanol generation performance of the OGs. To minimize the unpredictable ambient wind effect on the performance of the OG, the whole experimental setup is placed inside an open box, where a commercial ethanol sensor is fixed 1 cm above a working OG, continuously monitoring the nearby ethanol concentration by reading the voltage variation of a 5 k Ω resistor connected in series (**Extended Data Figs. 4a, b, c**). By switching the heating temperature of the OG from 45 °C to 50 °C, to 55 °C, and finally to 60 °C (**Supplementary Fig. 10**), the ambient ethanol concentrations could reach up to 531 ppm, 2821 ppm, and 4531 ppm, respectively, where the corresponding response times in raising up and recovering are 4 s @ 40 s (recovery time, RT), 7 s @ 72 s, and 8 s @ 129 s (**Extended Data Fig. 4d**). Since the human smell threshold for ethanol concentration is 80 ppm (33), it is obvious that higher heating temperature could contribute stronger olfactory feedback. In addition to the ethanol concentration measurement, we also conducted a human sensory test to investigate the heating temperature effect on the generated odor concentration, as shown in **Supplementary Fig. 11**. During the testing, all volunteers were wearing the Device 2 with 9 working OGs integrated for testing their responses to 9 different odors, where the temperatures of each OG was increased from 45 °C to 50 °C, 55 °C, and 60 °C with the lasting time of 1 min for each temperature point. By summarizing all volunteers’ response, we found that the recognition rates to these smells range from 0.73 to 1 with the average value of 0.93, further proving that the heating temperature of OGs is key for recognize odor concentration, and the heating temperatures ranging from 45 °C to 60 °C are sufficient to generate odors with the concentrations much larger than the human thresholds. **Extended Data Fig. 4e** shows the recovery time of generated ethanol concentration in air as a function of parallelly blown wind speed, ranging from 0 to 6.61 m/s, corresponding to a decreasing RT from 129 s to 9 s, which is induced by the increasing gas diffusion rate triggered by ambient wind. **Extended Data Fig. 4f** presents the electrical response of the commercial ethanol sensor as a function of the distance between the sensor and a working OG, and it is clear that longer distance could result in a higher delay time to trigger the sensor, including 1 cm @ 1.2 s, 3 cm @ 9.1 s, and 5 cm @ 15.6 s, which is further verified by the volunteer sensory test shown in **Supplementary Fig. 12**. To further investigate the odor generating performance of OGs inside Device 2, the ethanol sensor is embedded in Device 2 near to the OG integrating paraffin/ethanol (**Extended Data Figs. 4g, h**). Here, two different testing conditions are introduced: Condition one is that Device 2 releases ethanol for 5 mins, then shut down meanwhile an experimenter starts wearing the Device 2 until the monitored ethanol concentration recovers to original state (lower than 80 ppm), as shown in **Extended Data Figs. 4i, j**; Condition two is that the Device 2 releases ethanol for 5 mins, then shut down without further motion (**Extended Data Figs. 4i, j**), where it is obvious that human breathing could shorten ethanol recovery time (1.2 min @ Condition one and 3.1 min @ Condition two). To further investigate human breathing effect on generated odor recovery time, we conducted a volunteer test (**Supplementary Fig. 13**), that also associates with two testing conditions same as that in **Extended Data Figs. 4i, j**. Here, 9 different odors are adopted including lavender (a), orange (b), pineapple (c), green tea (d), lemon (e), peach (f), strawberry (g), minty (h), and lilac (i). The average recovery time of Condition one is 92.1 s, larger than 77.1 s of Condition two,

consistent with the conclusion shown in **Extended Data Figs. 4i, j**. As a result, it is obvious that the formerly generated odors will not interface with the newly generating odors after the corresponding odor recovery time, which is influenced by odor types and ambient airflow rate (**Extended Data Fig. 4 and Supplementary Fig. 13**). **Extended Data Figs. 4k, l** shows ethanol concentration generated by Device 2 as an experimenter is wearing the device, where the heating temperature of the working OG is target at 60 °C. Continuous breathing could obviously decrease the ethanol concentration to 1338 ppm around the ethanol sensor with a large fluctuation (± 180 ppm), which also demonstrates that the ethanol generation rate at the heating temperature of 60 °C is larger than human odor inhalation rate. As the generated odor concentration can be adjusted (**Extended Data Fig. 4d and Supplementary Fig. 11**), it is possible to avoid odor accumulation around human nose by adjusting the heating temperature of corresponding OGs in the Device 2.”

On Page 14 Line 22, the text has been modified as “**Fig. 4a and Movie S2** show a representative application of the skin-integrated olfaction interface for providing an immersive 4D movie watching experience (**Fig. 4ai**). Here, the subject in the movie is smelling a flower as it is approaching to her step by step while the OGs heating temperature simultaneously increases with the decreasing of the distance between the odor source and the subject nose, releasing a growing fragrance of roses for users in real world (**Figs. 4aia and 4aiii**). In the beginning of the movie as the flower is far away from the subject, the two OGs are in a state of stand by (heating temperature, 45 °C), then the heating temperatures start rising step by step from 45 °C, to 50 °C@5 s, to 55 °C@15 s, to 60 °C@25 s, then back to 45 °C@28 s (**Fig. 4aiv**). Here, we conducted a volunteer sensory test to investigate if users could react to fast temperature variation of OGs integrated in Device 1 (**Supplementary Fig. 19**). Following the time points in increasing the heating temperature of OGs shown in **Fig. 4aia and iv**, all volunteers’ reaction times to varying heating temperature of two OGs in the Device 1 are recorded in **Fig. 4aiv and Supplementary Fig. 19**, including 9.7 s (reaction time) @ 5 s (time point in increasing heating temperature), 20.8 s @ 15 s, and 28.4 s @ 25 s. As a result, our Device 1 could generate distinguished odor concentration to users, further demonstrating the great potential in movie watching. **Supplementary Fig. 20** shows a volunteer sensory test result in judging presence or absence of odors in silent OGs at room temperature. It is found that the recognition rates range from 0.22 to 0.89 with an average value of 0.43 for all volunteers, which demonstrates that human beings have distinguished odor concentration thresholds. In addition, different odor types also have distinguished recognition rate, ranging from 0.09 to 0.82. A second representative application of the olfaction interfaces is the message delivery through smell for a visual/auditory impaired person (**Fig. 4b**). By training visual/auditory impaired users to link some odors to specific events, such as lavender odor@ walking outdoors, pineapple odor@ having a lunch, and green tea odor@ teatime, the users could simultaneously react to the released odor from the face mask-based olfaction interface with 9 different odor options. During the training time, all the volunteers are required to continuously smell the odors generated by Device 2. The training associates with short-term (67 s for each odor, total of 10 min) and long-term (400 s for each odor, total of 60 min). When taking the test, the volunteers only have one chance to give their response to the generated odors with no fixed testing time. By adjusting various combinations of the 9 original odors, the olfaction interface is capable of offering extensive messages for users efficiently after training them a certain time. To prove the training time effect on the messages recognition rate, 9 volunteers worn the olfaction interface system with their eyes blinded and ears plugged for later 9 basic smell messages recognition test. Here, volunteers are asked to continuously smell 9 odors with equal training

time for each odor during the training process. As a result, the average recognition rate of the 9 smell messages is 55.6 % after a very short training time (10-min), while a longer training time (60-min) can significantly increase the recognition rate to 87.7 % (**Figs. 4bii and 4biii**). In this demonstration, we developed the smell message delivery system based on the OGs array instead of single OG due to the facts: (1) human being could easily distinguish different odor types, where one odor type corresponds to one message. (2) the delivered messages quantity could be significantly enhanced by blending several odors together, which could easily be realized by our Device 2, as shown in **Fig. 4eiii**. Although the single OG could also deliver smell messages by generating different odor concentration for users, it is difficult to build up a standard odor concentration division for different users who has distinguished odor concentration thresholds, as shown in **Supplementary Fig. 20**. **Fig. 4c** shows another application scenario where some odors generated from the olfactory interface system could be used for smoothing users' emotion. As odors could arouse human emotion by leading to the recall of emotional memories, the olfactory interface could be adopted for smoothing users' depressed mood from the stress (23) (**Fig. 4ci**). To verify the effect of odors on human emotion, 11 volunteers' reactions to 30 different odors (see details in Characterization) from the face mask based olfactory interface were collected as shown in **Fig. 4c** (all volunteers were in normal emotion before the testing, **Figs. 4cii and iii**), and the result demonstrates that the possibility of being joyful for these volunteers could reach up to 56%, 65%, 44%, and 56% when they smelled the grape, peach, orange, and strawberry odor, respectively (**Figs. 4civ, v**). It is also possible to give other emotions to users as they smelled clove and gardenia odors, inducing repulsive and peaceful emotions with the corresponding possibility of 0.82 and 0.51, as shown in **Fig. 4civ**. In addition, it is obvious that different users have distinguished emotion reactions to the same odors (**Figs. 4cv, vi**), which may be due to the volunteers' different emotional memories for same odor type. The large emotion reaction difference for each user may increase the difficulty in giving various desired emotions to users. **Fig. 4cv** shows the volunteers' emotion reaction to 30 odor types, where the results were collected from five times test for each volunteer. It is found that for each volunteer, their emotion reactions to some odor types keep constant, demonstrating that we could customize the odor types in the Device 2 for different users."

On Page 25 Line 9, the text has been modified as "**Characterization**. The perfumes adopted in Extended Data Fig. 3d are minty (perfume 1) and jasmine incense (perfumer 2). The odorous liquids used in Fig. 4b are lavender (a) @ walking outdoors(A), sweet orange (b) @ happiness (B), pineapple (c) having a lunch@ (C), green tea (d) @ teatime (D), lemon (e) @ summer (E), peach (f) @ sweetly (F), strawberry (g) @ lovely (G), minty (h) @ teeth brushing (H), and lilac (i) @ pure (I). Here, we measured the odor retention time for each pure perfumes shown Fig. 4b, ranging from 3.9 h to 141.8 h (**Supplementary Fig. 24**), and corresponding odor types and components have been summarized in **Supplementary Table 2**. By mixing the perfumes and wax at a heating temperature of 60 °C, 9 different odorous wax samples can be obtained, which can be still functional after exposing in air for three weeks (**Supplementary Fig. 25**). The odorous liquids used in Fig. 4c are ethanol (1), pineapple (2), grape (3), minty (4), rice (5), cream (6), gardenia (7), watermelon (8), vanilla (9), coffee milk (10), candy (11), coconut milk (12), coconut (13), milk (14), peach (15), pancake (16), orange (17), green tea (18), caramel (19), durian (20), lemon (21), strawberry (22), morning (23), ginger (24), clary sage (25), rosemary (26), lavender (27), clove (28), mojito (29), and cake (30)."

Extended Data Fig. 4. Ethanol concentration performance of the odor generators. **a, b, c** Ethanol concentration testing setup, where the test is placed inside an open box for minimizing the ambient wind interference to OGs. During the testing, a commercial resistance-variation-based ethanol sensor placed 1 cm above a working OG for low response time to generated ethanol around the OG. **d** Ethanol concentration generated by OG with 30 mg paraffin/ethanol (mass ratio, 10:3) with a controlled temperature switching between 45°C to 50°C, 55°C, and 60°C, respectively, corresponding to the increasing ethanol concentration peak values. **e** Ambient wind effects on the ethanol concentration dissipation rate, here an operating OG with a heating temperature of 60°C are suddenly shut down to cut off ethanol generation, and the wind is blown above the OG parallelly. **f** Delay time of a working OG as a function of the distance between the generator and ethanol sensor, which is induced by the ethanol diffusion rate. **g, h, i, j, k, l** Ethanol remaining time test in Device 2 with three testing conditions. Condition one is that experimenter continuously smell the generated ethanol after the OG with paraffin/ethanol embedded works for 5 min at a constant heating temperature of 60°C when he

is wearing the Device 2, and Condition two is that the ethanol sensor monitors the ethanol concentration after the OG works for 5 min in an open box (i, j). Condition three is that the ethanol sensor continuously monitors gas concentration as the experimenter is wearing the Device 2 (k, l).

Fig. S11. A volunteer testing showing the volunteers' odor recognition rate when they are wearing the Device 2 with 9 different perfume-based odors (see details in Characterization). Here, the temperature of each OGs will be increased from 45 °C to 60 °C at an interval of 5 °C, where each temperature will last 1 min before going up. The, the volunteers will be asked if they could sense the enhanced odor concentration.

Fig. S13. A volunteer testing showing the odor remaining time in Device 2. Here, all volunteers will be asked to smell the remaining odors in the Device 2, and record the time when the odors disappear. There are two testing conditions. Condition One is that volunteers continuously smell the odors when wearing the Device 2. Condition Two is that the volunteers come to smell the odors of the face mask every 10 s. Before volunteer smells the Device 2, the OG will generate the odor at a constant heating temperature of 60 °C in the face mask for 5 min.

Fig. S19. A volunteer testing showing the human reaction to the fast temperature variation of OGs with two odor types embedded when they are wearing Device 1 for testing. During the test, the Device 1 will be programmed to increase temperatures of OGs from 45 °C to 50 °C at

5 s, from 50 °C to 55°C at 15 s, from 55 °C to 60 °C at 25 s, then lasting 1 min until shutting down Device 1. Once the volunteers can sense the enhanced odor concentration, the time point will be recorded meanwhile. Two different odor types for two OGs in Device 1 are adopted, including lavender and lilac.

Fig. S20. A volunteer testing showing the recognition rate of volunteers in smelling the OGs at room temperature when they are the Device 1. Here, a, b, c, d, e, f, g, h, and i stand for lavender, orange, pineapple, green tea, lemon, peach, strawberry, minty, and lilac.

Fig. 4. Demonstrations of the olfactory interfaces. **a** A demonstration of the skin-integrated Device 1 in displaying olfaction feedback for providing an immersive experience to users during movie watching. Here, a girl in the movie is smelling a rose with the decreasing distance between the flower and her while the watcher could smell a more and more intense rose odor by wearing the olfaction interface. Here, OCL demonstrates the odor concentration level ranging from 1 to 3, corresponding to the low, middle, and high levels (iv). **b** A demonstration of the face-mask based olfaction interface in providing an alternative communication method, smell messages, for the disabled without the capability of vision and audio. By training users to link odors to specific information, the disabled users could efficiently communicate with others. **c** A demonstration of the Device 2 in assisting users to control their emotions as some specific odors could recall different human emotions. Here, all volunteers involved in the

emotion reaction test are verified in normal emotion range (**cii and ciii**). **d** A potential demonstration of the Device 2 for helping amnesic patients recall lost memories as odors perception is modulated by experience, leading to the recall of emotional memories. **e** Demonstration of the wireless olfaction interfaces in real time interaction between the user and a virtual subject for walking in a virtual garden surrounded by various fruit fragrance. By programming the Device 2, 9 OGs insides the device could work independently (**eii**), or work together (**eiii**).

Comment 3: It seems that face shape changes from person to person. Is it possible to fit your face mask to everyone even if the flexible materials are used? Is there any possibility that odor generator touches the skin around nose? Is it ok if the smell leaks from the face mask?

Our response: We thank the referee for this useful comment. We will answer the three questions one by one as following:

For the first question “Is it possible to fit your face mask to everyone even if the flexible materials are used”, we measured the practical situation by inviting users to wear the Device 2, all the users’ basic physical information is shown in Supplementary Fig. 7. Due to the flexible material (TPU, regular 3D printing material) used in face mask, the Device 2 can fit the three volunteers’ faces well. The authors think the face shape variates a lot among human beings, resulting in that the current face mask substrate may not fit all people in practical applications. But it is highly possible to customize suitable face mask according to the user’s face shape. We have put comments on this point on Page 6 Line 23.

For the second question “Is there any possibility that odor generator touches the skin around nose?”, we agree that touching between the device and the nose may cause thermal issue nearby the nose. To investigate this issue, we measured the thermal distribution of a working OG in two side views, as shown in Supplementary Figs. 18a, b, c. It is found that the air temperature 1.5 mm above the working OG is approaching to room temperature, which is quite safe to human skin. Then, we collected 11 volunteers’ distance value between their nose and working OGs when they are wearing Device 2, and the result has shown that all volunteers’ distance values are larger than 1.5 mm, demonstrating that Device 2 has no safety issue during applications. We have put comments on this point on Page 14 Line 6.

For the last question “Is it ok if the smell leaks from the face mask?”, we feel it is natural and Ok that odor generated by OGs may leak from the Device 2, as this is a natural diffusion process of smells. It is normal that the generated odor concentration is too high to be thoroughly inhaled by users, which may result in some odor leakage. In addition, to smooth users’ breathing as wearing the Device 2, we designed some breathing holes in the face mask, which may also accelerate the generated odors leakage from the face mask. For the leaked odors, it will fast dissipate in the open air. We have put comments on this point on Page 6 Line 23.

Modifications: On Page 6 Line 23, we modified the text as “In addition, an 8-channel multiplexer serve as a control terminal for reading the multiple-channel temperature data of the thermistors. 3D printed thermoplastic polyurethane (TPU) serving as the backbone materials of the face mask offers good flexibility, adaptivity, and customizable sizes for various face shapes, and thus realizes a user-friendly olfaction interface (**Extended Data Figs. 1e, f, and Supplementary Fig. 7**). For aesthetics, the face mask can be also painted into customized

colors, such as ash black that VR glasses/set typical use. In addition, the breathing holes in the face mask could ensure users' smooth breathing, which may result in a small number of odor leakage from the face mask.”

On Page 14 Line 6, we have modified as “**Supplementary Fig. 18** shows the safety testing of the olfactory devices as the heating temperatures of the OGs are stabilized around 60 °C. The thermal images of a working OG illustrate a fact that the ambient air temperature with distance of 1.5 mm from the working OG is around at room temperature (**Supplementary Figs. 18b, c**). Then, we measured the distance between human nose to working OGs from 11 volunteers (**Supplementary Fig. 18d**). It is clear that all volunteers (including 4 males and 7 females) could safely wear the Device 2 with all the distance values greater than 1.5 mm (**Supplementary Figs. 18e, f**).”

Fig. S7. Optical images of users wearing the Device 2, whose basic physical information have been provided. The flexible face mask substrate enables the wide applications for various users.

Fig. S18. Safety operation of Device 2 and Device 1. (a) A OG 3D model built up in software ABAQUS (Analysis User's Manual 6.14). (b, c) Two views of thermal distribution of a working OG with the working temperature stabilized around 60 °C, where the working temperature is the one around the thermistor for melting odorous wax insides OGs. (d) Schematic diagram of the inside layout of Device 2 when users are wearing the Device 2. Here, the distance between users' nose and working OGs decides the safety of Device 2 during operation. (e, f) 11 volunteers' distance values between volunteers' nose and working OGs, including 4 males and 7 females. All volunteers can safely wear the Device 2 during its operation as the minimum distance value is 2 mm, where the air temperature is room temperature. (g) Optical image of a user wearing the Device 1 on his upper lip with a value (23 mm) showing the distance between the OG and the nearby human nose. (h) Thermal distribution of the Device 1 bottom side with the working temperatures of two OGs stabilized around 60 °C. The temperature peak is 32.2 °C, demonstrating that the Device 1 has no potential risk to the attached human skin.

Comment 4: The authors use PDMS. Since smell seems to be absorbed by PDMS, the odor generator might not be able to be used many times because of remaining smell. Moreover, is there any smell adsorption at the face mask?

Our response: We thank the referee for this comment, which is an important issue for the odor generators. The reviewer pointed out the absorption behaviors of PDMS, and is also needed to consider other materials used for the olfaction system. So, the smell residual duration/remaining time on the device is an important issue need to be evaluated. To investigate the remaining time of various odors, we conducted a series of tests, and the results are shown in Extended Data

Figs. 4g, h, i, j and Supplementary Fig. 13.

In revision, the new results have been made and shown in Extended Data Figs. 4g, h, where we measured the ethanol remaining time after the corresponding OG operating for 5 mins in Device 2. Here, two different testing conditions are considered: Condition one is that Device 2 releases ethanol for 5 mins, then shut down meanwhile an experimenter starts wearing the Device 2 until the monitored ethanol concentration recovers to original state (lower than 80 ppm), as shown in Extended Data Figs. 4i, j. Condition two is that the Device 2 releases ethanol for 5 mins, then shut down without further motion (Extended Data Figs. 4i, j). It is obvious that human breathing could shorten ethanol recovery time (1.2 min @ Condition one and 3.1 min @ Condition two).

To further investigate perfume-based odors remaining time, we conducted a volunteer testing (Supplementary Fig. 13), where there are also two testing conditions same to that in Extended Data Figs. 4i, j. Here, 9 different odors are adopted including lavender (a), orange (b), pineapple (c), green tea (d), lemon (e), peach (f), strawberry (g), minty (h), and lilac (i). The average remaining time (also called recovery time, RT) of Condition one is 92.1 s, larger than 77.1 s of Condition two, consistent with the conclusion shown in Extended Data Figs. 4i, j. As a result, we think the both OGs and olfaction systems could be used repeatedly as the remaining odor in the devices could dissipate in air in a short time. We have modified the text on Page 11 Line 22.

Modifications: On Page 11 Line 22, the text has been modified as “To further investigate the odor generating performance of OGs insides Device 2, the ethanol sensor is embedded in Device 2 near to the OG integrating paraffin/ethanol (**Extended Data Figs. 4g, h**). Here, two different testing conditions are introduced: Condition one is that Device 2 releases ethanol for 5 mins, then shut down meanwhile an experimenter starts wearing the Device 2 until the monitored ethanol concentration recovers to original state (lower than 80 ppm), as shown in **Extended Data Figs. 4i, j**; Condition two is that the Device 2 releases ethanol for 5 mins, then shut down without further motion (**Extended Data Figs. 4i, j**), where it is obvious that human breathing could shorten ethanol recovery time (1.2 min @ Condition one and 3.1 min @ Condition two). To further investigate human breathing effect on generated odor recovery time, we conducted a volunteer test (**Supplementary Fig. 13**), that also associates with two testing conditions same as that in **Extended Data Figs. 4i, j**. Here, 9 different odors are adopted including lavender (a), orange (b), pineapple (c), green tea (d), lemon (e), peach (f), strawberry (g), minty (h), and lilac (i). The average recovery time of Condition one is 92.1 s, larger than 77.1 s of Condition two, consistent with the conclusion shown in **Extended Data Figs. 4i, j**. As a result, it is obvious that the formerly generated odors will not interface with the newly generating odors after the corresponding odor recovery time, which is influenced by odor types and ambient airflow rate (**Extended Data Fig. 4 and Supplementary Fig. 13**). **Extended Data Figs. 4k, l** shows ethanol concentration generated by Device 2 as an experimenter is wearing the device, where the heating temperature of the working OG is target at 60 °C. Continuous breathing could obviously decrease the ethanol concentration to 1338 ppm around the ethanol sensor with a large fluctuation (± 180 ppm), which also demonstrates that the ethanol generation rate at the heating temperature of 60 °C is larger than human odor inhalation rate.”

Fig. S13. A volunteer testing showing the odor remaining time in Device 2. Here, all volunteers will be asked to smell the remaining odors in the Device 2, and record the time when the odors disappear. There are two testing conditions. Condition One is that volunteers continuously smell the odors when wearing the Device 2. Condition Two is that the volunteers come to smell the odors of the face mask every 10 s. Before volunteer smells the Device 2, the OG will generate the odor at a constant heating temperature of 60 °C in the face mask for 5 min.

Extended Data Fig. 4. Ethanol concentration performance of the odor generators. **a, b, c** Ethanol concentration testing setup, where the test is placed inside an open box for minimizing the ambient wind interference to OGs. During the testing, a commercial resistance-variation-based ethanol sensor placed 1 cm above a working OG for low response time to generated ethanol around the OG. **d** Ethanol concentration generated by OG with 30 mg paraffin/ethanol (mass ratio, 10:3) with a controlled temperature switching between 45°C to 50°C, 55°C, and 60°C, respectively, corresponding to the increasing ethanol concentration peak values. **e** Ambient wind effects on the ethanol concentration dissipation rate, here an operating OG with a heating temperature of 60 °C are suddenly shut down to cut off ethanol generation, and the wind is blown above the OG parallelly. **f** Delay time of a working OG as a function of the distance between the generator and ethanol sensor, which is induced by the ethanol diffusion rate. **g, h, i, j, k, l** Ethanol remaining time test in Device 2 with three testing conditions. Condition one is that experimenter continuously smell the generated ethanol after the OG with paraffin/ethanol embedded works for 5 min at a constant heating temperature of 60°C when he

is wearing the Device 2, and Condition two is that the ethanol sensor monitors the ethanol concentration after the OG works for 5 min in an open box (i, j). Condition three is that the ethanol sensor continuously monitors gas concentration as the experimenter is wearing the Device 2 (k, l).

Comment 5: Although ethanol is easily evaporated, low-volatile compounds might stick to housing for long time. The author should mention the influence of remaining smell.

Our response: We thank the referee for this useful comment. We have realized this issue that some odor types may stick to the housing electronics for some time. To investigate the odor remaining time, we conducted a series of experiments, as shown in Extended Data Figs. 4d, e, g, h, i, j, and Supplementary Fig. 13.

In Extended Data Fig. 4d, we measured the ethanol remaining time (also recovery time, RT) as a function of the initial ethanol concentration at a constant ethanol releasing time of 5 s. It is found that the higher initial ethanol concentration could extend the recovery time from 40 s to 129 s with the corresponding heating temperature of 50 °C, 55 °C, and 60 °C, respectively. Here, the recovery time for ethanol is defined as the duration time when the ethanol concentration decreases to the human threshold (80 ppm). Therefore, if the latter odor is generated during the ethanol recovery period, users could still sense ethanol, influencing the next odor.

In Extended Data Fig. 4e, we investigate the parallelly blown airflow rate effect on the ethanol remaining time as the initial concentration reaches up to ~4531 ppm with the corresponding OG heating temperature of 60 °C. Before shutting down the OG, the ethanol has been released for 5 s, same as that in Extended Data Fig. 4d. It is found that higher airflow rate could largely shorten the ethanol recovery time due to the increased gas diffusion rate in open air. Due to the shortened ethanol recovery time realized by high airflow rate, it requires less time to generate a new odor after ethanol source is shut down.

In Extended Data Figs. 4i, j, we measured the ethanol remaining time in Device 2 as the odor has been released for 5 min at the highest heating temperature. Considering the practical application, two different testing conditions are introduced: Condition one is that Device 2 releases ethanol for 5 mins, then shut down meanwhile an volunteer starts wearing the Device 2 until the monitored ethanol concentration recovers to original state (lower than 80 ppm), as shown in Extended Data Figs. 4i, j. Condition two is that the Device 2 releases ethanol for 5 mins, then shut down without further motion (Extended Data Figs. 4i, j), where it is obvious that human breathing could shorten ethanol recovery time (1.2 min @ Condition one and 3.1 min @ Condition two).

To further investigate perfume-based odors remaining time, we conducted a volunteer test (Supplementary Fig. 13), where there are also two testing conditions same to that in Extended Data Figs. 4i, j. All volunteers will be asked to smell the remaining odors in the Device 2, and record the time when the odors disappear for the two testing conditions. Here, the 9 different odors are adopted including lavender (a), orange (b), pineapple (c), green tea (d), lemon (e), peach (f), strawberry (g), minty (h), and lilac (i). The average odor remaining time of Condition one is 92.1 s, larger than 77.1 s of Condition two, consistent with the conclusion shown in Extended Data Figs. 4i, j. As a result, odors generated by OGs will stick to the housing electronics for a time ranging from 56 s to 186 s on average for different odor types, as shown in Supplementary Fig. 13 and Extended Data Fig. 4i.

In summary, different odor types, different testing conditions, different odor releasing

time, and different ambient airflow rate could all influence the corresponding odor recovery time. As human being could sensitively sense the odor concentration variation, the odor recovery time could be directly sensed by users. We have modified the text on Page 11 Line 22.

Modifications: On Page 11 Line 22, the text has been modified as “To further investigate the odor generating performance of OGs insides Device 2, the ethanol sensor is embedded in Device 2 near to the OG integrating paraffin/ethanol (**Extended Data Figs. 4g, h**). Here, two different testing conditions are introduced: Condition one is that Device 2 releases ethanol for 5 mins, then shut down meanwhile an experimenter starts wearing the Device 2 until the monitored ethanol concentration recovers to original state (lower than 80 ppm), as shown in **Extended Data Figs. 4i, j**; Condition two is that the Device 2 releases ethanol for 5 mins, then shut down without further motion (**Extended Data Figs. 4i, j**), where it is obvious that human breathing could shorten ethanol recovery time (1.2 min @ Condition one and 3.1 min @ Condition two). To further investigate human breathing effect on generated odor recovery time, we conducted a volunteer test (**Supplementary Fig. 13**), that also associates with two testing conditions same as that in **Extended Data Figs. 4i, j**. Here, 9 different odors are adopted including lavender (a), orange (b), pineapple (c), green tea (d), lemon (e), peach (f), strawberry (g), minty (h), and lilac (i). The average recovery time of Condition one is 92.1 s, larger than 77.1 s of Condition two, consistent with the conclusion shown in **Extended Data Figs. 4i, j**. As a result, it is obvious that the formerly generated odors will not interface with the newly generating odors after the corresponding odor recovery time, which is influenced by odor types and ambient airflow rate (**Extended Data Fig. 4 and Supplementary Fig. 13**). **Extended Data Figs. 4k, l** shows ethanol concentration generated by Device 2 as an experimenter is wearing the device, where the heating temperature of the working OG is target at 60 °C. Continuous breathing could obviously decrease the ethanol concentration to 1338 ppm around the ethanol sensor with a large fluctuation (± 180 ppm), which also demonstrates that the ethanol generation rate at the heating temperature of 60 °C is larger than human odor inhalation rate. As the generated odor concentration can be adjusted (**Extended Data Fig. 4d and Supplementary Fig. 11**), it is possible to avoid odor accumulation around human nose by adjusting the heating temperature of corresponding OGs in the Device 2.”

Fig. S13. A volunteer testing showing the odor remaining time in Device 2. Here, all volunteers will be asked to smell the remaining odors in the Device 2, and record the time when the odors disappear. There are two testing conditions. Condition One is that volunteers continuously smell the odors when wearing the Device 2. Condition Two is that the volunteers come to smell the odors of the face mask every 10 s. Before volunteer smells the Device 2, the OG will generate the odor at a constant heating temperature of 60 °C in the face mask for 5 min.

Extended Data Fig. 4. Ethanol concentration performance of the odor generators. **a, b, c** Ethanol concentration testing setup, where the test is placed inside an open box for minimizing the ambient wind interference to OGs. During the testing, a commercial resistance-variation-based ethanol sensor placed 1 cm above a working OG for low response time to generated ethanol around the OG. **d** Ethanol concentration generated by OG with 30 mg paraffin/ethanol (mass ratio, 10:3) with a controlled temperature switching between 45°C to 50°C, 55°C, and 60°C, respectively, corresponding to the increasing ethanol concentration peak values. **e** Ambient wind effects on the ethanol concentration dissipation rate, here an operating OG with a heating temperature of 60°C are suddenly shut down to cut off ethanol generation, and the wind is blown above the OG parallelly. **f** Delay time of a working OG as a function of the distance between the generator and ethanol sensor, which is induced by the ethanol diffusion rate. **g, h, i, j, k, l** Ethanol remaining time test in Device 2 with three testing conditions. Condition one is that experimenter continuously smell the generated ethanol after the OG with paraffin/ethanol embedded works for 5 min at a constant heating temperature of 60°C when he

is wearing the Device 2, and Condition two is that the ethanol sensor monitors the ethanol concentration after the OG works for 5 min in an open box (i, j). Condition three is that the ethanol sensor continuously monitors gas concentration as the experimenter is wearing the Device 2 (k, l).

Comment 6: The proposed devices do not seem to have the capability of blending smells. If so, the number of smells to be presented is limited. Moreover, it is necessary to change the cartridge if a user changes the smell. The author should mention those points.

Our response: We thank the referee for this comment. The devices can blend smells by actuating multiple actuators at the same time. While the portfolio of blending is determined by the channels of the devices. The skin-interfaced one (Device 1) has two odor generators, while the mask device (Device 2) has the capability of blending 9 odors in many ways. In the revised manuscript, we add more data to better clarify this point with the results shown in Fig. 4e, where the device supports multiple different OGs working together, and all OGs integrated in Device 2 can independently work. Therefore, 9 odor generation channels in Device 2 could provide high odor options to users, meaning that users do not need to change the cartridge frequently. In the revised manuscript, we have put comments on this point on Page 17 Line 10.

Modifications: On Page 17 Line 10, the text has been modified as “**Fig. 4ei** shows a girl wearing a series of VR devices containing our olfaction interface at home for interacting with virtual environment, where the VR scenario associates with walking in a virtual garden surrounding by various plants, including apple trees, pear trees, etc. By picking up an apple near to the virtual subject nose, the user could synchronously smell apple odor generated by the olfaction interface. **Figs. 4eii, iii** show the electrical response of the face mask based olfactory interface with 9 OGs operating independently or at the same time for proving the capability of the olfaction system in coordinating each OGs working independently. To realize the 9 channel OGs independent operation, four 8 bits shift registers (74HC595, Texas Instruments Inc.) and an 8-channel multiplexer (ADG708BRUZ-REEL7, Analog Devices Inc.) are connected to the MCU for extending the digital GP I/Os and digital pins of the microcontroller, capable of controlling 9 OGs independently. To ensure the stable operation of the circuit, the operating OGs number is limited to 5 as the potential large ripple voltage and coupled electromagnetic field induced by the multiple working OGs (> 5) may affect the operation of the microcontroller. While blending 5 odors simultaneously can provide a great portfolio of odors feedbacks.”

Fig. 4. Demonstrations of the olfaction interfaces. **a** A demonstration of the skin-integrated Device 1 in displaying olfaction feedback for providing an immersive experience to users during movie watching. Here, a girl in the movie is smelling a rose with the decreasing distance between the flower and her while the watcher could smell a more and more intense rose odor by wearing the olfaction interface. Here, OCL demonstrates the odor concentration level ranging from 1 to 3, corresponding to the low, middle, and high levels (iv). **b** A demonstration of the face-mask based olfaction interface in providing an alternative communication method, smell messages, for the disabled without the capability of vision and audio. By training users to link odors to specific information, the disabled users could efficiently communicate with others. **c** A demonstration of the Device 2 in assisting users to control their emotions as some specific odors could recall different human emotions. Here, all volunteers involved in the

emotion reaction test are verified in normal emotion range (**cii and ciii**). **d** A potential demonstration of the Device 2 for helping amnesic patients recall lost memories as odors perception is modulated by experience, leading to the recall of emotional memories. **e** Demonstration of the wireless olfaction interfaces in real time interaction between the user and a virtual subject for walking in a virtual garden surrounded by various fruit fragrance. By programming the Device 2, 9 OGs insides the device could work independently (**eii**), or work together (**eiii**).

Comment 7: Where is the battery placed. Since it seems the power of a few hundreds mW is consumed for actuators and heater, In Extended data Fig.2, 12V and 3.7 battery are used. Their weights might be the problem.

Our response: We thank the referee for this comment. The power management system is based on batteries which are located behind users' neck. We added the detail information in the revised manuscript, as shown in Supplementary Fig. 23. The weight of the current power management is 107.6 g, which is a "OK" number for wearable devices. To ensure the comfort to users, we use PDMS and cloth to fully encapsulate the power management system meanwhile insulating the inner circuit away from mounted human skin. Benefitted from the flexible encapsulation layers, the power system is flexible and comfortable to attach onto human neck for long term operation. While in the future, we will working on the materials and devices to further optimize the power consumption of the OGs to decrease the burden of batteries. One solution method is to design a smaller OG, which may require a lower power input. We have put comments on this point on Page 22, Line 11.

Modifications: On Page 22 Line 11, the text has been modified as "For operating the Device 2, 7 digital and 2 ADC GP I/Os of a MCU (ATMEGA328P-MU, Microchip Technology Inc.) are exploited while the data collected by the MCU is directly transmitted to a Bluetooth module (WH-BLE103, Jinan USR IOT Technology Limited) installed on the equipment by serial communication for wireless communication with a computer. To provide 3.3, 5, and 16 V to the system, an external 12 V battery (1800 mAh) and a 3.7 V battery (2000 mAh) are converted through one LDO (LM2940LD-5.0/NOPB, Texas Instruments Inc.) and one DC-to-DC voltage boost converter (LM2731XMFX/NOPB, Texas Instrument Inc.), as shown in **Supplementary Fig. 23**. Here, the power management system mass is 107.6 g. By adopting the PDMS and cloth as the two encapsulation layers (PDMS and cloth), the power system could be flexible and comfortable to users for long-term application."

Fig. S23. Optical images of the flexible power management system, where there are two batteries, including a 3.7 V battery (2000 mAh) and a 12 V battery (1800 mAh).

Comment 8: The temperature of wax is about 60 degree Celsius. Is it enough to evaporate low-volatile compounds since those boiling points are typically around 200 degree Celsius? Moreover, is there any possibility that a user feels heat?

Our response: We thank the referee for this useful comment. The working principle of OGs is to melt odorous wax, which later releases the odor mixed insides. The melting point of the adopted wax is 50 °C, and all heating temperatures higher than 50 °C can fast melt the odorous wax. The odor releasing highly relies on gas free diffusion, induced by Brownian motion of odor molecules, which is similar to pure perfumes. Higher heating temperature of OGs could increase the intensity of the Brownian motion of odor molecules, contributing to a higher odor concentration in ambient air, which is also proven by the volunteer sensory test shown in Supplementary Fig. 11. It is possible to further increase the heating temperature of OGs to 100 °C if needed. In this manuscript, the highest heating temperature is 60 °C as this temperature could contribute a high odor concentration to users, as shown in Extended Data Fig. 4 and Supplementary Fig. 13. Too high heating temperature requires extremely high-power consumption, and raises serious safety issues. For the second question, we measured the thermal distribution of a working OG in two side views, as shown in Supplementary Figs. 18a, b, c. It is found that the air temperature 1.5 mm above the working OG is approaching to room

temperature, which is quite safe to human skin. Then, we collected 11 volunteers' distance value between their nose and working OGs when they are wearing Device 2, and the result has shown that all volunteers' distance values are larger than 1.5 mm, demonstrating that the users cannot feel heat when wearing the device. We have modified the text on Page 10 Line 22.

Modifications: On Page 10 Line 22, the text has been modified as “**Extended Data Fig. 4** shows the gaseous ethanol generation performance of the OGs. To minimize the unpredictable ambient wind effect on the performance of the OG, the whole experimental setup is placed inside an open box, where a commercial ethanol sensor is fixed 1 cm above a working OG, continuously monitoring the nearby ethanol concentration by reading the voltage variation of a 5 k Ω resistor connected in series (**Extended Data Figs. 4a, b, c**). By switching the heating temperature of the OG from 45 °C to 50 °C, to 55 °C, and finally to 60 °C (**Supplementary Fig. 10**), the ambient ethanol concentrations could reach up to 531 ppm, 2821 ppm, and 4531 ppm, respectively, where the corresponding response times in raising up and recovering are 4 s @ 40 s (recovery time, RT), 7 s @ 72 s, and 8 s @ 129 s (**Extended Data Fig. 4d**). Since the human smell threshold for ethanol concentration is 80 ppm (33), it is obvious that higher heating temperature could contribute stronger olfactory feedback. In addition to the ethanol concentration measurement, we also conducted a human sensory test to investigate the heating temperature effect on the generated odor concentration, as shown in **Supplementary Fig. 11**. During the testing, all volunteers were wearing the Device 2 with 9 working OGs integrated for testing their responses to 9 different odors, where the temperatures of each OG was increased from 45 °C to 50 °C, 55 °C, and 60 °C with the lasting time of 1 min for each temperature point. By summarizing all volunteers' response, we found that the recognition rates to these smells range from 0.73 to 1 with the average value of 0.93, further proving that the heating temperature of OGs is key for recognize odor concentration, and the heating temperatures ranging from 45 °C to 60 °C are sufficient to generate odors with the concentrations much larger than the human thresholds.”

On Page 14 Line 6, the text has been modified as “**Supplementary Fig. 18** shows the safety testing of the olfactory devices as the heating temperatures of the OGs are stabilized around 60 °C. The thermal images of a working OG illustrate a fact that the ambient air temperature with distance of 1.5 mm from the working OG is around at room temperature (**Supplementary Figs. 18b, c**). Then, we measured the distance between human nose to working OGs from 11 volunteers (**Supplementary Fig. 18d**). It is clear that all volunteers (including 4 males and 7 females) could safely wear the Device 2 with all the distance values greater than 1.5 mm (**Supplementary Figs. 18e, f**). For the Device 1, the open design allows the OGs to exhibit good thermal convection and also to maintain a safe distance to users with a big gap (23 mm) between OGs and user's nose, since the temperature at the skin interface is only 32.2 °C, as shown in **Supplementary Figs. 18g, h**.”

Fig. S11. A volunteer testing showing the volunteers' odor recognition rate when they are wearing the Device 2 with 9 different perfume-based odors (see details in Characterization). Here, the temperature of each OGs will be increased from 45 °C to 60 °C at an interval of 5 °C, where each temperature will last 1 min before going up. The, the volunteers will be asked if they could sense the enhanced odor concentration.

Extended Data Fig. 4. Ethanol concentration performance of the odor generators. **a, b, c** Ethanol concentration testing setup, where the test is placed inside an open box for minimizing the ambient wind interference to OGs. During the testing, a commercial resistance-variation-based ethanol sensor placed 1 cm above a working OG for low response time to generated ethanol around the OG. **d** Ethanol concentration generated by OG with 30 mg paraffin/ethanol (mass ratio, 10:3) with a controlled temperature switching between 45°C to 50°C, 55°C, and 60°C, respectively, corresponding to the increasing ethanol concentration peak values. **e** Ambient wind effects on the ethanol concentration dissipation rate, here an operating OG with a heating temperature of 60°C are suddenly shut down to cut off ethanol generation, and the wind is blown above the OG parallelly. **f** Delay time of a working OG as a function of the distance between the generator and ethanol sensor, which is induced by the ethanol diffusion rate. **g, h, i, j, k, l** Ethanol remaining time test in Device 2 with three testing conditions. Condition one is that experimenter continuously smell the generated ethanol after the OG with paraffin/ethanol embedded works for 5 min at a constant heating temperature of 60°C when he is wearing the Device 2, and Condition two is that the ethanol sensor monitors the ethanol concentration after the OG works for 5 min in an open box (i, j). Condition three is that the ethanol sensor continuously monitors gas concentration as the experimenter is wearing the Device 2 (k, l).

Fig. S18. Safety operation of Device 2 when users are wearing Device 2. (a) A OG 3D model built up in software ABAQUS (Analysis User's Manual 6.14). (b, c) Two views of thermal distribution of a working OG with the working temperature is target at 60 °C, where the working temperature is the one around the thermistor for melting odorous wax inside OGs. (d) Schematic diagram of the inside layout of Device 2 when users are wearing the Device 2. Here, the distance between users' nose and working OGs decides the safety of Device 2 during operation. (e, f) 11 volunteers' distance values between volunteers' nose and working OGs, including 4 males and 7 females. All volunteers can safely wear the Device 2 during its operation as the minimum distance value is 2 mm, where the air temperature is room temperature.

Comment 9: Is there any possibility that the smell is accumulated in space between the face mask and face even if the temperature changes rapidly? The authors should show the data of gas sensor or sensory test result.

Our response: We thank the referee for this useful comment. By monitoring the generated odor concentration variation after shutting down corresponding OG in Device 2, we could figure out if the odor could accumulate in the space between the face and face mask of Device 2. Therefore, we conducted a series of new experiment to investigate this issue, as shown in Extended Data Figs. 4g, h, i, j and Supplementary Fig. 13.

For the ethanol test shown in Extended Data Figs. 4g, h, i, j, we monitored the remaining ethanol concentration after the Device 2 releasing ethanol for 5 mins at the corresponding OG heating temperature of 60 °C. During the test, the experimenter starts wearing the Device after releasing ethanol for 5 mins with the results shown in Extended Data Figs. 4i, j. It takes 1.2 min to clear the remaining ethanol in the space between human face and face mask. Therefore, for the ethanol odor type, the odor will not accumulate in the space between the face mask and face for a long time (less than 1.2 min) as the human breathing could accelerate the odor diffusion to the surroundings through the preserved holes in the face mask, as shown in Supplementary Fig. 7. As shown in Extended Data Figs. 4k, l, the ethanol concentration variations when the experimenter is wearing the Device 2 with the ethanol OG continuously operating at 60 °C. The stabilized ethanol concentration is 1338 ppm, demonstrating that the odor generation rate is higher than human odor inhalation rate. As our olfaction system could adjust odor generation rate by varying the corresponding OG heating temperature (Extended Data Fig. 4d), it is possible to decrease the odor generation rate to a lower level than the users' odor inhalation rate, resulting in no odor accumulation during the odor releasing process.

For the 9 regular perfume-based odors test, we called for 11 volunteers to sense the remaining odors in Device 2 by wearing it after the device releasing the corresponding odor for 5 mins, and the average recovery time is 1.3 min, as shown in Supplementary Fig. 13. As a result, the generated odors will remain in the space between the face and the Device for a short time (56 s to 186 s on average for different odor types) after the corresponding OGs shutting down. Therefore, the perfume-based odors could also not accumulate in the space between human face and Device 2 for a long time (less than 186 s). We also conducted a new volunteer sensory test, as shown in Supplementary Fig. 11. In this sensory test, all volunteers are asked if they could obviously sense the enhanced odor concentration realized by increasing the corresponding OG heating temperature from 45 °C to 60 °C at a constant interval of 5 °C. The high average recognition rate (0.93) verifies that our olfaction system could adjust the generated perfume-based odors concentration. By adjusting the generated odor concentration, it is possible to avoid odor accumulation in the space between face and face mask by adjusting the heating temperature of OGs in Device 2.

In summary, we think the odor generated by Device 2 could not accumulate in space between the face mask and human face once the remaining odor is disappeared with the time less than 186 s. As many parameters could influence odor remaining time, including but not limited to the ambient airflow rate, heating temperature of OGs, the odor releasing time, odor types, and users' respiration characteristics, the conservative 186 s illustrated above may be further compressed according to the practical application scenario. For the Device 1, as the Device 1 adopt the open design with the OGs working in the open air, there is also no odor accumulation issue. We have put comments on this point on Page 11 Line 22.

Modifications: On Page 11 Line 22, the text has been modified as “To further investigate the

odor generating performance of OGs insides Device 2, the ethanol sensor is embedded in Device 2 near to the OG integrating paraffin/ethanol (**Extended Data Figs. 4g, h**). Here, two different testing conditions are introduced: Condition one is that Device 2 releases ethanol for 5 mins, then shut down meanwhile an experimenter starts wearing the Device 2 until the monitored ethanol concentration recovers to original state (lower than 80 ppm), as shown in **Extended Data Figs. 4i, j**; Condition two is that the Device 2 releases ethanol for 5 mins, then shut down without further motion (**Extended Data Figs. 4i, j**), where it is obvious that human breathing could shorten ethanol recovery time (1.2 min @ Condition one and 3.1 min @ Condition two). To further investigate human breathing effect on generated odor recovery time, we conducted a volunteer test (**Supplementary Fig. 13**), that also associates with two testing conditions same as that in **Extended Data Figs. 4i, j**. Here, 9 different odors are adopted including lavender (a), orange (b), pineapple (c), green tea (d), lemon (e), peach (f), strawberry (g), minty (h), and lilac (i). The average recovery time of Condition one is 92.1 s, larger than 77.1 s of Condition two, consistent with the conclusion shown in **Extended Data Figs. 4i, j**. As a result, it is obvious that the formerly generated odors will not interface with the newly generating odors after the corresponding odor recovery time, which is influenced by odor types and ambient airflow rate (**Extended Data Fig. 4 and Supplementary Fig. 13**). **Extended Data Figs. 4k, l** shows ethanol concentration generated by Device 2 as an experimenter is wearing the device, where the heating temperature of the working OG is target at 60 °C. Continuous breathing could obviously decrease the ethanol concentration to 1338 ppm around the ethanol sensor with a large fluctuation (± 180 ppm), which also demonstrates that the ethanol generation rate at the heating temperature of 60 °C is larger than human odor inhalation rate. As the generated odor concentration can be adjusted (**Extended Data Fig. 4d and Supplementary Fig. 11**), it is possible to avoid odor accumulation around human nose by adjusting the heating temperature of corresponding OGs in the Device 2.”

Fig. S11. A volunteer testing showing the volunteers’ odor recognition rate when they are wearing the Device 2 with 9 different perfume-based odors (see details in Characterization). Here, the temperature of each OGs will be increased from 45 °C to 60 °C at an interval of 5 °C, where each temperature will last 1 min before going up. The, the volunteers will be asked if they could sense the enhanced odor concentration.

Fig. S13. A volunteer testing showing the odor remaining time in Device 2. Here, all volunteers will be asked to smell the remaining odors in the Device 2, and record the time when the odors disappear. There are two testing conditions. Condition One is that volunteers continuously smell the odors when wearing the Device 2. Condition Two is that the volunteers come to smell the odors of the face mask every 10 s. Before volunteer smells the Device 2, the OG will generate the odor at a constant heating temperature of 60 °C in the face mask for 5 min.

Extended Data Fig. 4. Ethanol concentration performance of the odor generators. **a, b, c** Ethanol concentration testing setup, where the test is placed inside an open box for minimizing the ambient wind interference to OGs. During the testing, a commercial resistance-variation-based ethanol sensor placed 1 cm above a working OG for low response time to generated ethanol around the OG. **d** Ethanol concentration generated by OG with 30 mg paraffin/ethanol (mass ratio, 10:3) with a controlled temperature switching between 45°C to 50°C, 55°C, and 60°C, respectively, corresponding to the increasing ethanol concentration peak values. **e** Ambient wind effects on the ethanol concentration dissipation rate, here an operating OG with a heating temperature of 60°C are suddenly shut down to cut off ethanol generation, and the wind is blown above the OG parallelly. **f** Delay time of a working OG as a function of the distance between the generator and ethanol sensor, which is induced by the ethanol diffusion rate. **g, h, i, j, k, l** Ethanol remaining time test in Device 2 with three testing conditions. Condition one is that experimenter continuously smell the generated ethanol after the OG with paraffin/ethanol embedded works for 5 min at a constant heating temperature of 60°C when he

is wearing the Device 2, and Condition two is that the ethanol sensor monitors the ethanol concentration after the OG works for 5 min in an open box (i, j). Condition three is that the ethanol sensor continuously monitors gas concentration as the experimenter is wearing the Device 2 (k, l).

Comment 10: The reviewer understood that the temperature of the olfactory generator is precisely and rapidly controlled. However, The gas sensor response highly fluctuated in Extended data Fig.3 l. What is the reason for the fluctuation? Is there rapid recovery when the gas concentration decreases? What happens if low-volatile compound instead of ethanol is used? Although the authors show the temperature change in several situations, they should show the gas sensor response, not the temporal temperature change.

Our response: We thank the referee for this useful comment. Please find the response to the three questions in the following:

For the first question “What is the reason for the fluctuation?”, the reason is that the data was measured in the open air, where ambient airflow could induce the obvious ethanol concentration fluctuation. To solve this problem, in revision, we remeasured the ethanol concentration as a function of heating temperature of OGs in a box for minimizing the unpredictable ambient airflow effect on the results, as shown in new Extended Data Figs. 4a, b, c, d, e. It is obvious that higher heating temperature could contribute to a higher ethanol concentration, as shown in Extended Data Fig. 4d. We have put the comment on Page 10 Line 22.

For the second question “Is there rapid recovery when the gas concentration decreases?”, the ethanol concentration recovery time (RT) in air highly relies on the ambient wind speed, where we adjusted this factor from 0 to 6.61 m/s to do the RT test, and the corresponding RT was measured to be from 129 s to 9 s. We have put the comment on Page 11 Line 16.

For the third question “What happens if low-volatile compound instead of ethanol is used?”, we conducted two volunteer tests using 9 regular perfumes (see details in Characterization): one is that the 11 volunteers are asked if they could sense the enhanced odor concentration when the heating temperature of OGs integrated in Device 2 is increased from 45 °C to 60 °C at an interval of 5 °C, where each temperature will last 1 min before going up, as shown in Supplementary Fig. 11. It is found that the average recognition rate reaches up to 0.93, which proves that the heating temperature of OGs could decide the generated odor concentration for the perfume-based odors. The other one is an odor remaining time sensory test, as shown in Supplementary Fig. 13. We record the 9 different odor recovery time sensed by 11 volunteers, where the Device 2 releases the corresponding odor for 5 mins, then volunteers will be asked when the remaining odors in the Device 2 disappear, as shown in Supplementary Fig. 13. There are two testing conditions. Condition One is that volunteers continuously smell the odors when wearing the Device 2. Condition Two is that the volunteers come to smell the odors of the face mask every 10 s. Before volunteer smells the Device 2, the OG will generate the odor at a constant heating temperature of 60 °C in the face mask for 5 min. Here, 9 different odors are adopted including lavender (a), orange (b), pineapple (c), green tea (d), lemon (e), peach (f), strawberry (g), minty (h), and lilac (i). The average recovery time of Condition one is 92.1 s, larger than 77.1 s of Condition two. Some perfumes adopted in Supplementary Fig. 13 are low-volatile with the duration time in room temperature reaching up to dozens of hours, such as lilac and peach, as shown in Supplementary Fig. 24. We have put the comment on Page 11 Line 7.

Modifications: On Page 10 Line 22, we modified the text as “**Extended Data Fig. 4** shows the gaseous ethanol generation performance of the OGs. To minimize the unpredictable ambient wind effect on the performance of the OG, the whole experimental setup is placed inside an open box, where a commercial ethanol sensor is fixed 1 cm above a working OG, continuously monitoring the nearby ethanol concentration by reading the voltage variation of a 5 k Ω resistor connected in series (**Extended Data Figs. 4a, b, c**). By switching the heating temperature of the OG from 45 °C to 50 °C, to 55 °C, and finally to 60 °C (**Supplementary Fig. 10**), the ambient ethanol concentrations could reach up to 531 ppm, 2821 ppm, and 4531 ppm, respectively, where the corresponding response times in raising up and recovering are 4 s @ 40 s (recovery time, RT), 7 s @ 72 s, and 8 s @ 129 s (**Extended Data Fig. 4d**). Since the human smell threshold for ethanol concentration is 80 ppm (33), it is obvious that higher heating temperature could contribute stronger olfactory feedback.”

On Page 11 Line 16, we have modified the text as “**Extended Data Fig. 4e** shows the recovery time of generated ethanol concentration in air as a function of parallelly blown wind speed, ranging from 0 to 6.61 m/s, corresponding to a decreasing RT from 129 s to 9 s, which is induced by the increasing gas diffusion rate triggered by ambient wind.”

On Page 11 Line 7, we have modified the text as “In addition to the ethanol concentration measurement, we also conducted a human sensory test to investigate the heating temperature effect on the generated odor concentration, as shown in **Supplementary Fig. 11**. During the testing, all volunteers were wearing the Device 2 with 9 working OGs integrated for testing their responses to 9 different odors, where the temperatures of each OG was increased from 45 °C to 50 °C, 55 °C, and 60 °C with the lasting time of 1 min for each temperature point. By summarizing all volunteers’ response, we found that the recognition rates to these smells range from 0.73 to 1 with the average value of 0.93, further proving that the heating temperature of OGs is key for recognize odor concentration, and the heating temperatures ranging from 45 °C to 60 °C are sufficient to generate odors with the concentrations much larger than the human thresholds.”

On Page 12 Line 5, we have modified the text as “To further investigate human breathing effect on generated odor recovery time, we conducted a volunteer test (**Supplementary Fig. 13**), that also associates with two testing conditions same as that in **Extended Data Figs. 4i, j**. Here, 9 different odors are adopted including lavender (a), orange (b), pineapple (c), green tea (d), lemon (e), peach (f), strawberry (g), minty (h), and lilac (i). The average recovery time of Condition one is 92.1 s, larger than 77.1 s of Condition two, consistent with the conclusion shown in **Extended Data Figs. 4i, j**. As a result, it is obvious that the formerly generated odors will not interface with the newly generating odors after the corresponding odor recovery time, which is influenced by odor types and ambient airflow rate (**Extended Data Fig. 4 and Supplementary Fig. 13**).”

On Page 25 Line 9, the text has been modified as “**Characterization**. The perfumes adopted in **Extended Data Fig. 3d** are minty (perfume 1) and jasmine incense (perfumer 2). The odorous liquids used in **Fig. 4b** are lavender (a) @ walking outside(A), sweet orange (b) @ happiness (B), pineapple (c) having a lunch@ (C), green tea (d) @ teatime (D), lemon (e) @ summer (E), peach (f) @ sweet (F), strawberry (g) @ lovely (G), minty (h) @ teeth brushing (H), and lilac (i) @ pure (I). Here, we measured the odor retention time for each pure perfume

shown Fig. 4b, ranging from 3.9 h to 141.8 h (**Supplementary Fig. 24**), and corresponding odor types and components have been summarized in **Supplementary Table 2**. By mixing the perfumes and wax at a heating temperature of 60 °C, 9 different odorous wax samples can be obtained, which can be still functional after exposing in air for three weeks (**Supplementary Fig. 25**). The odorous liquids used in Fig. 4c are ethanol (1), pineapple (2), grape (3), minty (4), rice (5), cream (6), gardenia (7), watermelon (8), vanilla (9), coffee milk (10), candy (11), coconut milk (12), coconut (13), milk (14), peach (15), pancake (16), orange (17), green tea (18), caramel (19), durian (20), lemon (21), strawberry (22), morning (23), ginger (24), clary sage (25), rosemary (26), lavender (27), clove (28), mojito (29), and cake (30).”

Extended Data Fig. 4. Ethanol concentration performance of the odor generators. **a, b, c** Ethanol concentration testing setup, where the test is placed inside an open box for minimizing the ambient wind interference to OGs. During the testing, a commercial resistance-variation-based ethanol sensor placed 1 cm above a working OG for low response time to generated

ethanol around the OG. **d** Ethanol concentration generated by OG with 30 mg paraffin/ethanol (mass ratio, 10:3) with a controlled temperature switching between 45°C to 50°C, 55°C, and 60°C, respectively, corresponding to the increasing ethanol concentration peak values. **e** Ambient wind effects on the ethanol concentration dissipation rate, here an operating OG with a heating temperature of 60°C are suddenly shut down to cut off ethanol generation, and the wind is blown above the OG parallelly. **f** Delay time of a working OG as a function of the distance between the generator and ethanol sensor, which is induced by the ethanol diffusion rate. **g, h, i, j, k, l** Ethanol remaining time test in Device 2 with three testing conditions. Condition one is that experimenter continuously smell the generated ethanol after the OG with paraffin/ethanol embedded works for 5 min at a constant heating temperature of 60°C when he is wearing the Device 2, and Condition two is that the ethanol sensor monitors the ethanol concentration after the OG works for 5 min in an open box (i, j). Condition three is that the ethanol sensor continuously monitors gas concentration as the experimenter is wearing the Device 2 (k, l).

Fig. S11. A volunteer testing showing the volunteers' odor recognition rate when they are wearing the Device 2 with 9 different perfume-based odors (see details in Characterization). Here, the temperature of each OGs will be increased from 45 °C to 60 °C at an interval of 5 °C, where each temperature will last 1 min before going up. The, the volunteers will be asked if they could sense the enhanced odor concentration.

Fig. S13. A volunteer testing showing the odor remaining time in Device 2. Here, all volunteers will be asked to smell the remaining odors in the Device 2, and record the time when the odors disappear. There are two testing conditions. Condition One is that volunteers continuously smell the odors when wearing the Device 2. Condition Two is that the volunteers come to smell the odors of the face mask every 10 s. Before volunteer smells the Device 2, the OG will generate the odor at a constant heating temperature of 60 °C in the face mask for 5 min.

Fig. S24. The retention time of 9 pure perfumes. Here, a, b, c, d, e, f, g, h, and i stand for lavender, orange, pineapple, green tea, lemon, peach, strawberry, minty, and lilac.

Comment 11: Is it necessary to control the temperature precisely? If the authors would like to control the concentration, only Extended data Fig.3 l is not enough.

Our response: We thank the referee for this useful comment. According to the new added Extended Data Fig. 4d and Supplementary Fig. 11, it is necessary to control the heating temperature precisely for realizing distinguished odor concentration for users. For ethanol concentration measurement in Extended Data Fig. 4d, we investigate the heating temperature of OGs effect on the generated ethanol concentration, where the test was performed in an open box for minimizing the unpredictable ambient wind around the working OGs. It is found that increasing heating temperature of OGs could contribute to higher ethanol concentration. The volunteer test shown in Supplementary Fig. 11 is to investigate if volunteers could obviously sense the enhanced odor concentrations for 9 different odor types (see details in Characterization) generated by Device 2 when the heating temperature of OGs are increased from 45 °C to 50 °C, 55 °C, and 60 °C step by step. The high average recognition rate, 0.93, demonstrates that all volunteers confirm to sense the odor concentration variations by Device 2. We have modified the text on Page 10 Line 22.

Modifications: On Page 10 Line 22, the text has been modified as “**Extended Data Fig. 4** shows the gaseous ethanol generation performance of the OGs. To minimize the unpredictable ambient wind effect on the performance of the OG, the whole experimental setup is placed inside an open box, where a commercial ethanol sensor is fixed 1 cm above a working OG, continuously monitoring the nearby ethanol concentration by reading the voltage variation of a 5 kΩ resistor connected in series (**Extended Data Figs. 4a, b, c**). By switching the heating temperature of the OG from 45 °C to 50 °C, to 55 °C, and finally to 60 °C (**Supplementary Fig. 10**), the ambient ethanol concentrations could reach up to 531 ppm, 2821 ppm, and 4531 ppm, respectively, where the corresponding response times in raising up and recovering are 4 s @ 40 s (recovery time, RT), 7 s @ 72 s, and 8 s @ 129 s (**Extended Data Fig. 4d**). Since the human smell threshold for ethanol concentration is 80 ppm (33), it is obvious that higher heating temperature could contribute stronger olfactory feedback. In addition to the ethanol concentration measurement, we also conducted a human sensory test to investigate the heating temperature effect on the generated odor concentration, as shown in **Supplementary Fig. 11**. During the testing, all volunteers were wearing the Device 2 with 9 working OGs integrated for testing their responses to 9 different odors, where the temperatures of each OG was increased from 45 °C to 50 °C, 55 °C, and 60 °C with the lasting time of 1 min for each temperature point. By summarizing all volunteers’ response, we found that the recognition rates to these smells

range from 0.73 to 1 with the average value of 0.93, further proving that the heating temperature of OGs is key for recognize odor concentration, and the heating temperatures ranging from 45 °C to 60 °C are sufficient to generate odors with the concentrations much larger than the human thresholds.”

Extended Data Fig. 4. Ethanol concentration performance of the odor generators. **a, b, c** Ethanol concentration testing setup, where the test is placed inside an open box for minimizing the ambient wind interference to OGs. During the testing, a commercial resistance-variation-based ethanol sensor placed 1 cm above a working OG for low response time to generated ethanol around the OG. **d** Ethanol concentration generated by OG with 30 mg paraffin/ethanol (mass ratio, 10:3) with a controlled temperature switching between 45°C to 50°C, 55°C, and 60°C, respectively, corresponding to the increasing ethanol concentration peak values. **e** Ambient wind effects on the ethanol concentration dissipation rate, here an operating OG with a heating temperature of 60°C are suddenly shut down to cut off ethanol generation, and the

wind is blown above the OG parallelly. (f) Delay time of a working OG as a function of the distance between the generator and ethanol sensor, which is induced by the ethanol diffusion rate. g, h, i, j, k, l Ethanol remaining time test in Device 2 with three testing conditions. Condition one is that experimenter continuously smell the generated ethanol after the OG with paraffin/ethanol embedded works for 5 min at a constant heating temperature of 60°C when he is wearing the Device 2, and Condition two is that the ethanol sensor monitors the ethanol concentration after the OG works for 5 min in an open box (i, j). Condition three is that the ethanol sensor continuously monitors gas concentration as the experimenter is wearing the Device 2 (k, l).

Fig. S11. A volunteer testing showing the volunteers’ odor recognition rate when they are wearing the Device 2 with 9 different perfume-based odors (see details in Characterization). Here, the temperature of each OGs will be increased from 45 °C to 60 °C at an interval of 5 °C, where each temperature will last 1 min before going up. The, the volunteers will be asked if they could sense the enhanced odor concentration.

Comment 12: Fig.4 aiV. The authors showed the temperature change. However, the concentration change should be shown here. Otherwise, the sensory test should be done.

Our response: We thank the referee for this useful comment. As the odor types shown in Fig. 4a are based on floral perfumes for the 4D movie watching demonstration, we couldn’t find a commercial sensor to measure the perfume-based odor concentration. Therefore, we conduct a new volunteer sensory test to investigate the human reaction to odor concentration variations in Supplementary Fig. 19 and Fig. 4aiv. In the volunteer test, 11 volunteers are wearing the Device 1 with two floral odor types, including lavender and lilac, and the heating temperature of the two OGs integrated in the Device 1 are programmed to increase from 45 °C to 50 °C at 5 s, to 55 °C at 15 s, and to 60 °C at 25 s, which is same to the test shown in Figs. 4aiii, iv. With the variations of heating temperature of the OGs in Device 1 (Fig. 4iv), all volunteers’ reaction time to the obviously sensed odor concentration variations and corresponding odor concentration level (OCL) are recorded as shown in Supplementary Fig. 19 and Fig. 4iv, including 9.7 s (reaction time) @ 5 s (time point in increasing heating temperature), 20.8 s @ 15 s, and 28.4 s @ 25 s. As a result, our Device 1 could generate distinguished odor concentration to users, further demonstrating the great potential in movie watching. We have added the information on Page 14 Line 22.

Modifications: On Page 14 Line 22, the text has been modified as “**Fig. 4a** and **Movie S2** show a representative application of the skin-integrated olfaction interface for providing an immersive 4D movie watching experience (**Fig. 4ai**). Here, the subject in the movie is smelling

a flower as it is approaching to her step by step while the OGs heating temperature simultaneously increases with the decreasing of the distance between the odor source and the subject nose, releasing a growing fragrance of roses for users in real world (**Figs. 4a_{ii} and 4a_{iii}**). In the beginning of the movie as the flower is far away from the subject, the two OGs are in a state of stand by (heating temperature, 45 °C), then the heating temperatures start rising step by step from 45 °C, to 50 °C@5 s, to 55 °C@15 s, to 60 °C@25 s, then back to 45 °C@28 s (**Fig. 4a_{iv}**). Here, we conducted a volunteer sensory test to investigate if users could react to fast temperature variation of OGs integrated in Device 1 (**Supplementary Fig. 19**). Following the time points in increasing the heating temperature of OGs shown in **Fig. 4a_{iii} and iv**, all volunteers' reaction times to varying heating temperature of two OGs in the Device 1 are recorded in **Fig. 4a_{iv} and Supplementary Fig. 19**, including 9.7 s (reaction time) @ 5 s (time point in increasing heating temperature), 20.8 s @ 15 s, and 28.4 s @ 25 s. As a result, our Device 1 could generate distinguished odor concentration to users, further demonstrating the great potential in movie watching. **Supplementary Fig. 20** shows a volunteer sensory test result in judging presence or absence of odors in silent OGs at room temperature. It is found that the recognition rates range from 0.22 to 0.89 with an average value of 0.43 for all volunteers, which demonstrates that human beings have distinguished odor concentration thresholds. In addition, different odor types also have distinguished recognition rate, ranging from 0.09 to 0.82.”

Fig. 4. Demonstrations of the olfaction interfaces. **a** A demonstration of the skin-integrated Device 1 in displaying olfaction feedback for providing an immersive experience to users during movie watching. Here, a girl in the movie is smelling a rose with the decreasing distance between the flower and her while the watcher could smell a more and more intense rose odor by wearing the olfaction interface. Here, OCL demonstrates the odor concentration level ranging from 1 to 3, corresponding to the low, middle, and high levels (iv). **b** A demonstration of the face-mask based olfaction interface in providing an alternative communication method, smell messages, for the disabled without the capability of vision and audio. By training users to link odors to specific information, the disabled users could efficiently communicate with others. **c** A demonstration of the Device 2 in assisting users to control their emotions as some

specific odors could recall different human emotions. Here, all volunteers involved in the emotion reaction test are verified in normal emotion range (**cii and ciii**). **d** A potential demonstration of the Device 2 for helping amnesic patients recall lost memories as odors perception is modulated by experience, leading to the recall of emotional memories. **e** Demonstration of the wireless olfaction interfaces in real time interaction between the user and a virtual subject for walking in a virtual garden surrounded by various fruit fragrance. By programming the Device 2, 9 OGs inside the device could work independently (**eii**), or work together (**eiii**)

Fig. S19. A volunteer testing showing the human reaction to the fast temperature variation of OGs with two odor types embedded when they are wearing Device 1 for testing. During the test, the Device 1 will be programmed to increase temperatures of OGs from 45 °C to 50 °C at 5 s, from 50 °C to 55°C at 15 s, from 55 °C to 60 °C at 25 s, then lasting 1 min until shutting down Device 1. Once the volunteers can sense the enhanced odor concentration, the time point will be recorded meanwhile. Two different odor types for two OGs in Device 1 are adopted, including lavender and lilac.

Comment 13: It is not clear whether a participant can recognize the change of smell intensity because there is no result of sensory test.

Our response: We thank the referee for the comment. In revision, we conducted a sensory test (11 volunteers) to verify if participants could sense 9 different perfume-based odors concentration variation by our olfaction systems, as shown in Supplementary Fig. 11. In this sensory test, 11 volunteers were asked if they could sense the increase of odor concentration of the mask device when the heating temperature of the working OGs increased from 45 °C to 60 °C at an interval of 5 °C, where each temperature will last 1 min before going up. As a result, the volunteers' recognition rates range from 0.73 to 1 with the average value of 0.93, proving that our device could enable users to sense the change of desired odor intensity. We have put comments on this point on Page 11 Line 7.

Modifications: On Page 11 Line 7, the text has been modified as “In addition to the ethanol concentration measurement, we also conducted a human sensory test to investigate the heating temperature effect on the generated odor concentration, as shown in **Supplementary Fig. 11**. During the testing, all volunteers were wearing the Device 2 with 9 working OGs integrated for testing their responses to 9 different odors, where the temperatures of each OG was increased from 45 °C to 50 °C, 55 °C, and 60 °C with the lasting time of 1 min for each temperature point. By summarizing all volunteers' response, we found that the recognition rates to these smells range from 0.73 to 1 with the average value of 0.93, further proving that the heating temperature

of OGs is key for recognize odor concentration, and the heating temperatures ranging from 45 °C to 60 °C are sufficient to generate odors with the concentrations much larger than the human thresholds.”

Fig. S11. A volunteer testing showing the volunteers’ odor recognition rate when they are wearing the Device 2 with 9 different perfume-based odors (see details in Characterization). Here, the temperature of each OGs will be increased from 45 °C to 60 °C at an interval of 5 °C, where each temperature will last 1 min before going up. The, the volunteers will be asked if they could sense the enhanced odor concentration.

Comment 14: The authors mentioned the temporal sequence of the temperature at the top of page 11. How did they determine it? What is the relationship between temperature and concentration?

Our response: We thank the referee for this useful comment. Please find the answers to the two questions in the following:

For the first question “The authors mentioned the temporal sequence of the temperature at the top of page 11. How did they determine it?”, We developed a control panel to control the heating temperature of OG varying between 45 °C and 50 °C with each working cycle time of 20 s, as shown in Movie S1. For the Figs. 3f, g, we fixed an OG on a commercial oscillator, which supports controllable vibration amplitude and frequency, as shown in Fig. 3e and Supplementary Fig. 14. By changing the vibration amplitude and frequency of the oscillator, we could monitor the heating temperature of the OG through the self-developed control panel same to that in Fig. 3c. We have put the comment on Page 13 Line 9.

For the second question “What is the relationship between temperature and concentration?”, we investigate the heating temperature of OGs effect on the generated ethanol concentration (Supplementary Fig. 4d), where the test is performed in an open box for minimizing the unpredictable ambient wind around the working OGs. It is found that the higher heating temperature of OGs could contribute to higher ethanol concentration. The volunteer test shown in Supplementary Fig. 11 is to investigate if volunteers could obviously sense the enhanced odor concentrations for 9 different perfume-based odor types (see details in Characterization) generated by Device 2 when the heating temperature of OGs are increased from 45 °C to 50 °C, 55 °C, and 60 °C step by step. The high average recognition rate, 0.93, demonstrates that all volunteers could obviously sense the perfume-based odor concentration variations realized by the Device 2. We have modified the text on Page 11 Line 2.

Modifications: For the first question, we have modified the text on Page 13 Line 9 as “**Fig. 3c** shows the heating temperature response of the OG as repeatedly switching the temperature

between 45 °C and 50 °C for over 7000 working cycles, which is realized by a self-developed circuit (**Movie S1**), with enlarged data details in **Fig. 3d**. Along the long-term, continuous operation, the heating temperature signals of the OGs maintains stable and consistent values without any interruption, demonstrating its high stability and robustness. In addition to the long-term operation, vibration induced uncurtains are always a great concern as the wearer may keep moving. **Fig. 3e** and **Supplementary Figs. 14, 15** show two typical stability tests for providing vibration to a working OG with controllable vibration amplitude, frequency, and bending angles by adopting a commercial oscillator (**Supplementary Fig. 14**) and a self-developed programmable bending platform (**Supplementary Fig. 15**).

For the second question, we have modified the text on Page 11 Line 2 as “By switching the heating temperature of the OG from 45 °C to 50 °C, to 55 °C, and finally to 60 °C (**Supplementary Fig. 10**), the ambient ethanol concentrations could reach up to 531 ppm, 2821 ppm, and 4531 ppm, respectively, where the corresponding response times in raising up and recovering are 4 s @ 40 s (recovery time, RT), 7 s @ 72 s, and 8 s @ 129 s (**Extended Data Fig. 4d**). Since the human smell threshold for ethanol concentration is 80 ppm (33), it is obvious that higher heating temperature could contribute stronger olfactory feedback. In addition to the ethanol concentration measurement, we also conducted a human sensory test to investigate the heating temperature effect on the generated odor concentration, as shown in **Supplementary Fig. 11**. During the testing, all volunteers were wearing the Device 2 with 9 working OGs integrated for testing their responses to 9 different odors, where the temperatures of each OG was increased from 45 °C to 50 °C, 55 °C, and 60 °C with the lasting time of 1 min for each temperature point. By summarizing all volunteers’ response, we found that the recognition rates to these smells range from 0.73 to 1 with the average value of 0.93, further proving that the heating temperature of OGs is key for recognize odor concentration, and the heating temperatures ranging from 45 °C to 60 °C are sufficient to generate odors with the concentrations much larger than the human thresholds.”

Extended Data Fig. 4. Ethanol concentration performance of the odor generators. **a, b, c** Ethanol concentration testing setup, where the test is placed inside an open box for minimizing the ambient wind interference to OGs. During the testing, a commercial resistance-variation-based ethanol sensor placed 1 cm above a working OG for low response time to generated ethanol around the OG. **d** Ethanol concentration generated by OG with 30 mg paraffin/ethanol (mass ratio, 10:3) with a controlled temperature switching between 45°C to 50°C, 55°C, and 60°C, respectively, corresponding to the increasing ethanol concentration peak values. **e** Ambient wind effects on the ethanol concentration dissipation rate, here an operating OG with a heating temperature of 60°C are suddenly shut down to cut off ethanol generation, and the wind is blown above the OG parallelly. **(f)** Delay time of a working OG as a function of the distance between the generator and ethanol sensor, which is induced by the ethanol diffusion rate. **g, h, i, j, k, l** Ethanol remaining time test in Device 2 with three testing conditions. Condition one is that experimenter continuously smell the generated ethanol after the OG with paraffin/ethanol embedded works for 5 min at a constant heating temperature of 60°C when he

is wearing the Device 2, and Condition two is that the ethanol sensor monitors the ethanol concentration after the OG works for 5 min in an open box (i, j). Condition three is that the ethanol sensor continuously monitors gas concentration as the experimenter is wearing the Device 2 (k, l).

Fig. S11. A volunteer testing showing the volunteers’ odor recognition rate when they are wearing the Device 2 with 9 different perfume-based odors (see details in Characterization). Here, the temperature of each OGs will be increased from 45 °C to 60 °C at an interval of 5 °C, where each temperature will last 1 min before going up. The, the volunteers will be asked if they could sense the enhanced odor concentration.

Comment 15: The authors should mention that types of smells appear in Material and Method.

Our response: We thank the referee for this useful comment. We have added a new table (Supplementary Table 2) to summarize the odor types frequently used in the Fig. 4, Supplementary Figs. 11, 12, 13, 19, 20, and 24.

Modifications: On Page 25 Line 9, the text has been modified as “**Characterization.** The perfumes adopted in Extended Data Fig. 3d are minty (perfume 1) and jasmine incense (perfumer 2). The odorous liquids used in Fig. 4b are lavender (a) @ walking outdoors(A), sweet orange (b) @ happiness (B), pineapple (c) having a lunch@ (C), green tea (d) @ teatime (D), lemon (e) @ summer (E), peach (f) @ sweetly (F), strawberry (g) @ lovely (G), minty (h) @ teeth brushing (H), and lilac (i) @ pure (I). Here, we measured the odor retention time for each pure perfumes shown Fig. 4b, ranging from 3.9 h to 141.8 h (**Supplementary Fig. 24**), and corresponding odor types and components have been summarized in **Supplementary Table 2**. By mixing the perfumes and wax at a heating temperature of 60 °C, 9 different odorous wax samples can be obtained, which can be still functional after exposing in air for three weeks (**Supplementary Fig. 25**). The odorous liquids used in Fig. 4c are ethanol (1), pineapple (2), grape (3), minty (4), rice (5), cream (6), gardenia (7), watermelon (8), vanilla (9), coffee milk (10), candy (11), coconut milk (12), coconut (13), milk (14), peach (15), pancake (16), orange (17), green tea (18), caramel (19), durian (20), lemon (21), strawberry (22), morning (23), ginger (24), clary sage (25), rosemary (26), lavender (27), clove (28), mojito (29), and cake (30).”

Table S2. Basic information of the frequently used perfumes.

Odor name	Components	Types	Pure odor duration time (h)	Purchase link

Lavender	Lavandula angustifolia	Herbal note	19.8	https://item.taobao.com/item.htm?spm=a1z09.2.0.0.2f502e8dt3WivQ&id=559055572709& u=o2jk453d28b
Lemon	Chrysophoron, Musk, Grapefruit, Vetiver grass, Petitgrain, Bergamot, Vanilla, Flores aurantia, Cedar.	Fruit note	8.3	https://detail.tmall.com/item_o.htm?abbucket=4&id=644765865889&m=cca438a41f37ac4225f73511b702476&spm=a312a.7700824.w4011-23876712565.48.4577b863Kij1q6&sku_properties=1626521:1088826456
Strawberry	Strawberry, Peach, Mandarin orange, Lily of the valley, Violet, Musk.	Fruit note	24.3	https://detail.tmall.com/item_o.htm?abbucket=4&id=646128618010&m=cca438a41f37ac4225f73511b702476&spm=a312a.7700824.w4011-23876712565.64.4577b863Kij1q6&sku_properties=1626521:6210120
Minty	Mint, Mentha spicata, Niaouli, Musk. Amberggris.	Minty note	4	https://item.taobao.com/item.htm?spm=a1z09.2.0.0.2f502e8dt3WivQ&id=634588501729& u=o2jk453feed
Green tea	Lemon, Agrumi di Sicilia, Chamomile, Green tea, Jasmine tea, White flower, Chrysophoron, Honey, Ambrette seeds.	Woody note	31.5	https://detail.tmall.com/item_o.htm? u=o2jk453a828&id=656299236516&spm=a1z09.2.0.0.2f502e8dt3WivQ&skuld=4736630916824
Lilac	Eugenia caryophyllus	Flower note	141.8	https://item.taobao.com/item.htm?spm=a1z09.2.0.0.2f502e8dt3WivQ&id=578197849564& u=o2jk453353a
Peach	Citrus, Peach, Lily of the valley, Sandalwood	Fruit note	92.5	https://detail.tmall.com/item_o.htm? u=p2jk453e642&id=645656358985&spm=a1z09.2.0.0.2f502e8dt3WivQ&sku_properties=1626521:6210120
Pineapple	Pineapple	Fruit note	9.2	https://detail.tmall.com/item_o.htm? u=p2jk453f27a&id=645414838669&spm=a1z09.2.0.0.2f502e8dt3WivQ&sku_properties=1626521:6210120
Orange	Citrus aurantium dulcis	Citrus note	3.9	https://item.taobao.com/item.htm?spm=a1z09.2.0.0.2f502e8dt3WivQ&id=559145201790& u=o2jk4536a6f

Comment 16: Fig.4 cii. Did the authors use nine depressed volunteers? If so, how did the authors control the mental situation before sniffing the smell? The author mentioned the sweat orange was useful. Do they mean the sweet orange?

Our response: We thank the referee for the comment. In the original data of Fig. 4cii, we conduct user studies on 9 volunteers, who didn't have obvious depression symptoms. So, we reconduct an volunteer test to investigate the human emotion reaction to different odors generated by Device 2. To assess the new 11 volunteers' emotions before the volunteer test, all volunteers filled a standard emotion assessment form (Hospital Anxiety and Depression Scale, HADS), and the results have shown that all involved volunteers (4 males and 7 females) are in normal emotion range (Figs. 4cii, iii). During the test, the volunteers could decide the odors generation sequence by controlling the corresponding heating temperature of OGs in the Device 2, and we further increase the odor types number from the original 9 to current 30 for a comprehensive test. As a result, the possibility of being joyful for these volunteers could reach up to 56%, 65%, 44%, and 56% when they smelled the grape, peach, orange, and strawberry odor, respectively (Figs. 4civ, v). In addition, it is obvious that different users have distinguished emotion reactions to the 30 odors (Figs. 4cv, vi), which means that users could select their preferred odor types integrated insides Device 2 for emotion smoothing.

For the second question, thank the reviewer point out this typo, which should be "sweet orange", and we have corrected this throughout the manuscript. We have put comments on this point on Page 25 Line 9.

Modifications: On Page 16 Line 11, the text has been modified as "Fig. 4c shows another

application scenario where some odors generated from the olfactory interface system could be used for soothing users' emotion. As odors could arouse human emotion by leading to the recall of emotional memories, the olfactory interface could be adopted for soothing users' depressed mood from the stress (23) (**Fig. 4ci**). To verify the effect of odors on human emotion, 11 volunteers' reactions to 30 different odors (see details in Characterization) from the face mask based olfactory interface were collected as shown in **Fig. 4c** (all volunteers were in normal emotion before the testing, **Figs. 4cii and iii**), and the result demonstrates that the possibility of being joyful for these volunteers could reach up to 56%, 65%, 44%, and 56% when they smelled the grape, peach, orange, and strawberry odor, respectively (**Figs. 4civ, v**). It is also possible to give other emotions to users as they smelled clove and gardenia odors, inducing repulsive and peaceful emotions with the corresponding possibility of 0.82 and 0.51, as shown in **Fig. 4civ**. In addition, it is obvious that different users have distinguished emotion reactions to the same odors (**Figs. 4cv, vi**), which may be due to the volunteers' different emotional memories for same odor type. The large emotion reaction difference for each user may increase the difficulty in giving various desired emotions to users. **Fig. 4cv** shows the volunteers' emotion reaction to 30 odor types, where the results were collected from five times test for each volunteer. It is found that for each volunteer, their emotion reactions to some odor types keep constant, demonstrating that we could customize the odor types in the Device 2 for different users."

On Page 25 Line 9, the text has been modified as "**Characterization**. The perfumes adopted in Extended Data Fig. 3d are minty (perfume 1) and jasmine incense (perfumer 2). The odorous liquids used in Fig. 4b are lavender (a) @ walking outdoors(A), sweet orange (b) @ happiness (B), pineapple (c) having a lunch@ (C), green tea (d) @ teatime (D), lemon (e) @ summer (E), peach (f) @ sweet (F), strawberry (g) @ lovely (G), minty (h) @ teeth brushing (H), and lilac (i) @ pure (I). Here, we measured the odor retention time for each pure perfumes shown Fig. 4b, ranging from 3.9 h to 141.8 h (**Supplementary Fig. 24**), and corresponding odor types and components have been summarized in **Supplementary Table 2**. By mixing the perfumes and wax at a heating temperature of 60 °C, 9 different odorous wax samples can be obtained, which can be still functional after exposing in air for three weeks (**Supplementary Fig. 25**). The odorous liquids used in Fig. 4c are ethanol (1), pineapple (2), grape (3), minty (4), rice (5), cream (6), gardenia (7), watermelon (8), vanilla (9), coffee milk (10), candy (11), coconut milk (12), coconut (13), milk (14), peach (15), pancake (16), orange (17), green tea (18), caramel (19), durian (20), lemon (21), strawberry (22), morning (23), ginger (24), clary sage (25), rosemary (26), lavender (27), clove (28), mojito (29), and cake (30)."

Fig. 4. Demonstrations of the olfaction interfaces. **a** A demonstration of the skin-integrated Device 1 in displaying olfaction feedback for providing an immersive experience to users during movie watching. Here, a girl in the movie is smelling a rose with the decreasing distance between the flower and her while the watcher could smell a more and more intense rose odor by wearing the olfaction interface. Here, OCL demonstrates the odor concentration level ranging from 1 to 3, corresponding to the low, middle, and high levels (iv). **b** A demonstration of the face-mask based olfaction interface in providing an alternative communication method, smell messages, for the disabled without the capability of vision and audio. By training users to link odors to specific information, the disabled users could efficiently communicate with others. **c** A demonstration of the Device 2 in assisting users to control their emotions as some specific odors could recall different human emotions. Here, all volunteers involved in the

emotion reaction test are verified in normal emotion range (**cii and ciii**). **d** A potential demonstration of the Device 2 for helping amnesic patients recall lost memories as odors perception is modulated by experience, leading to the recall of emotional memories. **e** Demonstration of the wireless olfaction interfaces in real time interaction between the user and a virtual subject for walking in a virtual garden surrounded by various fruit fragrance. By programming the Device 2, 9 OGs inside the device could work independently (**eii**), or work together (**eiii**).

Comment 17: Fig.4e. The relationship between Fig.4ei and Fig. 4eii is not clear. In the text, the author mentioned they proved the capability of the olfaction system in coordinating each OG working independently. However, it seems temperatures of all channels changed synchronously.

Our response: We thank the referee for this useful comment. All OGs integrated in Device 2 could operate independently. To prove this, we program the Device 2 to make each OGs work for 20 s one by one, with the heating temperatures switching between 45 °C and 55 °C, as shown in new Fig. 4eii. In addition, our olfaction system also supports multiple OGs working together, as shown in Fig. 4eiii. We have modified the text on Page 17 Line 6.

Modifications: On Page 17 Line 6, the text has been modified as “Another important typical application of the olfaction interfaces is in VR/AR, as which arise extensive attentions recently especially since the conception of Metaverse for building a 3D virtual world in connection with real people through a wearable VR interface. To provide an immersive experience for users, the odor feedback provided by olfaction interfaces could enrich the functions and entertainment of the VR interface. **Fig. 4ei** shows a girl wearing a series of VR devices containing our olfaction interface at home for interacting with virtual environment, where the VR scenario associates with walking in a virtual garden surrounding by various plants, including apple trees, pear trees, etc. By picking up an apple near to the virtual subject nose, the user could synchronously smell apple odor generated by the olfaction interface. **Figs. 4eii, iii** show the electrical response of the face mask based olfactory interface with 9 OGs operating independently or at the same time for proving the capability of the olfaction system in coordinating each OGs working independently.”

Fig. 4. Demonstrations of the olfaction interfaces. **a** A demonstration of the skin-integrated Device 1 in displaying olfaction feedback for providing an immersive experience to users during movie watching. Here, a girl in the movie is smelling a rose with the decreasing distance between the flower and her while the watcher could smell a more and more intense rose odor by wearing the olfaction interface. Here, OCL demonstrates the odor concentration level ranging from 1 to 3, corresponding to the low, middle, and high levels (iv). **b** A demonstration of the face-mask based olfaction interface in providing an alternative communication method, smell messages, for the disabled without the capability of vision and audio. By training users to link odors to specific information, the disabled users could efficiently communicate with others. **c** A demonstration of the Device 2 in assisting users to control their emotions as some specific odors could recall different human emotions. Here, all volunteers involved in the

emotion reaction test are verified in normal emotion range (**cii and ciii**). **d** A potential demonstration of the Device 2 for helping amnesic patients recall lost memories as odors perception is modulated by experience, leading to the recall of emotional memories. **e** Demonstration of the wireless olfaction interfaces in real time interaction between the user and a virtual subject for walking in a virtual garden surrounded by various fruit fragrance. By programming the Device 2, 9 OGs inside the device could work independently (**eii**), or work together (**eiii**).

Comment 18: Just the temperature of the device is not enough. Gas sensor response or sensory test result should be shown.

Our response: We thank the referee for this useful comment. We have conducted a series of tests to investigate odor generation performance by OGs, as shown in Fig. 4, Extended Data Fig. 4, and Supplementary Figs. 11, 13, 19, 20. For ethanol concentration measurement in Extended Data Fig. 4, we investigate three potential effects on ethanol generation performance of OGs with paraffin/ethanol embedded, including the heating temperature of OGs (Extended Data Fig. 4d), ambient wind speed (Extended Data Fig. 4e), and distance between working OGs and the ethanol sensor (Extended Data Fig. 4f), where all the tests were performed in an open box for minimizing the unpredictable ambient wind around working OGs. It is found that increasing heating temperature of OGs could contribute higher ethanol concentration, higher ambient wind could induce faster ethanol recovery time (RT), and larger distance between OGs and the ethanol sensor will take a longer time to detect the gas, meaning a longer delay time. We also measured the remaining time of ethanol in Device 2 after the Device 2 releasing ethanol for 5 mins, as shown in Extended Data Figs. 4g, h, i, j. Here, we setup two test conditions: Condition one is that the experimenter starts wearing the Device 2 after it releasing ethanol for 5 mins. Condition two is that the Device 2 keeps static in the open box after releasing ethanol for 5 mins. It is obvious that Condition one takes a shorter recovery time than Condition two, which is induced by the human breathing. Extended Data Fig. 4 shows the electrical response of the ethanol sensor embedded in the Device 2 as the experimenter is wearing it, where the heating temperature of the working OG in Device 2 is target at 60 °C. It is found that human breathing could decrease the ethanol concentration around the sensor with obvious concentration fluctuation (± 180 ppm). In addition to ethanol concentration measurement, we also conducted a series of volunteer sensory tests, as shown in Fig. 4, Supplementary Figs. 11, 13, 19, 20. The reason why we conducted these volunteer sensory tests is that we couldn't find specific commercial odor sensors to measure perfumes-based odor concentrations generated by OGs. All volunteer tests call for 11 volunteers with 4 males and 7 females. The volunteer test shown in Supplementary Fig. 11 is to investigate if volunteers could obviously sense the enhanced odor concentrations for 9 different odor types (see details in Characterization) generated by Device 2 when the heating temperature of OGs are increased from 45 °C to 50 °C, 55 °C, and 60 °C step by step. The high average recognition rate, 0.93, demonstrates that all volunteers confirm to sense the odor concentration variations by Device 2. We also measured the odor remaining time in Device 2 after the device release odors for 5 mins, where we setup two testing conditions. Condition One is that volunteers continuously smell the odors when wearing the Device 2. Condition Two is that the volunteers come to smell the odors of the face mask every 10 s. Then, we recorded the time for 9 different odors in the two testing conditions when the volunteers couldn't sense the corresponding odors, with the result shown in Supplementary Fig. 13. Supplementary Fig. 19 is another one volunteer sensory test, focusing

on recording the human reaction to the fast odor concentration variation realized by the heating temperature variation of OGs in Device 1. During the test, the Device 1 will be programmed to increase temperatures of OGs from 45 °C to 50 °C at 5 s, from 50 °C to 55 °C at 15 s, from 55 °C to 60 °C at 25 s, then lasting 1 min until shutting down Device 1. Once the volunteers can sense the enhanced odor concentration, the time point will be recorded meanwhile. Two different odor types for two OGs in Device 1 are adopted, including lavender and lilac. Supplementary Fig. 20 shows a new volunteer sensory test for testing if volunteers could sense the odors by OGs at room temperature, and it is found that each volunteers have distinguished odor concentration thresholds to various odor types with the recognition rate ranging from 0.22 to 0.89. As a result, volunteers can obviously sense some specific odor types, such as lavender, strawberry, and lilac, but the possibility in sensing minty, orange, green tea, lemon, and peach is low, ranging from 0.27 to 0.09. Finally, we collected all volunteers' emotion reactions to 30 different odors generated by Device 2 (see details in Characterization), as shown in Fig. 4c. It is found that the possibility of being joyful for these volunteers could reach up to 56%, 65%, 44%, and 56% when they smelled the grape, peach, orange, and strawberry odor, respectively. In addition, it is obvious that different users have distinguished emotion reactions to the 30 odors (Figs. 4cv, vi), which means that users could select their preferred odor types integrated insides Device 2 for emotion soothing. Finally, we reorganize the application data in Fig. 4. We have modified the text on Page 10 Line 22.

Modifications: On Page 10 Line 22, the text has been modified as “**Extended Data Fig. 4** shows the gaseous ethanol generation performance of the OGs. To minimize the unpredictable ambient wind effect on the performance of the OG, the whole experimental setup is placed insides an open box, where a commercial ethanol sensor is fixed 1 cm above a working OG, continuously monitoring the nearby ethanol concentration by reading the voltage variation of a 5 k Ω resistor connected in series (**Extended Data Figs. 4a, b, c**). By switching the heating temperature of the OG from 45 °C to 50 °C, to 55 °C, and finally to 60 °C (**Supplementary Fig. 10**), the ambient ethanol concentrations could reach up to 531 ppm, 2821 ppm, and 4531 ppm, respectively, where the corresponding response times in raising up and recovering are 4 s @ 40 s (recovery time, RT), 7 s @ 72 s, and 8 s @ 129 s (**Extended Data Fig. 4d**). Since the human smell threshold for ethanol concentration is 80 ppm (33), it is obvious that higher heating temperature could contribute stronger olfactory feedback. In addition to the ethanol concentration measurement, we also conducted a human sensory test to investigate the heating temperature effect on the generated odor concentration, as shown in **Supplementary Fig. 11**. During the testing, all volunteers were wearing the Device 2 with 9 working OGs integrated for testing their responses to 9 different odors, where the temperatures of each OG was increased from 45 °C to 50 °C, 55 °C, and 60 °C with the lasting time of 1 min for each temperature point. By summarizing all volunteers' response, we found that the recognition rates to these smells range from 0.73 to 1 with the average value of 0.93, further proving that the heating temperature of OGs is key for recognize odor concentration, and the heating temperatures ranging from 45 °C to 60 °C are sufficient to generate odors with the concentrations much larger than the human thresholds. **Extended Data Fig. 4e** shows the recovery time of generated ethanol concentration in air as a function of parallelly blown wind speed, ranging from 0 to 6.61 m/s, corresponding to a decreasing RT from 129 s to 9 s, which is induced by the increasing gas diffusion rate triggered by ambient wind. **Extended Data Fig. 4f** presents the electrical response of the commercial ethanol sensor as a function of the distance between the sensor and a working OG, and it is clear that longer distance could result in a higher delay time to trigger the sensor, including 1 cm @ 1.2 s, 3 cm @ 9.1 s, and 5 cm @ 15.6 s, which is further verified by the

volunteer sensory test shown in **Supplementary Fig. 12**. To further investigate the odor generating performance of OGs insides Device 2, the ethanol sensor is embedded in Device 2 near to the OG integrating paraffin/ethanol (**Extended Data Figs. 4g, h**). Here, two different testing conditions are introduced: Condition one is that Device 2 releases ethanol for 5 mins, then shut down meanwhile an experimenter starts wearing the Device 2 until the monitored ethanol concentration recovers to original state (lower than 80 ppm), as shown in **Extended Data Figs. 4i, j**; Condition two is that the Device 2 releases ethanol for 5 mins, then shut down without further motion (**Extended Data Figs. 4i, j**), where it is obvious that human breathing could shorten ethanol recovery time (1.2 min @ Condition one and 3.1 min @ Condition two). To further investigate human breathing effect on generated odor recovery time, we conducted a volunteer test (**Supplementary Fig. 13**), that also associates with two testing conditions same as that in **Extended Data Figs. 4i, j**. Here, 9 different odors are adopted including lavender (a), orange (b), pineapple (c), green tea (d), lemon (e), peach (f), strawberry (g), minty (h), and lilac (i). The average recovery time of Condition one is 92.1 s, larger than 77.1 s of Condition two, consistent with the conclusion shown in **Extended Data Figs. 4i, j**. As a result, it is obvious that the formerly generated odors will not interface with the newly generating odors after the corresponding odor recovery time, which is influenced by odor types and ambient airflow rate (**Extended Data Fig. 4 and Supplementary Fig. 13**). **Extended Data Figs. 4k, l** shows ethanol concentration generated by Device 2 as an experimenter is wearing the device, where the heating temperature of the working OG is target at 60 °C. Continuous breathing could obviously decrease the ethanol concentration to 1338 ppm around the ethanol sensor with a large fluctuation (± 180 ppm), which also demonstrates that the ethanol generation rate at the heating temperature of 60 °C is larger than human odor inhalation rate. As the generated odor concentration can be adjusted (**Extended Data Fig. 4d and Supplementary Fig. 11**), it is possible to avoid odor accumulation around human nose by adjusting the heating temperature of corresponding OGs in the Device 2.”

On Page 14 Line 22, the text has been modified as “**Fig. 4a** and **Movie S2** show a representative application of the skin-integrated olfaction interface for providing an immersive 4D movie watching experience (**Fig. 4ai**). Here, the subject in the movie is smelling a flower as it is approaching to her step by step while the OGs heating temperature simultaneously increases with the decreasing of the distance between the odor source and the subject nose, releasing a growing fragrance of roses for users in real world (**Figs. 4aii and 4aiii**). In the beginning of the movie as the flower is far away from the subject, the two OGs are in a state of stand by (heating temperature, 45 °C), then the heating temperatures start rising step by step from 45 °C, to 50 °C@5 s, to 55 °C@15 s, to 60 °C@25 s, then back to 45 °C@28 s (**Fig. 4aiv**). Here, we conducted a volunteer sensory test to investigate if users could react to fast temperature variation of OGs integrated in Device 1 (**Supplementary Fig. 19**). Following the time points in increasing the heating temperature of OGs shown in **Fig. 4aiii and iv**, all volunteers’ reaction times to variating heating temperature of two OGs in the Device 1 are recorded in **Fig. 4aiv** and **Supplementary Fig. 19**, including 9.7 s (reaction time) @ 5 s (time point in increasing heating temperature), 20.8 s @ 15 s, and 28.4 s @ 25 s. As a result, our Device 1 could generate distinguished odor concentration to users, further demonstrating the great potential in movie watching. **Supplementary Fig. 20** shows a volunteer sensory test result in judging presence or absence of odors in silent OGs at room temperature. It is found that the recognition rates range from 0.22 to 0.89 with an average value of 0.43 for all volunteers, which demonstrates that human beings have distinguished odor concentration thresholds. In addition, different odor types also have distinguished recognition rate, ranging from 0.09 to 0.82.”

Extended Data Fig. 4. Ethanol concentration performance of the odor generators. **a, b, c** Ethanol concentration testing setup, where the test is placed inside an open box for minimizing the ambient wind interference to OGs. During the testing, a commercial resistance-variation-based ethanol sensor placed 1 cm above a working OG for low response time to generated ethanol around the OG. **d** Ethanol concentration generated by OG with 30 mg paraffin/ethanol (mass ratio, 10:3) with a controlled temperature switching between 45°C to 50°C, 55°C, and 60°C, respectively, corresponding to the increasing ethanol concentration peak values. **e** Ambient wind effects on the ethanol concentration dissipation rate, here an operating OG with a heating temperature of 60°C are suddenly shut down to cut off ethanol generation, and the wind is blown above the OG parallelly. **f** Delay time of a working OG as a function of the distance between the generator and ethanol sensor, which is induced by the ethanol diffusion rate. **g, h, i, j, k, l** Ethanol remaining time test in Device 2 with three testing conditions. Condition one is that experimenter continuously smell the generated ethanol after the OG with

paraffin/ethanol embedded works for 5 min at a constant heating temperature of 60°C when he is wearing the Device 2, and Condition two is that the ethanol sensor monitors the ethanol concentration after the OG works for 5 min in an open box (i, j). Condition three is that the ethanol sensor continuously monitors gas concentration as the experimenter is wearing the Device 2 (k, l).

Fig. S11. A volunteer testing showing the volunteers' odor recognition rate when they are wearing the Device 2 with 9 different perfume-based odors (see details in Characterization). Here, the temperature of each OGs will be increased from 45 °C to 60 °C at an interval of 5 °C, where each temperature will last 1 min before going up. The, the volunteers will be asked if they could sense the enhanced odor concentration.

Fig. S13. A volunteer testing showing the odor remaining time in Device 2. Here, all volunteers will be asked to smell the remaining odors in the Device 2, and record the time when the odors disappear. There are two testing conditions. Condition One is that volunteers continuously smell the odors when wearing the Device 2. Condition Two is that the volunteers come to smell the odors of the face mask every 10 s. Before volunteer smells the Device 2, the OG will generate the odor at a constant heating temperature of 60 °C in the face mask for 5 min.

Fig. S19. A volunteer testing showing the human reaction to the fast temperature variation of OGs with two odor types embedded when they are wearing Device 1 for testing. During the

test, the Device 1 will be programmed to increase temperatures of OGs from 45 °C to 50 °C at 5 s, from 50 °C to 55°C at 15 s, from 55 °C to 60 °C at 25 s, then lasting 1 min until shutting down Device 1. Once the volunteers can sense the enhanced odor concentration, the time point will be recorded meanwhile. Two different odor types for two OGs in Device 1 are adopted, including lavender and lilac.

Fig. S20. A volunteer testing showing the recognition rate of volunteers in smelling the OGs at room temperature when they are the Device 1. Here, a, b, c, d, e, f, g, h, and i stand for lavender, orange, pineapple, green tea, lemon, peach, strawberry, minty, and lilac.

Comment 19: Table S1 seems insufficient. OG part includes only a few reports and does not cover most of related works. On the other hand, the survey of odor sensor is not necessary.

Our response: We thank the referee for this useful comment. We have modified the Supplementary Table 1 with more papers related to odor generators, and also deleted some unnecessary papers in odor sensing.

Modifications: On Page 3 Line 12, the text has been modified as “The current olfaction generating technologies mainly associate either with big instrument to generate smell in a closed area/room or in-built bulky VR set, where the wired connection, large demission, dull smell generating function and low response time make these olfaction feedback methods far behind the development of visual/auditory based VR devices, and thus dramatically limit their application fields (13, 14). Distinguished with these smell generating/releasing systems, odor generators (OGs) in wearable formats enable to create a personalized and localized odorous environment, avoiding the odors interference and long response time of smells switching. To date, there are some reports of wearable OGs with the working principle mainly based on commercial liquid atomizers to atomize perfume into tiny droplets for later blowing out (15). While the clumsy working mechanics, fully rigid package with bulky bottles of liquid perfumes, and special maintenance requirements have indicated the inherent defects of these wearable OGs for realizing high-channel odor generators in a miniaturized, lightweight, and flexible format (**Supplementary Table 1**) (13, 14, 16-27). In addition, a paired intelligent electronic control panel is also essential for tele-operating the OGs with numerous selective odor types according to users’ requirements (28-32).”

Table S1. Comparison of the state-of-art olfactory interfaces with our work in the aspect of type, functional module, dimensions, scents, mechanical formats, communication method, and applications.

Type	Functional Module	Dimensions (mm)	Odor options	Mechanical formats	Communication method	Application	Ref.
Odor sensor	rGO/Mesoporous ZnO NSs Hybrid fibers	L=30 D=0.012	1	Flexible, wearable	N/A	high-performance e-textiles	(16)
	PANI/FMWCNT nanocomposite film	N/A	1	Flexible, transparent	N/A	environmental contaminant level or human physiological status	(17)
	wireless dosimetric chemical hazard badge	N/A	1	flexible, wearable	NFC	detection of nerve-agent	(18)
	PANi/ Au electrode tattoo	N/A	1	flexible, wearable	Bluetooth	detection of nerve-agent	(19)
	PEDOT:PSS/ Au interdigital electrodes	L=8 W=0.3	1	flexible, wearable	Bluetooth	detection of food spoilage	(20)
	SWCNT/ZnO transistors	N/A	1	flexible, wearable	Wired	identification of explosives	(21)
Odor generator	a surface acoustic wave device and micro-dispensing valves	N/A	1	rigid	Wired USB	VR game	(13)
	mass flow controllers	N/A	3	rigid	Wired	olfactory virtual reality for mice	(14)
	Commercial atomizer	N/A	3	rigid	Bluetooth	Improving users' sleep quality	(22)
	Commercial atomizer	L=110 W=56 T=35	1	rigid	Wi-Fi	VR game	(23)
	BFM based atomizer	N/A	4	rigid	Wired	VR technology	(24)
	Commercial atomizer	N/A	1	rigid	Wired	VR game	(25)
	Commercial atomizer	N/A	1	rigid	Wired	Olfaction display	(26)
	Physical phase change of odorous paraffin wax	N/A	6	rigid and non-wearable	Bluetooth	Olfaction training	(27)
	Physical phase change of odorous paraffin wax by controlling the heating temperature.	L=18 W=16 T=3	9 basic types	flexible, wearable	Bluetooth	VR/AR, human-machine interface, medical treatment, 4D movie, and message delivery.	this work

The newly added reference

22. J. Amores, M. Dotan, P. Maes, *Development and Study of Ezzence: A Modular Scent Wearable to Improve Wellbeing in Home Sleep Environments. Frontiers in psychology*, 550 (2022).
23. M. de Paiva Guimarães, J. M. Martins, D. R. C. Dias, R. d. F. R. Guimarães, B. B. Gnecco, *An olfactory display for virtual reality glasses. Multimedia Systems*, 1-11 (2022).
24. P. Yang et al., *Self-powered virtual olfactory generation system based on bionic fibrous membrane and electrostatic field accelerated evaporation. EcoMat*, e12298 (2022).
25. S. Kato, T. Nakamoto, in *2019 IEEE International Symposium on Olfaction and Electronic Nose (ISOEN)*. (IEEE, 2019), pp. 1-3.
26. Y. Wang, J. Amores, P. Maes, in *Proceedings of the 2020 CHI Conference on Human Factors in Computing Systems*. (2020), pp. 1-9.
27. A. Tiele, S. Menon, J. A. Covington, *Development of a Thermal-Based Olfactory Display for Aroma Sensory Training. IEEE Sensors Journal* 20, 631-636 (2019)

Responses to comments of Referee #3

Comments from Referee #3:

Summary Comment: This paper presents skin-interfaced olfactory feedback systems with wirelessly programmable capabilities based on arrays of flexible and miniaturized odor generators (OGs) for olfactory VR applications. The authors insist on significance in that the olfactory display proposed here can establish a bridge between the electronics and users for broad application fields ranging from immersive VR experiences because they are wireless and wearable format, showing two types of devices: skin-mountable form and glasses form. However, while they mentioned only two existing related papers where olfaction display systems mainly associate with a large instrument to generate smell in a closed area or an in-built bulky VR set, they expressed it as if this were the only paper to solve this problem. According to the prior study, wireless and portable olfactory display (OD) system has already been reported¹. Furthermore, the development of wireless OD masks for the VR experience is not new^{2, 3, 4}. And the miniaturized and skin-mountable form factors of the OD were explored⁵. However, this manuscript does not include and mention these prior articles that are directly related to the presented works. Also, the odor generator employing the heater and paraffin wax was reported earlier⁶. Thus, this odor generation system in this paper is not novel.

Considering all points mentioned above, the novelty and significance of this paper are very limited, making this work unacceptable to be published in Nature communication.

Similar previous works that were not cited in this paper:
1. Amores J, Dotan M, Maes P. Development and Study of Ezzence: A Modular Scent Wearable to Improve Wellbeing in Home Sleep Environments. *Frontiers in psychology*, 550 (2022).

2. de Paiva Guimarães M, Martins JM, Dias DRC, Guimarães RdFR, Gnecco BB. An olfactory display for virtual reality glasses. *Multimedia Systems*, 1-11 (2022).

3. Yang P, et al. Self - powered virtual olfactory generation system based on bionic fibrous membrane and electrostatic field accelerated evaporation. *EcoMat*, e12298 (2022).

4. Kato S, Nakamoto T. Wearable olfactory display with less residual odor. In: 2019 IEEE International Symposium on Olfaction and Electronic Nose (ISOEN)). IEEE (2019).

5. Wang Y, Amores J, Maes P. On-face olfactory interfaces. In: Proceedings of the 2020 CHI Conference on Human Factors in Computing Systems) (2020).

6. Tiele A, Menon S, Covington JA. Development of a Thermal-Based Olfactory Display for Aroma Sensory Training. *IEEE Sensors Journal* 20, 631-636 (2019).

Our response: We thank the referee for this useful comment. We have added the mentioned published articles in Supplementary Table 1. After carefully reviewing these published paper, these reported olfaction displays are distinguished with ours. We will give the detailed illustration as following:

For the first paper (*Frontiers in psychology*, 550, 2022), the olfaction interface, called Ezzence, is built on rigid platform with only three odor options, and its odor releasing mechanism is based on a commercial piezoelectric module based atomizer with microholes insides. Although this device could be worn around the users' chest, the distance between the device and human nose is far, which may result in longer odor transmission time to human nose and larger odor diffusion areas. The application area of Ezzence focuses on improving person's sleep quality. By contrast, our olfaction system support two wearable formats, including skin-integrated Device 1 and face mask based Device 2, and the short distance between our devices and human nose enables fast smell feedback (~1.4 s, Fig. 2f) and small odor diffusion area (only around users' nose). 9 different odor options in Device 2 further extend the application areas ranging from 4D movie watching, to smell message delivery, to medical treatment, to human emotion control, and to VR/AR. In addition, we have conducted a comprehensive electrical performance of our olfaction system, including the heating temperature controlling, odor concentration measurement, volunteer sensory tests, device long-term stability, etc, but we can't find any data to exhibit the electrical performance of Ezzence in this paper. Therefore, we don't think Ezzence reported in this paper is comparable to ours in terms of flexibility, user-friendliness, odor options number, application areas, odor concentration accuracy controlling, long-term stability, and odor variation response time.

For the second paper (*Multimedia Systems*, 1-11, 2022), this paper reported an olfactory display (OD) integrating with VR glasses for VR application. The working principle of this device is based on a commercial ultrasonic atomizer, same to the first paper (*Frontiers in psychology*, 550, 2022), and the size of the olfaction display control panel can reach up to 110 mm × 56 mm × 35 mm with a weight of 239 g, much larger than both our devices (Device 1, 7.4 g; Device 2, 26.3 g), meaning that the wearability and user-friendliness of the OD is worrying. In addition, the bulky OD only support one odor option, which further limiting its application in VR. Finally, we couldn't find any data to describe the performance of the OD in the paper, which is also not comparable to our olfaction interfaces.

For the third paper (EcoMat, e12298, 2022), this paper reported a virtual olfactory system based on liquid atomization technology. By combining a large switching module, a mechanical brushless motor, and bulky TENG module, the rigid olfaction system is too big to wear, as shown in Figs. S14, S26 of this paper, and it only supports 4 odor options. In addition, the authors could not find any figures or data to prove that this device can be used in VR application. At last, the paper doesn't provide any technical data and experiments to investigate the odor generation performance of the olfaction system. Compared to this device, our olfaction system is more user-friendly, lightweight, and excellent odor generation capability with sufficient data supported. In addition, low response time, long-term operation stability, excellent flexibility, and multiple odor options of our devices enable the olfaction systems a wide variety of applications, including 4D movie watching, smell message delivery, medical treatment, human emotion control and VR/AR based online teaching with supporting data, figures, and videos.

For the fourth paper (In: 2019 IEEE International Symposium on Olfaction and Electronic Nose (ISOEN)). IEEE, 2019), a rigid olfactory display is reported with the working principle of the regular atomization technology. The device could only support one odor option but a big size, as shown in Fig. 5 of the paper. In addition, there is no any data illustrating the performance of the device throughout the paper. The bulky size, one odor option, one application field (VR), and regular working principle distinguish this device with ours.

For the fifth paper (In: Proceedings of the 2020 CHI Conference on Human Factors in Computing Systems, 2020), a series of wearable olfaction interfaces are reported based on the atomization technology. The rigid but small wearables support one odor option. There is no any data illustrating the performance of the device throughout the paper. One odor option, no performance assessment, and regular working principle distinguish this device with ours.

For the sixth paper (*IEEE Sensors Journal* 20, 631-636, 2019), the olfaction display system is a bulky equipment which cannot be worn. The system supports 6 odor options during utilization, which is mainly for olfaction training. Except the fact that the working principle of this equipment is similar to ours, this equipment and our olfaction systems are completely different in all aspects, including wearability, overall size, circuit design, odor generator design, application areas, and integrated odor generator number. In addition, to achieve the rapid heating temperature variation, we design a miniaturized active cooling system to cooperate with the heating electrode. As a result, the odor generator could extremely fast decrease heating temperature of OGs from 60 to 45 in 1.5 s, which can not be found in this paper. As a result, the olfaction system reported in this paper is largely distinguished with ours.

In summary, the above six papers are all distinguished with ours in terms of working principle, flexibility, user-friendliness, odor options number, odor generation performance, long-term stability, data comprehension, and application areas. We have added these relevant papers in Supplementary Table 1, and made modification on Page 3 Line 12.

Modifications: On Page 3 Line 12, the text has been modified as “The current olfaction generating technologies mainly associate either with big instrument to generate smell in a closed area/room or in-built bulky VR set, where the wired connection, large demission, dull smell generating function and low response time make these olfaction feedback methods far behind the development of visual/auditory based VR devices, and thus dramatically limit their application fields (13, 14). Distinguished with these smell generating/releasing systems, odor generators (OGs) in wearable formats enable to create a personalized and localized odorous environment, avoiding the odors interference and long response time of smells switching. To date, there are some reports of wearable OGs with the working principle mainly based on commercial liquid atomizers to atomize perfume into tiny droplets for later blowing out (15). While the clumsy working mechanics, fully rigid package with bulky bottles of liquid perfumes, and special maintenance requirements have indicated the inherent defects of these wearable OGs for realizing high-channel odor generators in a miniaturized, lightweight, and flexible format (**Supplementary Table 1**) (13, 14, 16-27). In addition, a paired intelligent electronic control panel is also essential for tele-operating the OGs with numerous selective odor types according to users’ requirements (28-32).”

Table S1. Comparison of the state-of-art olfactory interfaces with our work in the aspect of type, functional module, dimensions, scents, mechanical formats, communication method, and applications.

Type	Functional Module	Dimensions (mm)	Odor options	Mechanical formats	Communication method	Application	Ref.
	rGO/Mesoporous ZnO NSs Hybrid fibers	L=30 D=0.012	1	Flexible, wearable	N/A	high-performance e-textiles	(16)

Odor sensor	PANI/FMWCNT nanocomposite film	N/A	1	Flexible, transparent	N/A	environmental contaminant level or human physiological status	(17)
	wireless dosimetric chemical hazard badge	N/A	1	flexible, wearable	NFC	detection of nerve-agent	(18)
	PANi/ Au electrode tattoo	N/A	1	flexible, wearable	Bluetooth	detection of nerve-agent	(19)
	PEDOT:PSS/ Au interdigital electrodes	L=8 W=0.3	1	flexible, wearable	Bluetooth	detection of food spoilage	(20)
	SWCNT/ZnO transistors	N/A	1	flexible, wearable	Wired	identification of explosives	(21)
Odor generator	a surface acoustic wave device and micro-dispensing valves	N/A	1	rigid	Wired USB	VR game	(13)
	mass flow controllers	N/A	3	rigid	Wired	olfactory virtual reality for mice	(14)
	Commercial atomizer	N/A	3	rigid	Bluetooth	Improving users' sleep quality	(22)
	Commercial atomizer	L=110 W=56 T=35	1	rigid	Wi-Fi	VR game	(23)
	BFM based atomizer	N/A	4	rigid	Wired	VR technology	(24)
	Commercial atomizer	N/A	1	rigid	Wired	VR game	(25)
	Commercial atomizer	N/A	1	rigid	Wired	Olfaction display	(26)
	Physical phase change of odorous paraffin wax	N/A	6	rigid and non-wearable	Bluetooth	Olfaction training	(27)
Physical phase change of odorous paraffin wax by controlling the heating temperature.	L=18 W=16 T=3	9 basic types	flexible, wearable	Bluetooth	VR/AR, human-machine interface, medical treatment, 4D movie, and message delivery.	this work	

The newly added reference:

22. J. Amores, M. Dotan, P. Maes, *Development and Study of Ezzence: A Modular Scent Wearable to Improve Wellbeing in Home Sleep Environments. Frontiers in psychology*, 550 (2022).
23. M. de Paiva Guimarães, J. M. Martins, D. R. C. Dias, R. d. F. R. Guimarães, B. B. Gnecco, *An olfactory display for virtual reality glasses. Multimedia Systems*, 1-11 (2022).
24. P. Yang et al., *Self-powered virtual olfactory generation system based on bionic fibrous membrane and electrostatic field accelerated evaporation. EcoMat*, e12298 (2022).
25. S. Kato, T. Nakamoto, in *2019 IEEE International Symposium on Olfaction and Electronic Nose (ISOEN). (IEEE, 2019)*, pp. 1-3.
26. Y. Wang, J. Amores, P. Maes, in *Proceedings of the 2020 CHI Conference on Human Factors in Computing Systems. (2020)*, pp. 1-9.
27. A. Tiele, S. Menon, J. A. Covington, *Development of a Thermal-Based Olfactory Display for Aroma Sensory Training. IEEE Sensors Journal* 20, 631-636 (2019)

Comment 1: For clarification, this manuscript should add and compare the device's performance in the previous articles mentioned above in table S1. It could be better to search with the keyword 'olfactory display.'

Our response: We thank the referee for this useful comment. We have added the previous

articles in Supplementary Table 1. Actually, the olfaction displays reported in the above published articles are distinguished with ours in terms of odor generators array density, flexibility, weight, response time, and odor generators number.

For the OGs array density, our olfaction systems provide two wearable formats, including the skin-integrated Device 1 (OGs array density, $0.79/\text{cm}^3$) and face mask based Device 2 ($0.88/\text{cm}^3$). The OGs array density values for the listed papers above are shown here, including $0.12/\text{cm}^3$ @ the first paper (*Frontiers in psychology*, 550, 2022), $0.005/\text{cm}^3$ @ the second paper (*Multimedia Systems*, 1-11, 2022), $0.014/\text{cm}^3$ (this conservative value is estimated from the Fig. S15 in this paper) @ the third paper (*EcoMat*, e12298, 2022), not available value (a bulky VR glasses supplement with only one odor option) @ the fourth paper (*In: 2019 IEEE International Symposium on Olfaction and Electronic Nose (ISOEN)*). *IEEE*, 2019), $0.05/\text{cm}^3$ (this conservative value is estimated from the Fig. 1 in this paper, and I don't take the rigid and big controlling circuit into account) @ the fifth paper (*In: Proceedings of the 2020 CHI Conference on Human Factors in Computing Systems*, 2020), and $0.005/\text{cm}^3$ @ the sixth paper (*IEEE Sensors Journal* 20, 631-636, 2019). It is obvious that our two olfaction systems both exhibit higher OGs array density over the six reported olfaction systems.

For flexibility, our Device 1 is a skin-integrated electronics, capable of tightly mounting onto curved human skin, as shown Fig. 1d and Supplementary Fig. 17. Various mouth motions will not separate the device with the interfaced skin, as shown in Supplementary Fig. 17. Our Device 2 is built up on a flexible face mask based on commercial flexible TPU (a 3D printing material), and the control panel is also based on flexible printed circuit board (FPCB) with PDMS encapsulated, which results in a fact that Device 2 could fit different users' face shape (Fig. 1e and Supplementary Fig. 7). By contrast, the olfaction systems reported in the six papers mentioned above are built on a rigid platform, exhibiting no flexibility.

For weight, the masses of our Device 1 and Device 2 are 7.4 g and 26.3 g, respectively, and the weight of one OG is 1.8 g. The weights of the olfaction systems reported in the mentioned papers are not available value @ the first paper, 239 g @ the second paper, not available value (containing a bulky control circuit and one brushless motor) @ the third paper, not available value (containing 15.9 g carbon as the filter) @ the fourth paper, not available value (containing an external circuit) @ the fifth paper, and 1700 g @ the sixth paper. According to the reported weights of these olfaction systems, both our two systems exhibit an obvious advantage in weight, which is more friendly to users during long-term applications.

For response time, only our paper and the sixth paper (*IEEE Sensors Journal* 20, 631-636, 2019) reported a response time in generating odors with the results of 1.4 s and 20 s, respectively. For the response time in decreasing heating temperature from $50\text{ }^\circ\text{C}$ to $45\text{ }^\circ\text{C}$, the result is only provided by our OG, $\sim 0.8\text{ s}$, due to the active cooling system, as shown in Fig. 2f. Therefore, our OG has a smaller response time than that reported in the sixth paper.

For OG numbers integrated in olfaction system, our Device 2 could integrate 9 OGs for independently operation, as shown in Fig. 4e. The maximum OGs number for these published papers are 3 @ the first paper, 1 @ the second paper, 4 @ the third paper, 1 @ the fourth paper, 1 @ the fifth paper, and 6 @ the sixth paper. As a result, our Device 2 could support the most OGs integrated among these olfaction systems.

In summary, our olfaction systems exhibit obvious advantages over the reported olfaction systems in terms of odor generators array density, flexibility, weight, response time, and odor generators number. We have put the comment on Page 3 Line 12.

Modifications: On Page 3 Line 12, the text has been modified as “The current olfaction generating technologies mainly associate either with big instrument to generate smell in a closed area/room or in-built bulky VR set, where the wired connection, large demission, dull smell generating function and low response time make these olfaction feedback methods far behind the development of visual/auditory based VR devices, and thus dramatically limit their application fields (13, 14). Distinguished with these smell generating/releasing systems, odor generators (OGs) in wearable formats enable to create a personalized and localized odorous environment, avoiding the odors interference and long response time of smells switching. To date, there are some reports of wearable OGs with the working principle mainly based on commercial liquid atomizers to atomize perfume into tiny droplets for later blowing out (15). While the clumsy working mechanics, fully rigid package with bulky bottles of liquid perfumes, and special maintenance requirements have indicated the inherent defects of these wearable OGs for realizing high-channel odor generators in a miniaturized, lightweight, and flexible format (**Supplementary Table 1**) (13, 14, 16-27). In addition, a paired intelligent electronic control panel is also essential for tele-operating the OGs with numerous selective odor types according to users’ requirements (28-32).”

Table S1. Comparison of the state-of-art olfactory interfaces with our work in the aspect of type, functional module, dimensions, scents, mechanical formats, communication method, and applications.

Type	Functional Module	Dimensions (mm)	Odor options	Mechanical formats	Communication method	Application	Ref.
Odor sensor	rGO/Mesoporous ZnO NSs Hybrid fibers	L=30 D=0.012	1	Flexible, wearable	N/A	high-performance e-textiles	(16)
	PANI/FMWCNT nanocomposite film	N/A	1	Flexible, transparent	N/A	environmental contaminant level or human physiological status	(17)
	wireless dosimetric chemical hazard badge	N/A	1	flexible, wearable	NFC	detection of nerve-agent	(18)
	PANI/ Au electrode tattoo	N/A	1	flexible, wearable	Bluetooth	detection of nerve-agent	(19)
	PEDOT:PSS/ Au interdigital electrodes	L=8 W=0.3	1	flexible, wearable	Bluetooth	detection of food spoilage	(20)
	SWCNT/ZnO transistors	N/A	1	flexible, wearable	Wired	identification of explosives	(21)
Odor generator	a surface acoustic wave device and micro-dispensing valves	N/A	1	rigid	Wired USB	VR game	(13)
	mass flow controllers	N/A	3	rigid	Wired	olfactory virtual reality for mice	(14)
	Commercial atomizer	N/A	3	rigid	Bluetooth	Improving users’ sleep quality	(22)
	Commercial atomizer	L=110 W=56 T=35	1	rigid	Wi-Fi	VR game	(23)
	BFM based atomizer	N/A	4	rigid	Wired	VR technology	(24)
	Commercial atomizer	N/A	1	rigid	Wired	VR game	(25)

	Commercial atomizer	N/A	1	rigid	Wired	Olfaction display	(26)
	Physical phase change of odorous paraffin wax	N/A	6	rigid and non-wearable	Bluetooth	Olfaction training	(27)
	Physical phase change of odorous paraffin wax by controlling the heating temperature.	L=18 W=16 T=3	9 basic types	flexible, wearable	Bluetooth	VR/AR, human-machine interface, medical treatment, 4D movie, and message delivery.	this work

The newly added reference:

22. *J. Amores, M. Dotan, P. Maes, Development and Study of Ezzence: A Modular Scent Wearable to Improve Wellbeing in Home Sleep Environments. Frontiers in psychology, 550 (2022).*
23. *M. de Paiva Guimarães, J. M. Martins, D. R. C. Dias, R. d. F. R. Guimarães, B. B. Gnecco, An olfactory display for virtual reality glasses. Multimedia Systems, 1-11 (2022).*
24. *P. Yang et al., Self-powered virtual olfactory generation system based on bionic fibrous membrane and electrostatic field accelerated evaporation. EcoMat, e12298 (2022).*
25. *S. Kato, T. Nakamoto, in 2019 IEEE International Symposium on Olfaction and Electronic Nose (ISOEN). (IEEE, 2019), pp. 1-3.*
26. *Y. Wang, J. Amores, P. Maes, in Proceedings of the 2020 CHI Conference on Human Factors in Computing Systems. (2020), pp. 1-9.*
27. *A. Tiele, S. Menon, J. A. Covington, Development of a Thermal-Based Olfactory Display for Aroma Sensory Training. IEEE Sensors Journal 20, 631-636 (2019)*

Comment 2: The author has emphasized the form factor of OD as the skin-interfaced system, although only two OG modules can be equipped to the format. Therefore, it is hard to understand the description below. As a result, to provide personalized odors in a small, localized area, new wearable olfaction interfacing technologies should exhibit advances as following: (a) the whole system should be built up on a soft substrate in a wearable or even skin-integrated format with miniaturized size and light weight; (b) as many odors with adjustable concentrations and long operation duration to support long term utilization without frequent replacement/maintenance. Could you explain why the OD should be directly mounted onto human skin in detail?

Our response: We thank the referee for this useful comment. Considering the generated odor variation response time and odor accumulation surrounding users, we think building up wearable olfaction systems, close to human nose, could effectively decrease odor response time with extremely short odor diffusion distance between odor sources and users' nose, and miniaturize the unnecessary odor accumulation around the users, which may influence people nearby.

From the response time point, the OD in this work shows obvious advantages. In revision, we have conducted a new experiment to quantize the delay time as a function of the distance between OGs and the odor sensor (or human nose), as shown in Extended Data Fig. 4f and Supplementary Fig.12. For the ethanol odor, the longer distance between the ethanol OG and ethanol sensor could induce a longer delay time for triggering the sensor with the results shown as: 1 cm @ 1.2 s, 3 cm @ 9.1 s, and 5 cm @ 15.6 s (Extended Data Fig. 4f). For perfume-based odor types, an experimenter is asked to smell the 9 different odors generated by OGs. Then, we recorded the delay time as a function of the distance between the experimenter's nose and OGs, with the results shown in Supplementary Fig. 12. Consistent with the result shown in

Extended Data Fig. 4f, the longer distance could result in a higher delay time for various odors (see odor types in Characterization).

From odor accumulation point, the longer distance between OGs and human nose demonstrates a lower odor concentration, as shown in Extended Data Fig. 4f, where the ethanol concentration generated by a same OG could induce three distinguished electrical response of the ethanol sensor as the ethanol concentration detected at 1 cm away from the OG is highest. With the increasing of the distance between OGs and human nose, more unnecessary odors will diffuse to the air surrounding the users, which may gradually accumulate to a high level for surrounding people. And, more odors are wasted, resulting in a shorter operation duration.

In summary, it is better to mount the olfaction system close to users' nose to achieve a low response time and small odor accumulation in the surroundings. To mount the olfaction system close to human nose, the skin-integrated Device 1 is mounted on the human upper lip, realizing a shortest distance between OGs and human nose, and the wearable Device 2 is in a face mask format, which enables the large OGs array to approach to the human nose, as shown in Figs. 1d, e, and Supplementary Figs. 7, 17. We have modified the text on Page 3 Line 24.

Modifications: On Page 3 Line 24, the text has been modified as “As a result, to provide personalized odors in a small localized area close to users' nose, new wearable olfaction interfacing technologies should exhibit advances as following: (a) the whole system should be built up on a soft substrate in a wearable or even skin-integrated format with miniaturized size and light weight; (b) as many odors with adjustable concentrations and long operation duration to support long term utilization without frequent replacement/maintenance; (c) the olfactory interface should support wireless and programmable operation, capable of interacting with users for various applications. (d) the odor sources should be easy-access and biocompatible. (e) rapid response time in bursting or suppressing odors and accurate odor concentration control are required for the olfaction system in VR/AR applications.”

On Page 5 Line 23, the text has been modified as “Instead, the face mask based Device 2 can integrate 9 different OGs together to realize the portfolio of hundreds odors stimulation by programing the activated numbers, heating temperature and operation time of the 9 OGs (**Fig. 1e and Extended Data Fig. 1**). Here, more odor options (9 OGs), higher OGs array density ($0.88/\text{cm}^3$), faster odor generation response time (1.44 s), and good flexibility enable wider application fields than the reported olfaction systems listed in **Supplementary Table 1.**”

Extended Data Fig. 4. Ethanol concentration performance of the odor generators. **a, b, c** Ethanol concentration testing setup, where the test is placed inside an open box for minimizing the ambient wind interference to OGs. During the testing, a commercial resistance-variation-based ethanol sensor placed 1 cm above a working OG for low response time to generated ethanol around the OG. **d** Ethanol concentration generated by OG with 30 mg paraffin/ethanol (mass ratio, 10:3) with a controlled temperature switching between 45°C to 50°C, 55°C, and 60°C, respectively, corresponding to the increasing ethanol concentration peak values. **e** Ambient wind effects on the ethanol concentration dissipation rate, here an operating OG with a heating temperature of 60°C are suddenly shut down to cut off ethanol generation, and the wind is blown above the OG parallelly. **f** Delay time of a working OG as a function of the distance between the generator and ethanol sensor, which is induced by the ethanol diffusion rate. **g, h, i, j, k, l** Ethanol remaining time test in Device 2 with three testing conditions. Condition one is that experimenter continuously smell the generated ethanol after the OG with paraffin/ethanol embedded works for 5 min at a constant heating temperature of 60°C when he

is wearing the Device 2, and Condition two is that the ethanol sensor monitors the ethanol concentration after the OG works for 5 min in an open box (i, j). Condition three is that the ethanol sensor continuously monitors gas concentration as the experimenter is wearing the Device 2 (k, l).

Fig. S12. Delay time as a function of the distance between OGs and user’s nose for 9 odor types. Here, a, b, c, d, e, f, g, h, and i stand for lavender, orange, pineapple, green tea, lemon, peach, strawberry, minty, and lilac.

Comment 3: Also, fabricated device 1 (skin-mounted device) in Figure 1D does not seem flexible enough to be conformally attached to curved human skin for a long time. Therefore, quantitative data support is needed to show the high flexibility of the device.

Our response: We thank the referee for this useful comment. We have conducted two new experiments to investigate the flexibility and stability of the Device 1, as shown in Supplementary Figs. 16, 17. To demonstrate the flexibility of the device, we fixed the Device 1 on a programmable bending platform with a controllable bending angles and frequency. During the continuous bending with a bending angle of 40° and a frequency of 0.33 Hz, the Device 1 continuously sends the voltage value of OG electromagnetic coils to a paired receiver wirelessly, as shown in Fig. 15. The stable voltage input (3 ± 0.02 V) proves the excellent stability and flexibility of the Device 1. To further demonstrate the flexibility of the Device 1, an experimenter is wearing the Device 1 with various mouth motions, as shown in Supplementary Fig. 17. The enlarged optical image details prove that the Device 1 could be conformally attached to human upper lip for stable operation. We have modified the text on Page 13 Line 15.

Modifications: On Page 13 Line 15, the text has been modified as “**Fig. 3e** and **Supplementary Figs. 14, 15** show two typical stability tests for providing vibration to a working OG with controllable vibration amplitude, frequency, and bending angles by adopting a commercial oscillator (**Supplementary Fig. 14**) and a self-developed programmable bending platform (**Supplementary Fig. 15**). As a result, millimeter-scaled vibration with an adjustable frequency ranging from 0 to 10 Hz doesn’t bring obvious interference on the electrical performance of the OG (**Figs. 2f, g**), but the wide angle and range of rotation could induce a slight temperature fluctuation of 1 °C to the OG (**Supplementary Fig. 15**), which is resulted from the acceleration of the heat dissipation during the wide range movement induced forced convection heat transfer between the OG and the surrounding air (22). However, this negligible fluctuation of temperature doesn’t cause obvious performance deterioration during operation. **Supplementary Fig. 16** demonstrates the high stability of Device 1, where the device is fixed

on a programmable bending platform for 2000 continuous bending cycles at the constant frequency and angle of 0.33 Hz and 40°. During the bending process, Device 1 wirelessly sends the voltage value into the electromagnet coils of OGs integrated to the paired receiver, and stable voltage input (3 ± 0.02 V) proves the excellent stability of Device 1. To demonstrate the flexibility of the Device 1, the device is mounted onto human upper lip with different mouth motions, and enlarged optical image details proves the tight attachment of the device onto human skin during various skin deformations (**Supplementary Fig. 17**).”

Fig. S16. Optical images and electrical response of the Device 1 at a bending angle of 40° for over 2000 cycles. Here, the electrical signal is the voltage into the electromagnetic coils of two OGs in Device 1, which is wirelessly read by a paired receiver. The stable voltage input demonstrates the normal operation of the Device 1 during continuous bending.

Fig. S17. Optical images of Device 1 mounted onto human upper lip. As users do various mouth motions, the Device 1 can be still tightly mounted onto the skin.

Comment 4: When the OD generates odors, the temperature of the heater could be over 50 °C. Could you ensure it does not cause damage to the skin where the device is mounted under continuous heating conditions?

Our response: We thank the referee for this useful comment. Obviously, the reviewer has a rich experimental experience. The heating temperature of OGs could reach up to 60 °C when operating, which raises up a safety issue during practical application. For the Device 1, the

heating electrode of the working OG will hang in air, and the bottom components of the OGs (like magnet, and PDMS ring) are approaching to room temperature, as shown in Supplementary Figs. 18b, c. As shown in Supplementary Fig. 17, the OGs in the Device 1 couldn't contact with human skin directly, meaning that the Device 1 has no safety issue during application. For the Device 2, the working OGs may have the potential to contact with the human skin during operation. Therefore, we call for 11 volunteers (4 males and 7 females), and measure the shortest distance between their nose and the working OGs as they are wearing the Device 2. The results have been shown in Supplementary Fig. 18. The two side-view thermal distributions of a working OG illustrate a fact that the air temperature 1.5 mm above the OG is approaching to room temperature, which is safe to human being. The minimum distance values collected from the 11 volunteers is 2 mm, larger than 1.5 mm, which means that the Device 2 is safe to utilize. We have modified the text on Page 13 Line 23.

Modifications: On Page 13 Line 23, the text has been modified as “**Supplementary Fig. 16** demonstrates the high stability of Device 1, where the device is fixed on a programmable bending platform for 2000 continuous bending cycles at the constant frequency and angle of 0.33 Hz and 40°. During the bending process, Device 1 wirelessly sends the voltage value into the electromagnet coils of OGs integrated to the paired receiver, and stable voltage input (3 ± 0.02 V) proves the excellent stability of Device 1. To demonstrate the flexibility of the Device 1, the device is mounted onto human upper lip with different mouth motions, and enlarged optical image details proves the tight attachment of the device onto human skin during various skin deformations (**Supplementary Fig. 17**). **Supplementary Fig. 18** shows the safety testing of the olfactory devices as the heating temperatures of the OGs are stabilized around 60 °C. The thermal images of a working OG illustrate a fact that the ambient air temperature with distance of 1.5 mm from the working OG is around at room temperature (**Supplementary Figs. 18b, c**). Then, we measured the distance between human nose to working OGs from 11 volunteers (**Supplementary Fig. 18d**). It is clear that all volunteers (including 4 males and 7 females) could safely wear the Device 2 with all the distance values greater than 1.5 mm (**Supplementary Figs. 18e, f**). For the Device 1, the open design allows the OGs to exhibit good thermal convection and also to maintain a safe distance to users with a big gap (23 mm) between OGs and user's nose, since the temperature at the skin interface is only 32.2 °C, as shown in **Supplementary Figs. 18g, h**.”

Fig. S17. Optical images of Device 1 mounted onto human upper lip. As users do various mouth motions, the Device 1 can be still tightly mounted onto the skin.

Fig. S18. Safety operation of Device 2 when users are wearing Device 2. (a) A OG 3D model built up in software ABAQUS (Analysis User's Manual 6.14). (b, c) Two views of thermal distribution of a working OG with the working temperature is target at 60 °C, where the working temperature is the one around the thermistor for melting odorous wax insides OGs. (d) Schematic diagram of the inside layout of Device 2 when users are wearing the Device 2. Here, the distance between users' nose and working OGs decides the safety of Device 2 during operation. (e, f) 11 volunteers' distance values between volunteers' nose and working OGs, including 4 males and 7 females. All volunteers can safely wear the Device 2 during its operation as the minimum distance value is 2 mm, where the air temperature is room temperature.

Comment 5: This OG hires paraffin wax containing the perfume. The odor resource is in an open system, not a closed reservoir. Therefore, it is required to quantitatively verify the slowly evaporated perfume under room temperature for an extended period.

Our response: We thank the referee for this useful comment. We have conducted two new experiment to test 9 frequently used perfumes duration time and the corresponding perfume/wax samples storage time, as shown in Supplementary Figs. 24 and 25. As shown in Supplementary 24, the 9 pure perfumes duration time ranges from 3.9 h to 141.8 h on average. Here, the 9 perfumes are lavender, orange, pineapple, green tea, lemon, peach, strawberry, minty, and lilac, which is used in Fig. 4, and Supplementary Figs. 11, 12, 13, 19, 20, 24. Then, we conducted a volunteer test to test if the 9 different odorous wax samples can still generate odors after exposing in air at room temperature for three weeks, as shown in Supplementary Fig. 25. 11 volunteers are involved in this test. During the test, all volunteers are wearing the Device 2 with the 9 different odorous wax samples embedded. The heating temperature of OGs is 60 °C for rapidly releasing the odors. As a result, the average recognition rates for each odor types are 1 @ lavender, 0.82 @ orange, 1 @ pineapple, 1 @ green tea, 1 @ lemon, 0.91 @ peach, 1 @ strawberry, 0.83 @ minty, and 1 @ lilac. It is worth mention that the recognition rates of minty and orange are much lower than the other 7 odor types, which may be due to their good volatility (4 h @ minty and 3.9 h @ orange), as shown in Supplementary Fig. 24. We have modified the text on Page 25 Line 9.

Modifications: On Page 25 Line 9, the text has been modified as “**Characterization.** The perfumes adopted in Extended Data Fig. 3d are minty (perfume 1) and jasmine incense (perfumer 2). The odorous liquids used in Fig. 4b are lavender (a) @ walking outdoors(A), sweet orange (b) @ happiness (B), pineapple (c) having a lunch@ (C), green tea (d) @ teatime (D), lemon (e) @ summer (E), peach (f) @ sweety (F), strawberry (g) @ lovely (G), minty (h) @ teeth brushing (H), and lilac (i) @ pure (I). Here, we measured the odor retention time for each pure perfumes shown Fig. 4b, ranging from 3.9 h to 141.8 h (**Supplementary Fig. 24**), and corresponding odor types and components have been summarized in **Supplementary Table 2**. By mixing the perfumes and wax at a heating temperature of 60 °C, 9 different odorous wax samples can be obtained, which can be still functional after exposing in air for three weeks (**Supplementary Fig. 25**). The odorous liquids used in Fig. 4c are ethanol (1), pineapple (2), grape (3), minty (4), rice (5), cream (6), gardenia (7), watermelon (8), vanilla (9), coffee milk (10), candy (11), coconut milk (12), coconut (13), milk (14), peach (15), pancake (16), orange (17), green tea (18), caramel (19), durian (20), lemon (21), strawberry (22), morning (23), ginger (24), clary sage (25), rosemary (26), lavender (27), clove (28), mojito (29), and cake (30).”

Fig. S24. The retention time of 9 pure perfumes. Here, a, b, c, d, e, f, g, h, and i stand for lavender, orange, pineapple, green tea, lemon, peach, strawberry, minty, and lilac.

Fig. S25. An volunteer test showing the 11 volunteers' recognition rates to the 9 different odorous wax samples, which is prepared three weeks ago, then exposing in air at room temperature with the relative humidity of 60%. During the testing, all volunteers are required to wear the Device 2 with the 9 different wax samples in each OGs. Here, OGs heating temperature is 60 °C. For odor types, a, b, c, d, e, f, g, h, and i stand for lavender, orange, pineapple, green tea, lemon, peach, strawberry, minty, and lilac.

Comment 6: Could the volunteers who put the device below their nose feel any perfume odor when it is not working? You described the human odor threshold as 80 ppm. So, if the perfume gas concentration value is shown at room temperature, the reader can check it.

Our response: We thank the referee for this useful comment. It is worth mention that the human ethanol odor threshold is 80 ppm, not all odor types. To investigate if users could sense the odor in OGs at room temperature, we conducted two new experiments, as shown in Extended Data Fig. 4d and Supplementary Fig. 20.

For ethanol odor, we use a commercial ethanol sensor to measure the ethanol concentration variation as the heating temperature of OG recovers from high temperature to 45 °C, as shown in Extended Data Fig. 4d. The stabilized ethanol concentration could reach to 47 ppm, far away from the human ethanol odor threshold value. Therefore, users could not sense the ethanol odor if the corresponding OG doesn't operate. As the room temperature is much lower than the stand-by heating temperature (45 °C) of OGs, the ethanol concentration will not exceed 47 ppm.

For the 9 perfume-based odor types, we conducted a volunteer sensory test, where 11 volunteers are required to smell the OGs at room temperature with 9 different odor types embedded, and the result shows that the average possibility of sensing the odors in OGs is 0.43, and it is also found that different volunteers and different odor types both have different recognition rates. For example, the possibility of sensing the minty is 0.09, but the possibility of the lavender can be up to 0.82. It is worth mention that the low-volatility orange and minty have low recognition rates, including 3.9 h (perfume retention time) @ 0.27, and 4 h @ 0.09, as shown in Supplementary Figs. 20, 24, and the high-volatility peach (92.5) has a high recognition rate (0.82). Therefore, the volatility of odor types may influence the odor recognition rate at room temperature. In the future, we consider adding a miniaturized mechanical structure to enhance the odor releasing and closing capability, which may decrease the possibility of sensing some specific odors in room temperature. We have modified the text on Page 15 Line 5.

Modifications: On Page 15 Line 5, the text has been modified as “Here, we conducted a volunteer sensory test to investigate if users could react to fast temperature variation of OGs integrated in Device 1 (**Supplementary Fig. 19**). Following the time points in increasing the heating temperature of OGs shown in **Fig. 4aⁱⁱⁱ and iv**, all volunteers’ reaction times to varying heating temperature of two OGs in the Device 1 are recorded in **Fig. 4a^{iv} and Supplementary Fig. 19**, including 9.7 s (reaction time) @ 5 s (time point in increasing heating temperature), 20.8 s @ 15 s, and 28.4 s @ 25 s. As a result, our Device 1 could generate distinguished odor concentration to users, further demonstrating the great potential in movie watching. **Supplementary Fig. 20** shows a volunteer sensory test result in judging presence or absence of odors in silent OGs at room temperature. It is found that the recognition rates range from 0.22 to 0.89 with an average value of 0.43 for all volunteers, which demonstrates that human beings have distinguished odor concentration thresholds. In addition, different odor types also have distinguished recognition rate, ranging from 0.09 to 0.82.”

On Page 25 Line 9, the text has been modified as “**Characterization.** The perfumes adopted in Extended Data Fig. 3d are minty (perfume 1) and jasmine incense (perfumer 2). The odorous liquids used in Fig. 4b are lavender (a) @ walking outdoors(A), sweet orange (b) @ happiness (B), pineapple (c) having a lunch@ (C), green tea (d) @ teatime (D), lemon (e) @ summer (E), peach (f) @ sweety (F), strawberry (g) @ lovely (G), minty (h) @ teeth brushing (H), and lilac (i) @ pure (I). Here, we measured the odor retention time for each pure perfumes shown Fig. 4b, ranging from 3.9 h to 141.8 h (**Supplementary Fig. 24**), and corresponding odor types and components have been summarized in **Supplementary Table 2**. By mixing the perfumes and wax at a heating temperature of 60 °C, 9 different odorous wax samples can be obtained, which can be still functional after exposing in air for three weeks (**Supplementary Fig. 25**). The odorous liquids used in Fig. 4c are ethanol (1), pineapple (2), grape (3), minty (4), rice (5), cream (6), gardenia (7), watermelon (8), vanilla (9), coffee milk (10), candy (11), coconut milk (12), coconut (13), milk (14), peach (15), pancake (16), orange (17), green tea (18), caramel (19), durian (20), lemon (21), strawberry (22), morning (23), ginger (24), clary sage (25), rosemary (26), lavender (27), clove (28), mojito (29), and cake (30).”

Fig. S20. A volunteer testing showing the recognition rate of volunteers in smelling the OGs at room temperature when they are the Device 1. Here, a, b, c, d, e, f, g, h, and i stand for lavender, orange, pineapple, green tea, lemon, peach, strawberry, minty, and lilac.

Fig. S24. The retention time of 9 pure perfumes. Here, a, b, c, d, e, f, g, h, and i stand for lavender, orange, pineapple, green tea, lemon, peach, strawberry, minty, and lilac.

Comment 7: To help understand the reader, can you express the temperature change as time-dependent data in Extended Data Figure 31?

Our response: We thank the referee for this useful comment. We have remeasured the ethanol concentration generated by OGs for minimizing the potential ambient wind effect on the generated ethanol concentration fluctuation, as shown in Extended Data Fig. 4d. The corresponding heating temperature of the OG is also remeasured, as shown in Supplementary Fig. 10. It is obvious that the higher heating temperature of the working OG could contribute to higher ethanol concentration. It is worth mention that we conducted the test in an open box for minimizing the unpredictable ambient wind, which may have a large influence on the ethanol concentration results, proven in Extended Data Fig. 4e. We have modified the test on Page 10 Line 22.

Modifications: On Page 10 Line 22, the text has been modified as “**Extended Data Fig. 4** shows the gaseous ethanol generation performance of the OGs. To minimize the unpredictable ambient wind effect on the performance of the OG, the whole experimental setup is placed inside an open box, where a commercial ethanol sensor is fixed 1 cm above a working OG, continuously monitoring the nearby ethanol concentration by reading the voltage variation of a 5 k Ω resistor connected in series (**Extended Data Figs. 4a, b, c**). By switching the heating temperature of the OG from 45 °C to 50 °C, to 55 °C, and finally to 60 °C (**Supplementary Fig. 10**), the ambient ethanol concentrations could reach up to 531 ppm, 2821 ppm, and 4531 ppm, respectively, where the corresponding response times in raising up and recovering are 4 s @ 40 s (recovery time, RT), 7 s @ 72 s, and 8 s @ 129 s (**Extended Data Fig. 4d**). Since the human smell threshold for ethanol concentration is 80 ppm (33), it is obvious that higher heating temperature could contribute stronger olfactory feedback. In addition to the ethanol concentration measurement, we also conducted a human sensory test to investigate the heating temperature effect on the generated odor concentration, as shown in **Supplementary Fig. 11**. During the testing, all volunteers were wearing the Device 2 with 9 working OGs integrated for testing their responses to 9 different odors, where the temperatures of each OG was increased from 45 °C to 50 °C, 55 °C, and 60 °C with the lasting time of 1 min for each temperature point. By summarizing all volunteers’ response, we found that the recognition rates to these smells range from 0.73 to 1 with the average value of 0.93, further proving that the heating temperature of OGs is key for recognize odor concentration, and the heating temperatures ranging from 45 °C to 60 °C are sufficient to generate odors with the concentrations much larger than the human thresholds.”

Fig. S10. Electrical response of a working OG with the heating temperature switching between 45 °C and 50 °C, 55 °C, and 60°C with the generated ethanol concentration shown in Extended Data Fig. 4d.

Comment 8: Also, as seen in the Extended Data Figure 3l, the concentration of emitted ethanol gas seems unstable despite keeping the same temperature. Can you explain the instability of measured gas concentration? The author has exhibited stable and reliable temperature data that is not the output data the user wants to know (perfume gas concentration) in Figures 4a, e, and 5d. Therefore, it should be clearly explained to prove the reliability and fidelity of this OG system employing the heater and paraffin.

Our response: We thank the referee for this useful comment. The big ethanol concentration fluctuation shown in the original Extended Data Fig. 3l may be induced by the unpredictable ambient wind, which is later proven in new Extended Data Fig. 4e. To minimize the ambient wind influence, we remeasured the ethanol concentration as a function of the heating temperature of OGs in an open box with walls surrounded, as shown in Extended Data Figs. 4a, b. Benefitted from the new testing condition, by switching the heating temperature of the OG between 45 °C and 50 °C, 55 °C, and 60 °C (Supplementary Fig. 10), the ambient ethanol concentrations could reach up to 531 ppm, 2821.5 ppm, and 4531 ppm, respectively, with small odor concentration fluctuations (Extended Data Fig. 4d).

For the second question, we have conducted a series of ethanol concentration tests and volunteer sensory tests for investigating the odor generation performance of the olfaction interfaces, as shown in Fig. 4, Extended Data Fig. 4, and Supplementary Figs. 11, 12, 13, 19, 20. For ethanol concentration measurement in Extended Data Fig. 4, we investigate three potential effects on ethanol generation performance of OGs with paraffin/ethanol embedded, including the heating temperature of OGs (Extended Data Fig. 4d), ambient wind speed (Extended Data Fig. 4e), and distance between working OGs and the ethanol sensor (Extended Data Fig. 4f), where all the tests were performed in an open box for minimizing the unpredictable ambient wind around working OGs. It is found that increasing heating temperature of OGs could contribute higher ethanol concentration, higher ambient wind could induce faster ethanol recovery time (RT), and larger distance between OGs and the ethanol sensor will take a longer time to detect the gas, meaning a longer delay time. We also measured the remaining time of ethanol in Device after the Device 2 releasing ethanol for 5 mins, as shown in Extended Data Figs. 4g, h, i, j. Here, we setup two test conditions: Condition one is that the experimenter starts wearing the Device 2 after it releasing ethanol for 5 mins. Condition two is that the Device 2 keeps static in the open box after releasing ethanol for 5 mins. It is obvious that Condition one takes a shorter recovery time than Condition two, which is induced

by the human breathing. Extended Data Fig. 4 shows the electrical response of the ethanol sensor embedded in the Device 2 as the experimenter is wearing it, where the heating temperature of the working OG in Device 2 is target at 60 °C. It is found that human breathing could decrease the ethanol concentration around the sensor with obvious concentration fluctuation (± 180 ppm).

For the fidelity issue, we also conducted a series of volunteer sensory tests, as shown in Fig. 4, Supplementary Figs. 11, 13, 19, 20. The reason why we conducted these volunteer sensory tests is that we couldn't find specific commercial odor sensors to measure perfumes-based odor concentrations generated by OGs. All volunteer tests call for 11 volunteers with 4 males and 7 females. The volunteer test shown in Supplementary Fig. 11 is to investigate if volunteers could obviously sense the enhanced odor concentrations for 9 different odor types (see details in Characterization) generated by Device 2 when the heating temperature of OGs are increased from 45 °C to 50 °C, 55 °C, and 60 °C step by step. The high average recognition rate, 0.93, demonstrates that all volunteers confirm to sense the odor concentration variations by Device 2. We also measured the odor remaining time in Device 2 after the device release odors for 5 mins, where we setup two testing conditions. Condition One is that volunteers continuously smell the odors when wearing the Device 2. Condition Two is that the volunteers come to smell the odors of the face mask every 10 s. Then, we recorded the time for 9 different odors in the two testing conditions when the volunteers couldn't sense the corresponding odors, with the result shown in Supplementary Fig. 13. Supplementary Fig. 19 is another one volunteer sensory test, focusing on recording the human reaction to the fast odor concentration variation realized by the heating temperature variation of OGs in Device 1. During the test, the Device 1 will be programmed to increase temperatures of OGs from 45 °C to 50 °C at 5 s, from 50 °C to 55 °C at 15 s, from 55 °C to 60 °C at 25 s, then lasting 1 min until shutting down Device 1. Once the volunteers can sense the enhanced odor concentration, the time point will be recorded meanwhile. Two different odor types for two OGs in Device 1 are adopted, including lavender and lilac. Supplementary Fig. 20 shows a new volunteer test for confirming if volunteers could sense the odors by OGs at room temperature, and it is found that each volunteers have distinguished odor concentration thresholds to various odor types with the recognition rate ranging from 0.22 to 0.89. As a result, volunteers can obviously sense some specific odor types, such as lavender, strawberry, and lilac, but the possibility in sensing minty, orange, green tea, lemon, and peach is low, ranging from 0.27 to 0.09. Finally, we collected all volunteers' emotion reactions to 30 different odors generated by Device 2 (see details in Characterization), as shown in Fig. 4c. It is found that the possibility of being joyful for these volunteers could reach up to 56%, 65%, 44%, and 56% when they smelled the grape, peach, orange, and strawberry odor, respectively. In addition, it is obvious that different users have distinguished emotion reactions to the 30 odors (Figs. 4cv, vi), which means that users could select their preferred odor types integrated insides Device 2 for emotion smoothing. Finally, we reorganize the application data in Fig. 4. We have modified the text on Page 16 Line 11.

Modifications: On Page 16 Line 11, the text has been modified as “**Fig. 4c** shows another application scenario where some odors generated from the olfactory interface system could be used for smoothing users' emotion. As odors could arouse human emotion by leading to the recall of emotional memories, the olfactory interface could be adopted for smoothing users' depressed mood from the stress (23) (**Fig. 4ci**). To verify the effect of odors on human emotion, 11 volunteers' reactions to 30 different odors (see details in Characterization) from the face mask based olfactory interface were collected as shown in **Fig. 4c** (all volunteers were in normal emotion before the testing, **Figs. 4cii and iii**), and the result demonstrates that the possibility of

being joyful for these volunteers could reach up to 56%, 65%, 44%, and 56% when they smelled the grape, peach, orange, and strawberry odor, respectively (**Figs. 4civ, v**). It is also possible to give other emotions to users as they smelled clove and gardenia odors, inducing repulsive and peaceful emotions with the corresponding possibility of 0.82 and 0.51, as shown in **Fig. 4civ**. In addition, it is obvious that different users have distinguished emotion reactions to the same odors (**Figs. 4cv, vi**), which may be due to the volunteers' different emotional memories for same odor type. The large emotion reaction difference for each user may increase the difficulty in giving various desired emotions to users. **Fig. 4cv** shows the volunteers' emotion reaction to 30 odor types, where the results were collected from five times test for each volunteer. It is found that for each volunteer, their emotion reactions to some odor types keep constant, demonstrating that we could customize the odor types in the Device 2 for different users."

On Page 10 Line 22, the text has been modified as "**Extended Data Fig. 4** shows the gaseous ethanol generation performance of the OGs. To minimize the unpredictable ambient wind effect on the performance of the OG, the whole experimental setup is placed inside an open box, where a commercial ethanol sensor is fixed 1 cm above a working OG, continuously monitoring the nearby ethanol concentration by reading the voltage variation of a 5 k Ω resistor connected in series (**Extended Data Figs. 4a, b, c**). By switching the heating temperature of the OG from 45 °C to 50 °C, to 55 °C, and finally to 60 °C (**Supplementary Fig. 10**), the ambient ethanol concentrations could reach up to 531 ppm, 2821 ppm, and 4531 ppm, respectively, where the corresponding response times in raising up and recovering are 4 s @ 40 s (recovery time, RT), 7 s @ 72 s, and 8 s @ 129 s (**Extended Data Fig. 4d**). Since the human smell threshold for ethanol concentration is 80 ppm (33), it is obvious that higher heating temperature could contribute stronger olfactory feedback. In addition to the ethanol concentration measurement, we also conducted a human sensory test to investigate the heating temperature effect on the generated odor concentration, as shown in **Supplementary Fig. 11**. During the testing, all volunteers were wearing the Device 2 with 9 working OGs integrated for testing their responses to 9 different odors, where the temperatures of each OG was increased from 45 °C to 50 °C, 55 °C, and 60 °C with the lasting time of 1 min for each temperature point. By summarizing all volunteers' response, we found that the recognition rates to these smells range from 0.73 to 1 with the average value of 0.93, further proving that the heating temperature of OGs is key for recognize odor concentration, and the heating temperatures ranging from 45 °C to 60 °C are sufficient to generate odors with the concentrations much larger than the human thresholds. **Extended Data Fig. 4e** shows the recovery time of generated ethanol concentration in air as a function of parallelly blown wind speed, ranging from 0 to 6.61 m/s, corresponding to a decreasing RT from 129 s to 9 s, which is induced by the increasing gas diffusion rate triggered by ambient wind. **Extended Data Fig. 4f** presents the electrical response of the commercial ethanol sensor as a function of the distance between the sensor and a working OG, and it is clear that longer distance could result in a higher delay time to trigger the sensor, including 1 cm @ 1.2 s, 3 cm @ 9.1 s, and 5 cm @ 15.6 s, which is further verified by the volunteer sensory test shown in **Supplementary Fig. 12**. To further investigate the odor generating performance of OGs inside Device 2, the ethanol sensor is embedded in Device 2 near to the OG integrating paraffin/ethanol (**Extended Data Figs. 4g, h**). Here, two different testing conditions are introduced: Condition one is that Device 2 releases ethanol for 5 mins, then shut down meanwhile an experimenter starts wearing the Device 2 until the monitored ethanol concentration recovers to original state (lower than 80 ppm), as shown in **Extended Data Figs. 4i, j**; Condition two is that the Device 2 releases ethanol for 5 mins, then shut down without further motion (**Extended Data Figs. 4i, j**), where it is obvious that human breathing

could shorten ethanol recovery time (1.2 min @ Condition one and 3.1 min @ Condition two). To further investigate human breathing effect on generated odor recovery time, we conducted a volunteer test (**Supplementary Fig. 13**), that also associates with two testing conditions same as that in **Extended Data Figs. 4i, j**. Here, 9 different odors are adopted including lavender (a), orange (b), pineapple (c), green tea (d), lemon (e), peach (f), strawberry (g), minty (h), and lilac (i). The average recovery time of Condition one is 92.1 s, larger than 77.1 s of Condition two, consistent with the conclusion shown in **Extended Data Figs. 4i, j**. As a result, it is obvious that the formerly generated odors will not interface with the newly generating odors after the corresponding odor recovery time, which is influenced by odor types and ambient airflow rate (**Extended Data Fig. 4 and Supplementary Fig. 13**). **Extended Data Figs. 4k, l** shows ethanol concentration generated by Device 2 as an experimenter is wearing the device, where the heating temperature of the working OG is target at 60 °C. Continuous breathing could obviously decrease the ethanol concentration to 1338 ppm around the ethanol sensor with a large fluctuation (± 180 ppm), which also demonstrates that the ethanol generation rate at the heating temperature of 60 °C is larger than human odor inhalation rate. As the generated odor concentration can be adjusted (**Extended Data Fig. 4d and Supplementary Fig. 11**), it is possible to avoid odor accumulation around human nose by adjusting the heating temperature of corresponding OGs in the Device 2.”

On Page 15 Line 5, the text has been modified as “Here, we conducted a volunteer sensory test to investigate if users could react to fast temperature variation of OGs integrated in Device 1 (**Supplementary Fig. 19**). Following the time points in increasing the heating temperature of OGs shown in Fig. 4aiii and iv, all volunteers’ reaction times to enhanced odor concentration are recorded in **Supplementary Fig. 19**, including 9.7 s (reaction time) @ 5 s (time point in increasing heating temperature), 20.8 s @ 15 s, and 28.4 s @ 25 s, which further demonstrates the great potential in movie watching. **Supplementary Fig. 20** shows a volunteer sensory test result in judging presence or absence of odors in silent OGs at room temperature. It is found that the recognition rates range from 0.22 to 0.89 with an average value of 0.43 for all volunteers, which demonstrates that human beings have distinguished odor concentration thresholds. In addition, different odor types also have distinguished recognition rate, ranging from 0.09 to 0.82.”

On Page 25 Line 9, the text has been modified as “**Characterization**. The perfumes adopted in Extended Data Fig. 3d are minty (perfume 1) and jasmine incense (perfumer 2). The odorous liquids used in Fig. 4b are lavender (a) @ walking outdoors(A), sweet orange (b) @ happiness (B), pineapple (c) having a lunch@ (C), green tea (d) @ teatime (D), lemon (e) @ summer (E), peach (f) @ sweetly (F), strawberry (g) @ lovely (G), minty (h) @ teeth brushing (H), and lilac (i) @ pure (I). Here, we measured the odor retention time for each pure perfumes shown Fig. 4b, ranging from 3.9 h to 141.8 h (**Supplementary Fig. 24**), and corresponding odor types and components have been summarized in **Supplementary Table 2**. By mixing the perfumes and wax at a heating temperature of 60 °C, 9 different odorous wax samples can be obtained, which can be still functional after exposing in air for three weeks (**Supplementary Fig. 25**). The odorous liquids used in Fig. 4c are ethanol (1), pineapple (2), grape (3), minty (4), rice (5), cream (6), gardenia (7), watermelon (8), vanilla (9), coffee milk (10), candy (11), coconut milk (12), coconut (13), milk (14), peach (15), pancake (16), orange (17), green tea (18), caramel (19), durian (20), lemon (21), strawberry (22), morning (23), ginger (24), clary sage (25), rosemary (26), lavender (27), clove (28), mojito (29), and cake (30).”

Fig. 4. Demonstrations of the olfaction interfaces. **a** A demonstration of the skin-integrated Device 1 in displaying olfaction feedback for providing an immersive experience to users during movie watching. Here, a girl in the movie is smelling a rose with the decreasing distance between the flower and her while the watcher could smell a more and more intense rose odor by wearing the olfaction interface. Here, OCL demonstrates the odor concentration level ranging from 1 to 3, corresponding to the low, middle, and high levels (iv). **b** A demonstration of the face-mask based olfaction interface in providing an alternative communication method, smell messages, for the disabled without the capability of vision and audio. By training users to link odors to specific information, the disabled users could efficiently communicate with others. **c** A demonstration of the Device 2 in assisting users to control their emotions as some specific odors could recall different human emotions. Here, all volunteers involved in the

emotion reaction test are verified in normal emotion range (**cii and ciii**). **d** A potential demonstration of the Device 2 for helping amnesic patients recall lost memories as odors perception is modulated by experience, leading to the recall of emotional memories. **e** Demonstration of the wireless olfaction interfaces in real time interaction between the user and a virtual subject for walking in a virtual garden surrounded by various fruit fragrance. By programming the Device 2, 9 OGs insides the device could work independently (**eii**), or work together (**eiii**).

Extended Data Fig. 4. Ethanol concentration performance of the odor generators. (**a, b, c**) Ethanol concentration testing setup, where the test is placed inside an open box for minimizing the ambient wind interference to OGs. During the testing, a commercial resistance-variation-based ethanol sensor placed 1 cm above a working OG for low response time to generated ethanol around the OG. (**d**) Ethanol concentration generated by OG with 30 mg paraffin/ethanol (mass ratio, 10:3) with a controlled temperature switching between 45°C to 50°C, 55°C, and

60°C, respectively, corresponding to the increasing ethanol concentration peak values. (e) Ambient wind effects on the ethanol concentration dissipation rate, here an operating OG with a heating temperature of 60°C are suddenly shut down to cut off ethanol generation, and the wind is blown above the OG parallelly. (f) Delay time of a working OG as a function of the distance between the generator and ethanol sensor, which is induced by the ethanol diffusion rate. (g, h, i, j, k, l) Ethanol remaining time test in Device 2 with three testing conditions. Condition one is that experimenter continuously smell the generated ethanol after the OG with paraffin/ethanol embedded works for 5 min at a constant heating temperature of 60°C when he is wearing the Device 2, and Condition two is that the ethanol sensor monitors the ethanol concentration after the OG works for 5 min in an open box (i, j). Condition three is that the ethanol sensor continuously monitors gas concentration as the experimenter is wearing the Device 2 (k, l).

Fig. S11. A volunteer testing showing the volunteers' odor recognition rate when they are wearing the Device 2 with 9 different perfume-based odors (see details in Characterization). Here, the temperature of each OGs will be increased from 45 °C to 60 °C at an interval of 5 °C, where each temperature will last 1 min before going up. The, the volunteers will be asked if they could sense the enhanced odor concentration.

Fig. S13. A volunteer testing showing the odor remaining time in Device 2. Here, all volunteers will be asked to smell the remaining odors in the Device 2, and record the time when the odors disappear. There are two testing conditions. Condition One is that volunteers continuously smell the odors when wearing the Device 2. Condition Two is that the volunteers come to smell the odors of the face mask every 10 s. Before volunteer smells the Device 2, the OG will generate the odor at a constant heating temperature of 60 °C in the face mask for 5 min.

Fig. S19. A volunteer testing showing the human reaction to the fast temperature variation of OGs with two odor types embedded when they are wearing Device 1 for testing. During the test, the Device 1 will be programmed to increase temperatures of OGs from 45 °C to 50 °C at 5 s, from 50 °C to 55°C at 15 s, from 55 °C to 60 °C at 25 s, then lasting 1 min until shutting down Device 1. Once the volunteers can sense the enhanced odor concentration, the time point will be recorded meanwhile. Two different odor types for two OGs in Device 1 are adopted, including lavender and lilac.

Fig. S20. A volunteer testing showing the recognition rate of volunteers in smelling the OGs at room temperature when they are the Device 1. Here, a, b, c, d, e, f, g, h, and i stand for lavender, orange, pineapple, green tea, lemon, peach, strawberry, minty, and lilac.

REVIEWER COMMENTS

Reviewer #1 (Remarks to the Author):

The authors have addressed all my comments comprehensively. I think the current version is with good quality to be accepted by the Nature Communications.

Reviewer #2 (Remarks to the Author):

The authors made much effort to improve the manuscript. However, the following questions still remain. The further improvement is required.

Answer to comment2

The authors added Extended data Fig.4 e to show the reduction of recovery time when the wind speed increases.

However, wind speed seems too strong. The player feels the wind except the case when the wind speed is zero. The authors should mention that the situation here is not practical if they want to keep this figure.

Extended Figure 4.h

Is it possible to monitor gas concentration when a player wears the olfactory display?

The sensor seems too large for placing it inside the mask.

Answers to comment2, comment4 and comment5

Fig,S13 shows odor remaining time. However, the recovery time is not sufficient. The authors should describe the interpretation of the figure and the technology limitation at the current stage.

Answer to comment8

Question is whether low-volatile compound with boiling point higher than 200 degree Celsius can be generated.

Answer to comment19

Odor sensor part is not related to this article. It should be removed. Moreover, the author should check the literatures in odor generator part again to describe more accurate information.

Reviewer #3 (Remarks to the Author):

All responses to the review comments are satisfactory.

Responses to comments of Referee #1

Comments from Referee #1:

Summary Comment: The authors have addressed all my comments comprehensively. I think the current version is with good quality to be accepted by the Nature Communications.

Our response: We thank the referee for the positive comments.

Modifications: None

Responses to comments of Referee #2

Comments from Referee #2:

Summary Comment: The authors made much effort to improve the manuscript. However, the following questions still remain. The further improvement is required.

Our response: We thank the referee for these positive comments. We carefully addressed the issues, as listed below, and we revised our manuscript accordingly.

Modifications: None

Comment 1: Answer to comment 2. The authors added Extended data Fig.4 e to show the reduction of recovery time when the wind speed increases. However, wind speed seems too strong. The player feels the wind except the case when the wind speed is zero. The authors should mention that the situation here is not practical if they want to keep this figure. Extended Figure 4.h. Is it possible to monitor gas concentration when a player wears the olfactory display?

The sensor seems too large for placing it inside the mask.

Our response: We thank the referee for this comment. For the first question, in this paper, we developed two olfaction interfaces with five potential applications ranging from 4D movie watching, to smell message delivery, emotion influence, medical treatment, and VR, which covers both indoor and outdoor applications. For example, smell message delivery application may happen outdoors, and 4D movie watching application is an indoor activity. For indoor applications, the airflow rate can reach up to 1.92 m/s reported in the reference (*Building and Environment*, 198, 107887). For outdoor applications, the maximum tested airflow (6.61 m/s) is in Beaufort wind scale 4 (*Beaufort wind force scale. Met Office. Retrieved 27 November 2015*), which will not induce any discomfort to users. We discussed this issue in the revised manuscript, and the corresponding comments can be found on Page 11 Line 16.

For the second question, it is possible to monitor ethanol gas concentration when a player wears the olfaction interface, as shown in Supplementary Fig. S13 (newly added fig). The commercial, miniaturized ethanol sensor can be placed inside the Device 2 for directly monitoring the generated ethanol gas concentration during application. The integrated circuit for collecting data from the commercial ethanol sensor is embedded inside the power management system, which can directly power the sensor system, as shown in Supplementary Fig. 13. However, there are no commercial, miniaturized sensor for real time monitoring the generated perfume based odor gas concentration. As a result, it is possible to integrate the ethanol sensor with the olfaction interface system to monitor the working status. In the future, we plan to develop various miniaturized gas sensors to directly integrate with each OGs for real time monitoring the generated gas concentration. We put comments on this point on Page 11 Line 24.

Modifications: On Page 11 Line 24, we have modified the text as “To further investigate the odor generating performance of OGs inside Device 2, the ethanol sensor can be also embedded in Device 2 near to the OGs (**Supplementary Fig. S13 and Extended Data Figs. 4g, h**).”

On Page 11 Line 16, we have modified the text as “Extended Data Fig. 4e shows the recovery time of generated ethanol concentration in air as a function of parallelly blown wind speed, ranging from 0 to 6.61 m/s, corresponding to a decreasing RT from 129 s to 9 s, which is induced by the increasing gas diffusion rate triggered by ambient wind. Since the indoor wind speed can reach up to 1.92 m/s, the odor recovery time can be significantly shortened with the assistance of air flow (33). While using the olfaction interface outdoors is more favorable on the stand point of recovery time, as outdoor typically owns stronger wind.”

Fig. S13. Optical images of the Device 2 with the commercial ethanol sensor integrated.

The newly added references:

33. W. Li et al., *Effects of ceiling fans on airborne transmission in an air-conditioned space. Building and Environment* 198, 107887 (2021).

Comment 2: Fig. S13 shows odor remaining time. However, the recovery time is not sufficient. The authors should describe the interpretation of the figure and the technology limitation at the current stage.

Our response: We thank the referee for this comment. In the revised manuscript, we have added more interpretation of Supplementary Fig. 14 (originally Fig. S13 the reviewer mentioned). We list all odor recovery time for each odor types in two test conditions (Condition one and Condition two) as following “the average recovery times of lavender (a), orange (b), pineapple (c), green tea (d), lemon (e), peach (f), strawberry (g), minty (h), and lilac (i) are 114 s @ Condition one and 71 s @ Condition two, 91 s and 68 s, 80 s and 70 s, 64 s and 56, 84 s and 65 s, 77 s and 66 s, 135 s and 102 s, 83 s and 93 s, and 100 s and 99 s, respectively.” Except the minty odor type, the other 8 odor types show longer recovery time of Condition one than that of Condition two. In addition, the overall average recovery time of Condition one is 92.1 s, larger than 77.1 s of Condition two, consistent with the conclusion shown in **Extended Data Figs. 4i, j**. The comments can be found on Page 12 Line 7.

As the reviewer mentioned, the recovery time is not sufficient, which is one of the current limitations. The odor remaining time (over hundreds of seconds) will induce an obvious delay time for users when they need switching odor types frequently. To solve this issue, we

consider to miniaturize the overall size of OGs while increasing airtightness of the Device 2, which can provide comparable odor sensation to users with less generated odor gas, demonstrating a less odor remaining time. In addition, smaller size of OGs could realize a higher channel OGs array, providing more odor types in a limited application area around human nose. We have put the comment on Page 19 and Line 21.

Modifications: On Page 12 Line 7, we have modified the text as following “To further investigate human breathing effect on generated odor recovery time, we conducted a user study test (**Supplementary Fig. 14**), that also associates with two testing conditions same as that in **Extended Data Figs. 4i, j**. Here, as there are no available commercial odor sensors to monitor the odors concentration adopted in **Supplementary Fig. 14**, we adopted a volunteer test to obtain the odors recovery time with 11 volunteers involved. In **Supplementary Fig. 14**, 9 different odors are adopted including lavender (a), orange (b), pineapple (c), green tea (d), lemon (e), peach (f), strawberry (g), minty (h), and lilac (i) with the average recovery times of 114 s @ Condition one and 71 s @ Condition two for lavender, 91 s and 68 s for orange, 80 s and 70 s for pineapple, 64 s and 56 for green tea, 84 s and 65 s for lemon, 77 s and 66 s for peach, 135 s and 102 s for strawberry, 83 s and 93 s for minty, and 100 s and 99 s for lilac, respectively. Except the minty odor type, the other 8 odor types show longer recovery time of Condition one that of Condition two. In addition, the overall average recovery time of Condition one is 92.1 s, longer than Condition two of 77.1, which is consistent with the conclusion shown in **Extended Data Figs. 4i, j**. As a result, it is obvious that the formerly generated odors would not interference with the newly generating odors after the corresponding odor recovery time (**Extended Data Fig. 4 and Supplementary Fig.14**)”

On Page 19 Line 21, we have modified the text as following “The improvement of the system we could make in the future is illustrated in the following: The long odor remaining time could induce a long delay time when users need switching odor types frequently (**Extended Data Figs. 4d, e, i and Supplementary Fig. 14**). To solve this issue, miniaturizing the overall size of OGs and enhancing airtightness of the olfaction systems would be helpful. In addition, smaller size of OGs could also realize a higher channel OGs array.”

Comment 3: Question is whether low-volatile compound with boiling point higher than 200 degree Celsius can be generated.

Our response: We thank the referee for this useful comment. To investigate the question mentioned above, we conducted two experiments: One is that we measured the odor retention time of 30 odor types used in Fig. 4c, as shown in **Supplementary Fig. 26** at a heating temperature of 200°C. It is found that some odor types could continuously release odors for over 30 mins at a heating temperature of 200°C, and there are no boiling phenomena observed for these odor types, like mojito (**Supplementary Fig. 26**). The other one experiment is that we observe the physical status of the 30 odor types at a heating temperature of 240°C, and the optical images for each odor types have been summarized in **Supplementary Table 3**. The result has shown that some odor types will not boil at 240°C. In addition, 11 volunteers can obviously sense the 30 odors generated by our devices, as shown in Fig. 4c. As a result, we can conclude that our olfaction system could adopt low-volatile compound with the boiling point higher than 200°C as the odorous additive for generating smell to users. We have put the comment on Page 26 Line 9.

Modifications: On Page 26 and Line 9, we have modified the text as following “**Characterization.** The perfumes adopted in Extended Data Fig. 3d are minty (perfume 1) and jasmine incense (perfumer 2). The odorous liquids used in **Fig. 4b** are lavender (a) @ walking outdoors(A), sweet orange (b) @ happiness (B), pineapple (c) having a lunch@ (C), green tea (d) @ teatime (D), lemon (e) @ summer (E), peach (f) @ sweety (F), strawberry (g) @ lovely (G), minty (h) @ teeth brushing (H), and lilac (i) @ pure (I). Here, we measured the odor retention time for each pure perfumes shown in **Fig. 4b**, ranging from 3.9 h to 141.8 h (**Supplementary Fig. 25**), and corresponding odor types and components have been summarized in **Supplementary Table 2**. **Supplementary Fig. S26** shows the retention time of 30 different odor types adopted in **Fig. 4c** at a heating temperature of 200°C. Among the 30 odor types, a low-volatile odor compound with boiling point > 200°C (mojito/wax, 30 mg) could continuously release odor for 91 mins. **Supplementary Table 3** shows the physical status of the 30 odor types used in **Fig. 4c** at the heating temperature of 240°C, and it is obvious that some of odor types still don’t reach its boiling point at 240°C, including rice, vanilla, coffee milk, etc. It is concluded that our olfaction system could adopt low-volatile odor types as odorous additive into wax for providing olfaction display to users.”

Fig. S26. The retention time and optical images of 30 odor types adopted in Fig. 4c. From No. 1 to No. 30, the odor types are ethanol, pineapple, grape, mint, rice, cream, gardenia, watermelon, vanilla, coffee milk, candy, coconut milk, coconut, milk, peach, pancake, orange, green tea, caramel, durian, lemon, strawberry, morning, ginger, clary sage, rosemary, lavender, clove, mojito, and cake, respectively. Among the 30 odor types, some odor types could continuously release smell at a high heating temperature of 200°C for over 1 h, such as mojito and coffee milk. There is no boiling phenomenon observed for the mojito odor type. It is concluded that the mojito odor type is a low-volatile one with boiling point higher than 200°C. It has been proven that volunteers could obviously sense the mojito odor generated by Device 2 shown in Fig. 4c, therefore, it is proven that the low-volatile compound with boiling point higher than 200°C can be generated by adopting Device 2.

Table S3. Performance of 30 different odor types used in Fig. 4c at a heating temperature of 240°C.

No.	Odor types	Can it release odor at heating temperature of 60°C through OGS?	Boiling point	Optical images of the odorour wax at the heating temperature of 280°C
1	ethanol	Yes	78°C	2	pineapple	Yes	≤ 240°C	
3	grape	Yes	$\leq 240^{\circ}\text{C}$	4	mint	Yes	$> 240^{\circ}\text{C}$	5	rice	Yes	$> 240^{\circ}\text{C}$	6	cream	Yes	$> 240^{\circ}\text{C}$	7	gardenia	Yes	$\leq 240^{\circ}\text{C}$	8	Watermelon	Yes	$\leq 240^{\circ}\text{C}$	9	vanilla	Yes	$> 240^{\circ}\text{C}$	10	coffee milk	Yes	$> 240^{\circ}\text{C}$	11	candy	Yes	$> 240^{\circ}\text{C}$	12	Coconut milk	Yes	$> 240^{\circ}\text{C}$	13	Coconut	Yes	$> 240^{\circ}\text{C}$	14	milk	Yes	$> 240^{\circ}\text{C}$	15	peach	Yes	$\leq 240^{\circ}\text{C}$	16	pancake	Yes	$> 240^{\circ}\text{C}$	17	orange	Yes	$> 240^{\circ}\text{C}$	18	green tea	Yes	$\leq 240^{\circ}\text{C}$	19	caramel	Yes	$\leq 240^{\circ}\text{C}$	20	durian	Yes	$> 240^{\circ}\text{C}$	21	lemon	Yes	$> 240^{\circ}\text{C}$	22	strawberry	Yes	$\leq 240^{\circ}\text{C}$	23	morning	Yes	$> 240^{\circ}\text{C}$	24	ginger	Yes	$> 240^{\circ}\text{C}$	25	clary sage	Yes	$> 240^{\circ}\text{C}$	
26	Rosemary	Yes	> 240°C	27	Lavender	Yes	> 240°C	28	clove	Yes	> 240°C	29	mojito	Yes	> 240°C	30	cake	Yes	> 240°C	
Comment 4: Odor sensor part is not related to this article. It should be removed. Moreover, the author should check the literatures in odor generator part again to describe more accurate information.

Our response: We thank the referee for this useful comment. We have modified Supplementary Table 1 accordingly.

Modifications: We have modified Supplementary Table 1 as following:

Table S1. Comparison of the state-of-art olfactory interfaces with our work in the aspect of functional module, dimensions, scents, mechanical formats, response time, recovery time, communication method, and applications.

Functional Module	Dimensions (mm)	Odor options	Mechanical formats	Response time	Recovery time	Communication method	Application	Ref.
a surface acoustic wave device and micro-dispensing valves	N/A	1	rigid	10 s	N/A	Wired USB	VR game	(13)
mass flow controllers	N/A	3	rigid	2.5 s	N/A	Wired	olfactory virtual reality for mice	(14)
Commercial atomizer	L=500 W=500 T=400	3	rigid	5.8 s	N/A	Bluetooth	Improving users' sleep quality	(16)
Commercial atomizer	L=110 W=56 T=35	1	rigid	6 s	N/A	Wi-Fi	VR game	(17)
BFM based atomizer	N/A	4	rigid	3.1 s	5.16 s	Wired	VR technology	(18)
Commercial atomizer	N/A	1	rigid	N/A	N/A	Wired	VR game	(19)
Commercial atomizer	N/A	1	rigid	N/A	N/A	Wired	Olfaction display	(20)
Physical phase change of odorous paraffin wax	N/A	6	rigid and non-wearable	<180 s & >120 s	30	Bluetooth	Olfaction training	(21)
Commercial atomizer	L=67 W=60 T= 82	6	flexible, wearable	N/A	N/A	Bluetooth	VR game	(22)
Multisensory VR System	N/A	3	rigid	5.7 s	N/A	Wired	VR game	(23)
Direct-Injection	L=55 W=15 T=40	3	wearable	0.5 s	0.5 s	Wired USB	Olfaction training	(24)

SAW Atomizer	N/A	8	rigid	6 s	N/A	Wired	Olfactory display for multimedia content and virtual reality	(25)
Commercial atomizer	L=150 W=150 D=60	4	rigid	N/A	N/A	Bluetooth	museum exhibitions	(26)
Physical phase change of odorous paraffin wax by controlling the heating temperature.	L=18 W=16 T=3	9 basic types	flexible, wearable	1.44 s	40 s for ethanol gas with the peak concentration of 531 ppm.	Bluetooth	VR/AR, human-machine interface, medical treatment, 4D movie, and message delivery.	this work

The newly added reference:

22. S. Zou, X. Hu, Y. Ban, S. i. Warisawa, in *2022 IEEE Conference on Virtual Reality and 3D User Interfaces (VR)*. (IEEE, 2022), pp. 474-482.
23. A. Bahremand et al., in *2022 IEEE Conference on Virtual Reality and 3D User Interfaces (VR)*. (IEEE, 2022), pp. 241-249.
24. T. Yamada, S. Yokoyama, T. Tanikawa, K. Hirota, M. Hirose, in *IEEE Virtual Reality Conference (VR 2006)*. (IEEE, 2006), pp. 199-206.
25. T. Nakamoto, S. Ito, S. Kato, G. P. Qi, *Multicomponent olfactory display using solenoid valves and SAW atomizer and its blending-capability evaluation. IEEE Sensors Journal* 18, 5213-5218 (2018).
26. M. Bordegoni, M. Carulli, S. Bader, in *2019 IEEE International Symposium on Olfaction and Electronic Nose (ISOEN)*. (IEEE, 2019), pp. 1-3.

Responses to comments of Referee #3

Comments from Referee #3:

Summary Comment: All responses to the review comments are satisfactory.

Our response: We thank the referee for the positive comment.

Modification: None.

REVIEWERS' COMMENTS

Reviewer #2 (Remarks to the Author):

Answer to comment 3:

The authors made Table S3. But samples except No.1 seems mixture of several compounds.

Since the boiling point of the mixture cannot be defined, that word should be changed. I think just temperature is fine.

Other revisions are fine.

Responses to comments of Referee #2

Comments from Referee #2:

Comment: Answer to comment 3: The authors made Table S3. But samples except No.1 seems mixture of several compounds. Since the boiling point of the mixture can not be defined, that word should be changed. I think just temperature is fine. Other revisions are fine.

Our response: We thank the referee for this comment. We have modified the illustration of Table S3 and manuscript accordingly. We have put the comment on Page 25 Line 17.

Modifications: On Page 25 Line 17, we have modified the text as “**Supplementary Table 3** shows the physical status of the 30 odor types used in **Fig. 4c** at the heating temperature of 240°C, and it is obvious that some of odor types will not boil at 240°C, including rice, vanilla, coffee milk, etc. It is concluded that our olfaction system could adopt low-volatile odor types as odorous additive into wax for providing olfaction display to users.”

Table S3. Performance of 30 different odor types used in Fig. 4c at a heating temperature of 240°C.

No.	Odor types	Can it release odor at heating temperature of 60°C through OGs?	Heating temperature	Optical images of the odorour wax at the heating temperature of 280°C
1	ethanol	Yes	78°C	2	pineapple	Yes	240°C	3	grape	Yes	240°C	4	mint	Yes	240°C	5	rice	Yes	240°C	6	cream	Yes	240°C	7	gardenia	Yes	240°C	8	Watermelon	Yes	240°C	9	vanilla	Yes	240°C	10	coffee milk	Yes	240°C	11	candy	Yes	240°C	12	Coconut milk	Yes	240°C	
13	Coconut	Yes	240°C	14	milk	Yes	240°C	15	peach	Yes	240°C	16	pancake	Yes	240°C	17	orange	Yes	240°C	18	green tea	Yes	240°C	19	caramel	Yes	240°C	20	durian	Yes	240°C	21	lemon	Yes	240°C	22	strawberry	Yes	240°C	23	morning	Yes	240°C	24	ginger	Yes	240°C	25	clary sage	Yes	240°C	26	Rosemary	Yes	240°C	27	Lavender	Yes	240°C	28	clove	Yes	240°C	29	mojito	Yes	240°C	30	cake	Yes	240°C	
Responses to comments of Editor

Comments from Editor:

Comment: For Table S1, the description for reference 19 should be defined as “surface acoustic wave device and micro-dispensing valves” instead of commercial atomizer.

Our response: We thank the referee for this comment. We have modified the Table S1 accordingly. In addition, according to the Format requirement of Nature Communications, we have modified the reference number in Supplementary Information.

Modifications: The modified Table S1 has been shown in the following:

Table S1. Comparison of the state-of-art olfactory interfaces with our work in the aspect of functional module, dimensions, scents, mechanical formats, response time, recovery time, communication method, and applications.

Functional Module	Dimensions (mm)	Odor options	Mechanical formats	Response time	Recovery time	Communication method	Application	Ref.
a surface acoustic wave device and micro-dispensing valves	N/A	1	rigid	10 s	N/A	Wired USB	VR game	(1)
mass flow controllers	N/A	3	rigid	2.5 s	N/A	Wired	olfactory virtual reality for mice	(2)
Commercial atomizer	L=500 W=500 T=400	3	rigid	5.8 s	N/A	Bluetooth	Improving users' sleep quality	(3)
Commercial atomizer	L=110 W=56 T=35	1	rigid	6 s	N/A	Wi-Fi	VR game	(4)
BFM based atomizer	N/A	4	rigid	3.1 s	5.16 s	Wired	VR technology	(5)
surface acoustic wave device and micro-dispensing valves	N/A	1	rigid	N/A	N/A	Wired	VR game	(6)
Commercial atomizer	N/A	1	rigid	N/A	N/A	Wired	Olfaction display	(7)
Physical phase change of odorous paraffin wax	N/A	6	rigid and non-wearable	<180 s & >120 s	30	Bluetooth	Olfaction training	(8)
Commercial atomizer	L=67 W=60 T= 82	6	flexible, wearable	N/A	N/A	Bluetooth	VR game	(9)
Multisensory VR System	N/A	3	rigid	5.7 s	N/A	Wired	VR game	(10)
Direct-Injection	L=55 W=15 T=40	3	wearable	0.5 s	0.5 s	Wired USB	Olfaction training	(11)
SAW Atomizer	N/A	8	rigid	6 s	N/A	Wired	Olfactory display for multimedia content and virtual reality	(12)
Commercial atomizer	L=150 W=150 D=60	4	rigid	N/A	N/A	Bluetooth	museum exhibitions	(13)
Physical phase change of odorous paraffin wax by controlling the heating temperature.	L=18 W=16 T=3	9 basic types	flexible, wearable	1.44 s	40 s for ethanol gas with the peak concentration of 531 ppm.	Bluetooth	VR/AR, human-machine interface, medical treatment, 4D movie, and message delivery.	this work